# AWI-ICENet1: A convolutional neural network retracker for ice altimetry

Veit Helm[1], Alireza Dehghanpour[1,4], Ronny Hänsch[2], Erik Loebel[3], Martin Horwath[3], and Angelika Humbert[1,4]

[1]Alfred-Wegener-Institut Helmholtz-Zentrum für Polar- und Meeresforschung, Bremerhaven, Germany
[2]German Aerospace Centre, Germany
[3]Technische Universität Dresden, Institut für Planetare Geodäsie, Dresden, Germany
[4]University of Bremen, Department of Geosciences, Bremen, Germany

**Correspondence:** Veit Helm (veit.helm@awi.de)

**Abstract.** The Greenlandic and Antarctic Ice Sheet are important indicators of climate change and major contributors to sea level rise. Hence, precise, long-term observations of ice mass change are required to assess their contribution to sea level rise. This can be achieved by three different methods: Directly by measuring regional changes in the Earth's gravity field using the Gravity Recovery and Climate Experiment GRACE(FO) satellite system, or indirectly by measuring changes in ice thickness using satellite altimetry, or by estimating changes of the mass budget using a combination of regional climate model data output and ice discharge across the grounding line based on multi-sensor satellite radar observations of ice velocity (Hanna et al., 2013). Satellite radar altimetry has been used to measure elevation change since 1992 using a combination of various missions. It has been shown that, next to the surface slope and complex topography, one of the most challenging issues is the spatial and temporal variability of radar pulse penetration into the snow pack. This results in an inaccurate measurement of the true surface elevation and consequently affects surface elevation change (SEC) estimates. To increase the accuracy of surface elevations retrieved by retracking the radar return waveform and thus reduce the uncertainty in SEC, we developed a deep convolutional neural network architecture (AWI-ICENet1). The AWI-ICENet1 is trained using a simulated reference data set with 3.8 million waveforms, taking into account different surface slopes, topography, and attenuation. The successfully trained network is finally applied as AWI-ICENet1-retracker to the full time series of CryoSat-2 Low Resolution Mode (LRM) waveforms over both ice sheets. We compare the AWI-ICENet1 retrieved SEC with estimates of conventional retrackers like TFMRA and ESA ICE1 and ESA ICE2 products. Our results show less uncertainty and a greatly diminished effect of time variable radar penetration, reducing the need to apply corrections based on a close relationship with backscatter- and/or leading edge width, as typically done in SEC processing. This technique provides new opportunities to utilize convolutional neural networks in the processing of satellite altimetry data, which can be applied to historical, recent, and future missions.

## 1 Introduction

Ice sheet mass loss is a major contributor to sea level rise, as already the Greenland Ice Sheet (GrIS) has contributed $21.0 \pm 1.9$ mm (Otosaka et al., 2023) between 1992-2022 and Antarctica $7.6 \pm 3.9$ mm between 1992-2017 (Shepherd et al., 2018). The non-

linearity of mass loss from Antarctica is driven by West Antarctica, where glacier acceleration and retreat has caused an increasing contribution from 1992 onwards (Rignot et al., 2002, 2014; Mouginot et al., 2014; Scheuchl et al., 2016; Milillo et al., 2022; Christie et al., 2023). The stability of the West Antarctic Ice Sheet (WAIS) is a major concern (Joughin and Alley, 2011), as even a single glacier, Thwaites Glacier, has the potential to contribute 65 cm alone to sea level rise (Scambos et al., 2017). This exemplifies the need for accurate observation of ice sheet mass loss. Due to the self gravitation effect, local sea level rise along the world's coastlines strongly depends on the spatial distribution of ice sheet mass loss (Larour et al., 2017). This is where the advantage of altimeters comes into play. Altimeter data can provide ice sheet mass loss in a high spatial resolution compared to gravimeters. The accuracy of the altimeter based mass loss products is defined by the accuracy with which the individual elevation measurement is conducted, by the repeat cycle, the spatial interpolation scheme, and the conversion from volume change to mass change including firn densification. With this study, we focus on the improvement on the accuracy of the individual elevation measurement.

Ku-band satellite altimeters have been surveying ice sheets since the early 1990's in a substantial coverage, beginning with ESA's ERS-1 and ERS-2, followed by the ENVISAT 2002-2012. With the launch of CryoSat-2 in 2010 the first altimeter dedicated to study the Earth's cryosphere went operational. CryoSat-2 orbits at an unusually high inclination, reaching latitudes of 88°North and South. In addition it came along with major improvements to measure icy surfaces. Next to the conventional Low Resolution Mode (LRM), a Synthetic Aperture Mode (SAR) mode was accomplished to increase the spatial resolution in along track direction. To be able to locate the point of closest approach in sloped terrain in across track direction a second antenna is mounted enabling to measure the phase difference in the so called SAR interferometer (SARIn) mode. This allows to estimate the angle of arrival and thus to relocate the ground return. However, in this study we will focus on CryoSat-2 data in LRM mode. The first space-borne radar altimeter in Ka-band is SARAL/Altika launched in 2013. The latest radar altimeter is the pair of Sentinel-3 (S3A/S3B) that have been operating since 2016/2018 and both have a Ku-band altimeter onboard that operate in SAR mode. Next to the radar altimeters, two laser altimeters surveyed polar areas: NASA's ICESat-1 operated from 2003-09 (Zwally et al., 2014) and since 2018 ICESat-2 is operational (Markus et al., 2017; Smith et al., 2020). The great advantage of laser altimetry is the high precision of a single distance measurement and its low penetration into dry snow (Smith et al., 2020; Studinger et al., 2023). In addition, due to its small footprint size and high pointing precision the spatial assignment of the observations is high, even in areas of large slope and complex terrain (Smith et al., 2020). The disadvantage of laser altimetry is cloud cover, which prevents surface measurements and leads to data gaps. Distorted elevation measurements due to snow drift might also affect accuracy. As we use data from all six available laser beams and a three-year measurement period from January 2019 to December 2021, the data coverage is exceptionally good. The advantages of high precision, dense sampling, small footprint size and low penetration depth outweigh the disadvantages of occasional data loss due to cloud cover. Therefore, we use ICESat-2-based estimates of rates of elevation change as a reference to compare our radar altimetry-based results. As dry ice and snow are transparent for radar waves, the penetration of the radar signal complicates the detection of the true surface. In general the returned power over ice sheets consists of surface and volume components: surface scattering at the air-snow interface, scattering from internal layers, and volume scattering from scattering at snow grains. While the onset of surface scattering leads to a sharp rise in power (often called leading edge), volume scattering affects the gentle

decline in power over time (the trailing slope (TSL)). In general the waveform shape of a return echo mainly depends on the local surface topography in the area of the radar footprint and the small scale surface roughness, meaning the roughness on scales of the radar wavelength. This leads to distinct differences between waveforms from rough terrains and waveforms from smooth surface topography. On top of this the volume scattering acts as kind of low pass filter, leading to a widening of the waveform while adding more energy to the waveform tail but also enlarging the leading edge width (LEW), especially for conventional LRM waveforms. Volume scattering is caused by scattering of the radar wave that penetrates into the ice sheet, at snow and firn grains. This depends on the size of the grains, while the absorption loss of the radar wave is mainly governed by temperature. As a result the volume scattering varies widely over ice sheets due to differences in the snow and firn properties. As a consequence, the penetration of the radar signal in Ku-band can lead to a bias in elevation detection in the order of 10-20 cm (Larue et al., 2021) or even more, depending on the retracking method used to measure the range between the antenna mounted on the satellite and the surface. It has been shown by e.g. Davis (1997); Helm et al. (2014); Nilsson et al. (2016) that the choice of the retracker which aims to retrack the range at the lower part of the leading edge, like the Threshold Centre Of Gravity (TCOG) or Threshold First Maxima Retracker Algorithm (TFMRA), can strongly suppress the radar penetration bias. However, a remaining signal contribution is still left, which partly can be corrected using additional waveform parameters such as the LEW or TSL and/or the radar backscatter as proposed by e.g. Flament and Rémy (2012); Simonsen and Sørensen (2017); Schröder et al. (2019). In this study we aim to reduce this penetration bias directly in the waveform retracking by employing a machine learning approach.

Machine learning and in particular Deep Learning (DL) offers a data-driven alternative to traditional physical/statistical approaches for modeling the functional relationship between measurements and target variables. It is particularly successful in cases where the true relationship between input and output is too complex to be approximated by traditional models, which often only capture a part of the full spectrum or make simplifying assumptions to be tractable. Convolutional neural Networks (ConvNets), first proposed for more general computer vision applications such as image classification (LeCun et al., 2015), are a class of DL architectures that have shown tremendous success in a broad variety of Earth observation applications including land cover/use classification (e.g. from multi-spectral time series (Campos-Taberner et al., 2020)), image processing (e.g. speckle reduction of synthetic aperture radar images (Dalsasso et al., 2022)), and estimation of geo-/bio-physical parameters (e.g. forest height estimation (Lang et al., 2022)). In recent years Machine learning has been applied to various kinds of image data in polar areas. E.g. Loebel et al. (2022, 2023) monitored calving front motion at sub-seasonal resolution for 23 Greenlandic outlet glaciers using a U-Net (Ronneberger et al., 2015) on multi-spectral Landsat-8 imagery data. Baumhoer et al. (2019) extracted automatically Antarctic Glacier and Ice Shelf Fronts from Sentinel-1 Imagery by using a U-Net to create a dense time series of the Antarctic coastline to assess calving front change. Mohajerani et al. (2021) used a fully-convolutional neural network to automatically delineate glacier grounding lines in differential interferometric synthetic-aperture radar data. They applied their approach to more than 20000 interferograms along the Getz Ice Shelf in West Antarctica and demonstrate that grounding zones are one order of magnitude wider than expected. Beside satellite imagery airborne or ground based radar images of Ice-Penetrating Radar systems have been extensively studied and new insights could be achieved through application of machine learning approaches in recent years. Liu-Schiaffini et al. (2022) propose a deep learning model based

on convolutional neural networks and continuous conditional random fields (CCRFs) to automate ice bed identification. They deployed their approach to high-capability radar sounder (HiCARS) radargrams and were able to capture the global ice bed geometry as well as identifying fine-grained basal details even in areas with complex and rough ice bed conditions. Kamangir et al. (2018) presented a deep hybrid wavelet network for detecting ice surface and bottom boundaries, compared it with other edge detection approaches by using the NASA Operation IceBridge Mission data set. Dong et al. (2022) designed a Neural Network Fusion, called EisNet to extract next to the bedrock also internal layers from radiostratigraphic data. EisNet composes of three coupled deep neural networks which are based on U-Net architecture. Other applications of machine learning approaches deal with the segmentation of different structures in radargrams. E.g. García et al. (2021) developed an automatic analysis technique based on W-Net (Xia and Kulis, 2017), a fully convolutional auto-encoder to distinguish floating ice over ice shelves from grounded ice in coastal areas in radargrams recorded with the Multichannel Coherent Radar Depth Sounder MCoRDS2. Another segmentation scheme to segment radargrams into en-glacial layers, bedrock, basal units, and noise-limited regions such as the echo-free zone (EFZ) is based on a U-Net with attention gates and the Atrous Spatial Pyramid Pooling (ASPP) module is proposed by Donini et al. (2022). Their focus is the identification and mapping of basal layer and basal units and the network was successfully applied to two datasets acquired in North Greenland and West Antarctica using the MCoRDS3 data set. A very similar approach was developed by Cai et al. (2020) using bilateral filtering to reduce noise and a deep residual learning (He et al., 2016) as well as the ASPP module to classify free space, internal layers, bedrock, and noise (including EFZ region) and applied it to MCoRDS and MCoRDS2 radar images acquired between 2009 and 2011 in Antarctica. Finally, Ghosh and Bovolo (2022) constructed the TransSounder, a hybrid TransUNet-TransFuse architectural framework, to systematically characterize the different subsurface targets and compared it to other state of the art frameworks, by using a MCoRDS radar depth sounder dataset. All the above mentioned ML approaches are using images or two dimensional data sets as input and thus differ from the classical one dimensional echoes or waveforms detected by satellite altimetry. However, Machine learning has been applied in various other studies for waveform analysis: Müller et al. (2017) analyses altimetry data in the Arctic to detect open water within sea ice cover using an unsupervised clustering (i.e. k-medoids) of the radar echos to subdivide the waveforms based on different characteristics and subsequently classifies them via k-Nearest-Neighbors. Lee et al. (2016) use Random Forests to detect cracks between ice floes to improve the estimation of sea ice thickness. Random Forests are also used by Shen et al. (2017b, a) to classify sea ice type based on waveform data. These studies focus on the classification of the waveforms to detect different surface types. However, the regression task to accurately estimate surface elevation has been barely addressed. Fayad et al. (2021) used DL for the detection of surface heights from space-borne laser altimeter data of the GEDI mission (Dubayah et al., 2020). Fayad et al. (2021) used two ConvNets, a one dimensional for the individual waveform and reshaping it into two-dimensional representation to constrain biophysical parameters, such as canopy height and wood volume. Their results confirm, that ConvNets can be used to extract useful information from LiDAR waveforms and compare well with classical but complex and expensive random forest methodologies. Furthermore, Fayad et al. (2021) find that the 1D representation of the waveform produced slightly less accurate results than its 2D counterpart, both, for single and multi parameter output (estimation of canopy height and wood volume at the same time). They argue, that the reason for this being a larger gradient around an information peak, such as a vegetation or ground return, is generally larger in

the 2D representation of the waveform. As the data set contains peaks and the aim is to detect those peaks, the filters of the 2D-ConvNet model are better adapted to recognize signal content which are concentrated in small areas with high signal contrast,

(Fayad et al., 2021). However, over ice sheets we only deal with one prominent return waveform, which is an integrated signal originating from a large footprint with a diameter of roughly 15 km including contributions of the upper snow/firn layer up to a depth of less than 10 m. Therefore, signal gradients are not as large and single or multi peak waveform are usually only occurring in very complex terrain. Furthermore, the noise level of a single radar waveform is much higher than for a LiDAR waveform, which results in noise peaks on top of the gentle signal. In addition, the radar waveforms consist of only 128

samples in contrast to 1444 samples of the GEDI waveform. This would lead to small image sizes of 12x12 pixels as a 2D representation of a single waveform, including some padding, and would allow less convolutional layers than the 1D approach, if pooling layers or strided convolutional layers are inserted to reduce the number of parameters to learn. Furthermore, our application developed for satellite radar altimetry, is also very contrasting to typical application of 2D DL approaches such as layer or feature detection or classification within images (radargrams) recorded by radar depth sounders. Those systems can

penetrate up to 4 km of ice and thus are capable to provide detailed information of internal structures, bed rock as well as basal features within the recorded radargrams. Here, 2D ConvNets are used to capture spatially correlated signals in the along-track direction. Since the receive range window of a satellite radar altimeter is adjusted to follow the terrain by the onboard tracker, consecutive waveforms are not necessarily aligned and may jump within the radar range window, especially when the satellite samples changing undulating surfaces such as ice sheets. This can lead to erroneous results when using a 2D ConvNet that

captures spatially correlated signals. However, over the open ocean or in coastal altimetry applications, a 2D approach could be promising. Since neither peak detection, image classification nor spatial correlated layer detection is the objective of our approach, we decided that a 1D representation of the ConvNet is sufficient to accomplish our task of accurately retracking the beginning of the leading edge of a single waveform. In the following, we use single waveforms of CryoSat-2 and represent them as sequential data to a 1D ConvNet that applies a series of processing layers (in particular convolutions with learned kernels

along the time dimension of the waveform) to automatically extract features and agglomerate information. The output of the network is the retracked range that corresponds to the snow/firn surface. In order to engage supervised machine learning for processing of the satellite radar altimeter waveforms, a large data set with known range is needed. In contrast to (Fayad et al., 2021) who trained their models on a subset of GEDI waveforms, where ground truth measurements existed, a ground truth based learning approach cannot be achieved here. The area covered by airborne or ground-borne soundings of the ice surface

using laser scanners or GNSS traverses are orders of magnitudes smaller than those of satellite measurements. Space borne laser altimetry as ICESat-2 to be used as test data set in a DL approach to improve radar derived elevation measurements is in our opinion also not applicable. The reasons for this are the very different footprints of the two systems. While the ICESat-2 laser points to areas of less than $0.02\,km^2$, satellite radar altimeters illuminate large areas of up to $10\,km^2$ , so that the two are not spatially assigned and cannot be directly compared with each other. Even more, the large scale topographic undulation

and surface slope influence the waveform shape but also involve a slope correction in the post processing to reposition the radar elevation measurement to its point of closest approach. As this correction cannot be extracted from the waveform shape itself a direct comparison between laser or GNSS derived surface elevation and radar derived elevation as a ground-truth for

a DL approach is not possible. Instead, we make use of simulated waveforms to create a large synthetic reference data set used for training and testing of the new ConvNet retracker. To represent the satellite altimeter waveforms as best as possible, we take the local surface topography over the ice sheet into account. This way, we can create basically an infinite number of training samples to learn a neural network. After the training phase we apply the new ConvNet retracker to measured CryoSat-2 waveforms and derive elevation and elevation change estimates, which we compare to ICESat-2 derived elevation change data products. The remainder of this paper is structured as follows: In Section 2 we first present our approach to simulate waveforms, followed by presenting the used ConvNet, provide a brief overview of other retrackers and summarize the used satellite altimeter data and our approach to estimate rates of elevation change. Section 3 shows the performance of the new AWI-ICENet1 on simulated waveforms before we present results on real satellite altimeter data over selected sites and ice sheet wide. In Section 4 we first discuss the results achieved with AWI-ICENet1 on simulated data before we compare it to other retrackers. Finally, the estimates of the rates of elevation change are evaluated and compared with those of ICESat-2.

## 2  Material and Methods

### 2.1  Simulated waveforms

For the success of our approach to develop a DL retracker capable of minimizing the effects of variations in backscatter and radar speckle on range measurements, a good reference dataset is essential. We have defined the following criteria for our reference data set that must be met by the simulated waveforms:

- represent observed CryoSat-2 LRM waveforms as best as possible

- large variability of waveforms shapes

- large number of waveforms for optimal training results

To meet the criteria and to ensure a good training result, several features must be included in the simulation:

- representation of different real topographic situations

- allow different bulk attenuation rates (to mimic volume scattering component)

- add Gaussian noise to each waveform (to mimic radar speckle)

- add noise floor prior the leading edge

- repositioning of waveform within range window to cope tracker gate variations

- include CryoSat-2 system characteristics (Ku-Band frequency (wavelength), antenna gain pattern, range resolution)

We randomly selected 1000 locations within the LRM zone limits of Antarctica as shown in Fig. 1. At each location a reference surface echo $P_{r_s}$ is simulated, using the classical radar equation integrated over the illuminated surface area (Eq. 1, Brown (1977)):

$$P_{r_s}(t) = \frac{P_t \lambda^2}{(4\pi)^3} \int_A \frac{\sigma_S^0(\theta) G(\theta)}{R^4} \, dA \tag{1}$$

with wavelength $\lambda = c/f_c$, speed of light $c$, Ku-Band centre frequency $f_c = 13.5$ GHz, the transmit power $P_t$, the backscatter cross-section $\sigma_S^0(\theta)$ and $R$ the range from the radar to the surface element dA. For simplicity $\sigma_S^0(\theta) = 10 \, dB$ is chosen to be homogeneous and without any angular dependency within the radar footprint and the antenna gain pattern $G(\theta)$ is defined as an elliptical 2D Gaussian function:

$$G_{\theta_{al}\theta_{ac}} = G_0 \exp\left(-\left[\frac{\theta_{al}^2}{\beta_{al}^2} + \frac{\theta_{ac}^2}{\beta_{ac}^2}\right]\right) \tag{2}$$

with the look angles $\theta_{ac} = \tan(R_x(x,y)/R_z(x,y))$ and $\theta_{al} = \tan(R_y(x,y)/R_z(x,y))$ and the constants $\beta_{al} = \theta_{al_{3db}}/\sqrt{4\log(2)}$ and $\beta_{ac} = \theta_{ac_{3db}}/\sqrt{4\log(2)}$. Centered at each location $(x,y)$, $R_z(x,y) = h_{\text{sat}} - h_s(x,y)$ is estimated by using a mean satellite altitude of $h_{\text{sat}} = 730$ km and $h_s(x,y)$ is derived from an high resolution (20 m pixel resolution) interpolated sub grid of the input digital elevation model (DEM). As input DEM we used a slightly smoothed (kernel size of 3 km) version of the REMA DEM (Howat et al., 2019) with pixel resolution of 1 km to mirror the effect of an integrated signal within the pulse limited footprint. Each 2D-range pattern $R(x,y) = \sqrt{R_x(x,y)^2 + R_y(x,y)^2 + R_z(x,y)^2}$ covers the radar footprint and has dimensions of 30 km $\times$ 30 km re-sampled to 20 m pixel resolution. For the elliptical Gaussian 2D-antenna pattern we use an antenna beam width of $\theta_{al_{3db}} = 1.3°$ and $\theta_{ac_{3db}} = 1.15°$ as given by Wingham et al. (2006). To mimic tracker gate variations and to force the network not to retrack always at the same incremental position we fractional shifted $P_{r_s}$ randomly within the range gate while updating the reference range epoch.

In the next step we added different volume contributions following the approach of Legrésy and Rémy (1997) by using an exponential decay function (Eq. 3)

$$V(z) = \exp(-r(z)L_A/4.342945) \tag{3}$$

and a homogeneous layered model consisting of $N = 128$ layers with a depth resolution of $\Delta R_z = 0.468$ m, which correspond to the range bin size and the number of range bins per observed CryoSat-2 LRM waveform. For simplicity the loss factor $(L_A)$ accounts for loss contributions due to volume scattering, absorption and stratification of the snow pack in a combined manner. This loss factor, called bulk attenuation rate in the following, is adjustable to be able to vary the volume contribution. The final received power $P_{r_v}$, including surface and volume contributions, is given by (Eq. 4)

$$P_{r_v} = \sum_{i=rt_{ref}}^{N-1-rt_{ref}} P_{r_s} V(z_{i-rt_{ref}}). \tag{4}$$

Although the model architecture is simple, it represents the general effect of absorption and scattering leading to an attenuation of the radar wave within snow/firn due to different physical properties like temperature, surface density grain size and layering as explained by Lacroix et al. (2008); Adodo et al. (2018).

In a simplified model performance test, in which a simulated waveform of a flat surface was fitted to observed CryoSat-2 LRM waveforms only by adjusting $L_A$ in a minimum least square estimator (MLE), we found very high correlations between measured and fitted waveforms of 0.9 and higher. The test results are shown in Fig. 2 and Fig. A1 for Antarctica and Greenland, where the median of the fitted attenuation rate and the median of the correlation between fitted and observed model are shown on a grid of $5\,\text{km} \times 5\,\text{km}$ pixel resolution. Based on the findings in Fig. 2(a) we generated 95 reference waveforms at each location using an attenuation rate between 1 db and 20 dB with a step size of 0.2 dB. Finally, we generated 40 different noisy samples of each of the reference waveforms by adding 40 different randomly generated samples of Gaussian noise. We thereby increased the training data set to 3.8 Mio noisy waveforms ($m = 1000 \times 95 \times 40$).

$$\Psi_i = P_{r_v} * \epsilon_s + \epsilon_f \tag{5}$$

while $\epsilon_s$ represents radar speckle and $\epsilon_f$ a Gaussian noise floor. Both were selected to match the noise of CryoSat-2 waveforms. We denote the final noisy waveforms hereafter referred to $(\Psi_i)_{i \in I}$, where $I = \{1, \ldots, m\}$ is the index set of the waveforms. Each waveform consists of 128 range bins, matching the data structure of the CryoSat-2 satellite altimetry waveform product. For each of the noisy waveforms the actual true surface or reference range is known and is used as reference for the ConvNet training. A typical example of the different simulation steps is shown in Figure 3, where the observed CryoSat-2 waveform at the same geographical position (Lake Vostok) is overlaid in blue. $L_A$ was estimated by a MLE to be 1.5 dB/m. More example waveforms simulated in different topographic environments are shown in the Appendix in Figures A2, A3, A4 .

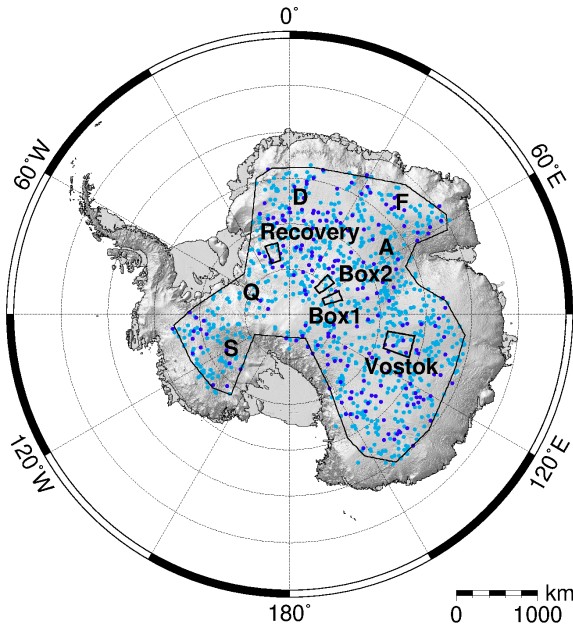

**Figure 1.** Overview of the four selected regions used in the analysis. The entirety of the points show the randomly selected locations where waveforms are simulated. Light blue dots show the 80% used for training and dark blue the 20% used for testing the ConvNet.

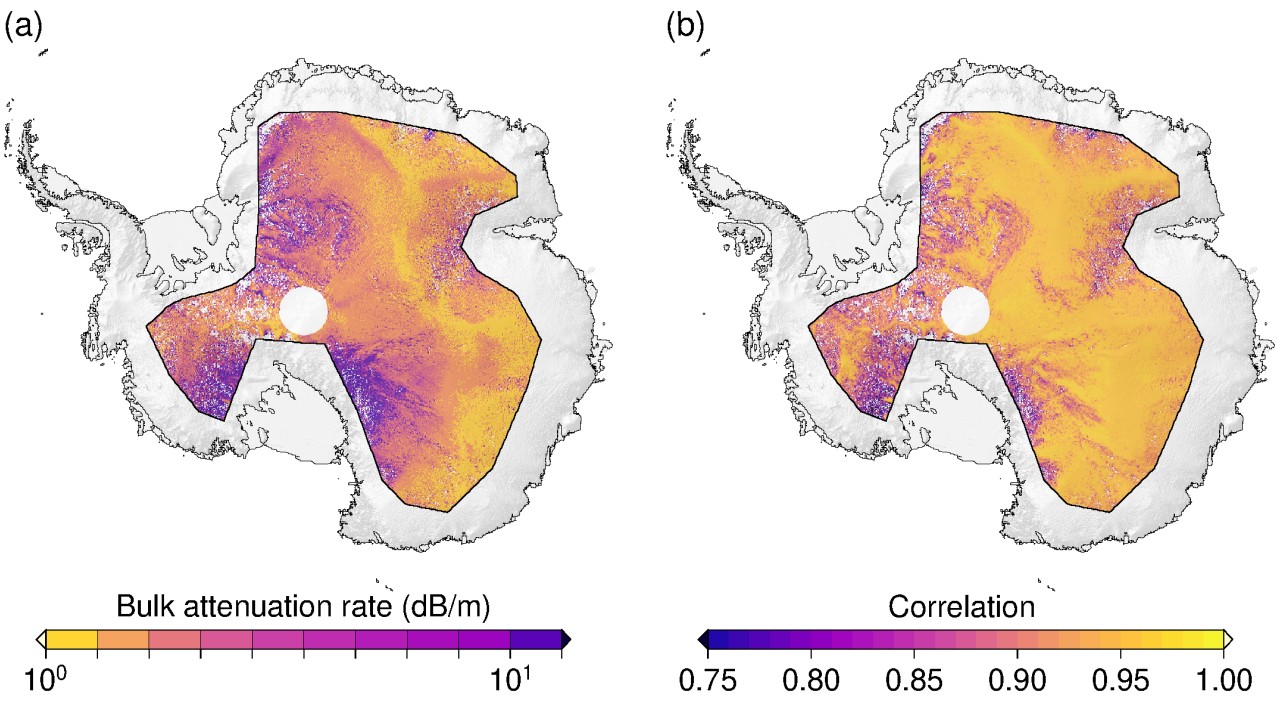

**Figure 2.** Results of the test where a flat surface waveform model is fitted to real CryoSat-2 waveforms by adjusting the attenuation $L_A$ as the only parameter. (a) Gridded median of the attenuation rate estimated by a MLE fit and (b) median of the correlation between observed and fitted waveforms.

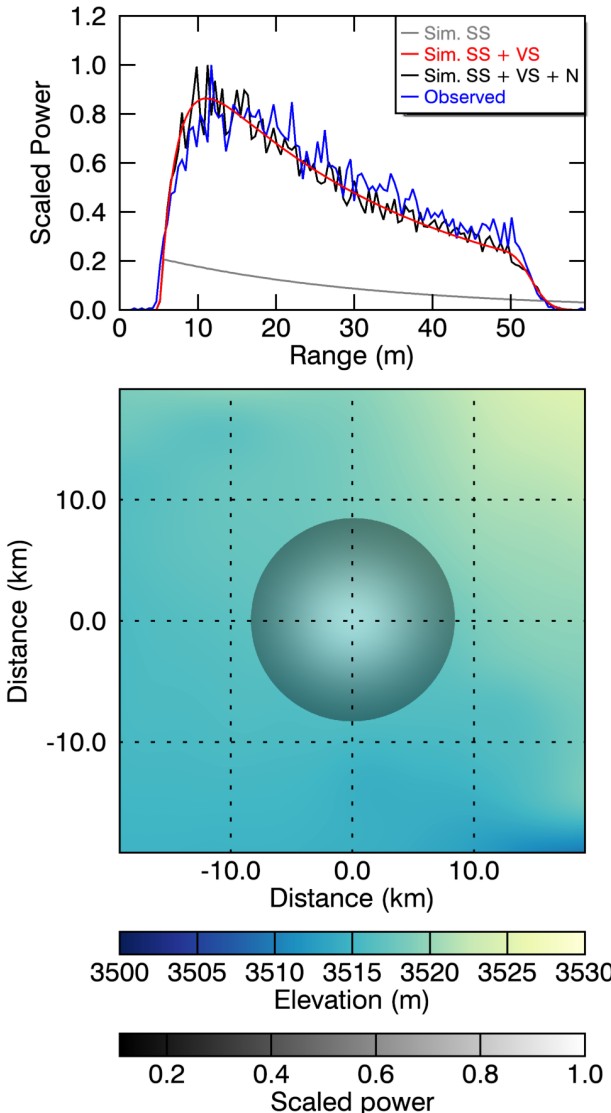

**Figure 3.** Example of the different simulation steps shown for a typical waveform over Lake Vostok. Blue line denotes the observed CryoSat-2 LRM waveform and the gray line the simulated surface waveform (SS) based on Eq. 1. The red line represents the simulated waveform including volume scattering (SS+VS) according to Eq. 3, while the black line depicts the final simulated waveform including surface and volume scattering as well as noise (SS+VS+N) as given in Eq.5. The lower panel represents the 2D elevation model and scaled $P_{r_s}$, which is mainly controlled by the Gaussian antenna pattern.

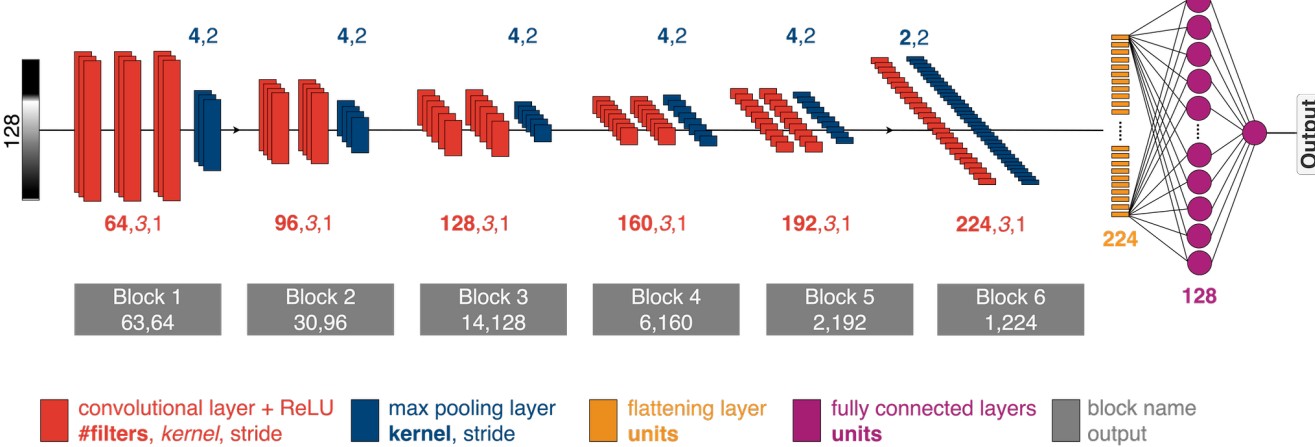

**Figure 4.** A ConvNet is used to map an individual waveform to the corresponding range that correlates with the actual surface. We use standard combinations of convolutions, ReLU, batch-norm, max-pooling, and drop-out layers. The final feature layer is flattened and used as input to a fully-connected layer.

## 2.2 AWI-ICENet1

We use a ConvNet $f$ to model the relationship between a (real or simulated) waveform $\Psi$ and the corresponding retracked range $r$, i.e. the network provides an estimate $\hat{r} = f(\Psi)$ of the retracked range, which corresponds to the true surface, based on the provided input data. During training, only simulated waveforms are used. The network is then applied to real waveforms during inference.

The used ConvNet, which is similar to the 1D ConvNet of Fayad et al. (2021), consists of six blocks of stacked convolutional layers with ReLU activation functions, batch-normalization, maximum pooling and drop-out (with overlapping pooling windows similar to Krizhevsky et al. (2012)). This is in contrast to Fayad et al. (2021) who use strided convolutional layers instead of pooling layers to reduce the number of parameters to learn. In our network design the batch normalization is applied after the activation, whereas in the network of Fayad et al. (2021) it is the other way around. The final feature layer is flattened and processed by a fully connected layer with 128 units. Since the waveforms are 1D input data, all layers and operations are one-dimensional as well. Fig. 4 shows an overview of the network architecture.

The cost function $\mathcal{L}$ is a standard mean squared error (MSE) loss between the true and estimated range of the simulated waveform, i.e.

$$MSE = \frac{1}{|\mathcal{B}|} \sum_{(r,\Psi) \in \mathcal{B}} (r - f(\Psi))^2, \tag{6}$$

where $\mathcal{B} \subset \mathcal{D}_{Tr}$ is the batch, i.e. a subset of the training set $\mathcal{D}_{Tr}$.

The network is implemented in TensorFlow and trained for 25 epochs with a batch size of 128 via an Adam optimiser (Kingma and Ba, 2014) with L2 regularization.

We split the available training data into five different folds and perform five independent runs with training data $\mathcal{D}_{Tr}$ consisting of four folds, while the fifth fold serves as test data $\mathcal{D}_{Te}$. Performance is averaged over these five runs. Model performance is assessed via MSE

$$MSE = \frac{1}{|\mathcal{D}_{Te}|} \sum_{(r,\Psi)\in\mathcal{D}_{Te}} (r - f(\Psi))^2,$$ (7)

the mean absolute error (MAE)

$$MAE = \frac{1}{|\mathcal{D}_{Te}|} \sum_{(r,\Psi)\in\mathcal{D}_{Te}} |r - f(\Psi)|,$$ (8)

and the root mean squared error (RMSE)

$$RMSE = \sqrt{\frac{1}{|\mathcal{D}_{Te}|} \sum_{(r,\Psi)\in\mathcal{D}_{Te}} |r - f(\Psi)|^2}.$$ (9)

## 2.3 Satellite altimetry data

In this study we use CryoSat-2 Level 1B (including measured waveforms) and Level 2I (including range estimates by ESA retracker) products of the Baseline E low resolution mode (LRM) provided by the European Space Agency (ESA). We make use of the ICE1 and ICE2 retracker solutions given in the Level 2I product and apply according to Helm et al. (2014) the TFMRA and the newly developed AWI-ICENet1 retracker to the Level 1B waveform product. ICE1 is based on a threshold centre of gravity retracker (TCOG) (Wingham et al., 1986; Davis, 1997), while ICE2 the UCL land-ice retracker fits a Brown model (Brown, 1977) adapted for CryoSat-2. To analyse the performance of the AWI-ICENet1 in comparison to the other three retracker, we make use of the ATL06.006 ICESat2 data product provided by NASA (Smith et al., 2023). We use data from all six beams within the time span from January 2019 to December 2021. Instead of using the quality flag given in the ATL06 product we filter the data based on the version 2 of the REMA Antarctic elevation model in 1 km pixel resolution (Howat et al., 2022) and version 4.1 of the ArcticDEM mosaic in 500 m pixel resolution (Porter et al., 2023). All data points with a difference larger than ± 100 m are excluded from further processing.

## 2.4 Elevation change and empirical corrections for time-variable radar-penetration effects

Surface elevation change (SEC) processing as applied by various groups use different strategies to minimize the effect of radar penetration. In most cases the backscatter and/or additional waveform shape parameters such as the leading edge width (LEW) and the trailing edge slope (TES), estimated by the ICE2 retracker introduced by Legresy et al. (2005); Frappart et al. (2016), are used. Those waveform parameters are not provided by other retracker such as the offset centre of gravity (OCOG by Wingham et al. (1986)), a threshold centre of gravity (TCOG by Davis (1997)) or the threshold first maxima retracker algorithm (TFMRA by Helm et al. (2014)). Here, we make use of the LEW and backscatter provided by the ESA LRM-Level 2I ICE2 and ICE1, respectively. The decision to use the backscatter of ICE1 for all retrackers is due to the lower sensitivity of the OCOG amplitude to speckle noise, so that the backscatter of successive waveforms is less noisy, but large-scale and time-dependent

fluctuations are preserved. After retracking, the geo-referenced surface elevation is determined for each of the retracking approaches using orbital information such as altitude, latitude and longitude and additional geophysical correction included in the ESA products. In addition, the refined slope correction (Roemer et al., 2007) is applied to relocate the echo to its point of closest approach. This results in a large point cloud of geo-referenced elevation measurements for each of the retracker. Li et al. (2022) developed the LEPTA method, an improved version of the relocation slope correction which includes points in the underlying DEM that contribute to the rise of the leading edge. Their results show an improved cross point error (CPE) between CryoSat-2 and ICESat-2 compared to the method of (Roemer et al., 2007). However, as we only consider intra-mission cross point errors and apply the same slope correction to all retracker solutions the slope correction method does not play any role in our CPE analysis.

The interpolated elevation anomaly product and rates of elevation change (dhdt) are generated using a slightly different approach as described in e.g. McMillan et al. (2016); Schröder et al. (2019); Nilsson et al. (2022). For each pixel with a size of 1 km x 1 km we collect all geo-referenced data points within a variable distance ranging from 500 m to 2500 m (step width 500 m) and correct for topography using a bi-linear interpolation of the REMA-DEM and/or ArcticDEM, respectively, rather than fitting any kind of linear or quadratic surface as McMillan et al. (2016); Schröder et al. (2019); Nilsson et al. (2022). The variable search radius is enlarged step wise until a threshold of number of points is reached. This threshold is defined to match at least 75 % of the selected time period ($n_{months}$) and the following criteria: For CryoSat-2: $n_{months} * 6$ and due to the higher data coverage of six beams and less along track point spacing for ICESat2: $n_{months} * 48$ . This kind of processing allows to minimize uncertainties due to unresolved topography within the search radius but keeping enough data points for the linear regression as the search radius is tried to keep as low as possible. Processing costs for pixels with very dense data coverage in the interior of Antarctica are kept low as only a small radius and thus less data points are selected. At the same time less unobserved pixels in areas of coarse data coverage remain as the search radius can be enlarged up to 2500 m. We then estimate rates of elevation change using a linear regression for each pixel with sufficient data coverage (Criteria: $max(time) - min(time) > 50\%$ of selected time period and $n_{points} > n_{months}$), again without using additional information such as LEW, TeS, backscatter or seasonal components. The residuals are averaged to monthly residuals per pixel. Both gridded products, the trend and the monthly residual grids, are finally interpolated using inverse distance weighting with variable radius to form the final interpolated grids with 5 km posting. Furthermore, the backscatter and LEW information are processed in the same way without any trend or topographic correction. We calculate different variants of corrections for transient penetration effects by using empirical linear relations between $\Delta h$ and LEW, $\Delta h$ and backscatter, or $\Delta h$ and LEW and backscatter. Instead of doing this in the context of multi-parameter fitting as in e.g. Flament and Rémy (2012); Simonsen and Sørensen (2017); Schröder et al. (2019), we do it on the level of spatially interpolated monthly anomalies of $\Delta h$, LEW and backscatter, following the approach of Nilsson et al. (2022). We assume, that changes of electromagnetic properties of the ice-sheet surface are driven by atmospheric processes that affect the temperature and surface density at the kilometer scale and can thus explain the time-varying elevation anomalies (hereafter referred to as $\Delta h$) as shown by Lacroix et al. (2008)). Our approach of applying the correction to averaged, interpolated products reduces the high uncertainty of single waveform parameter estimates. This is reflected in high correlations between $\Delta h$ and anomalies of LEW and backscatter as presented in figures A5, A6, A7, A8 for Greenland

and Antarctica. Our final product contains four monthly elevation estimates for each of the four retracker solutions used to

315 investigate $\Delta h$ and to derive dhdt. These are compared with those derived by ICESat-2 using the same processing strategy.

- – $\Delta h$ and dhdt without any correction

- – LEW corrected $\Delta h$ and dhdt

- – Backscatter corrected $\Delta h$ and dhdt

- – LEW and backscatter corrected $\Delta h$ and dhdt.

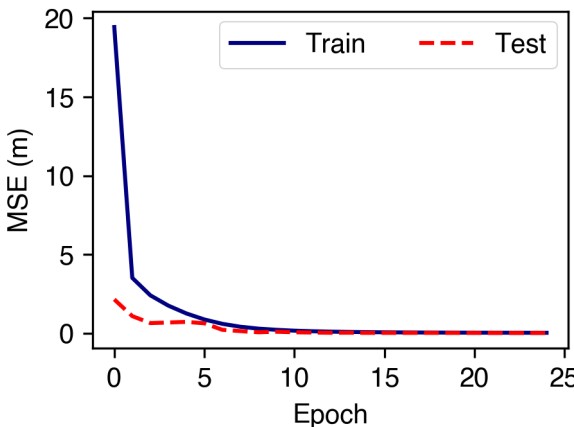

**Figure 5.** Training loss evolution of the AWI-ICE1Net ConvNet.

 **3 Results**

To evaluate the performance of the new AWI-ICENet1 retracker, different tests are conducted, which are summarized and structured as follows. First, the AWI-ICENet1 is evaluated using statistical metrics such as the MSE based on the simulated test data set. Learning curve and K-fold cross validation statistics are shown. Second, the simulated data set is retracked using the TFMRA retracker of Helm et al. (2014) and compared to AWI-ICENet1. Third, the new retracker is applied to CryoSat-2 LRM data and a monthly cross point analysis is carried out in four selected regions (ROI) of Antarctica, to assess the retracker's ability to provide reliable elevation estimates in areas with different surface topography and bulk attenuation rate. Fourth, the monthly cross point analysis is carried out across the LRM zone using all four retrackers (AWI-ICENet1, TFMRA and ESA products ICE1 and ICE2). To assess the retracker's performance in terms of its ability to minimize transient penetration effect, the monthly elevation anomaly product is evaluated within the ROIs and spatially over the whole LRM zone of Antarctica and Greenland. Finally, empirical correction to reduce time variant penetration bias are applied to all retrackers and evaluated.

## 3.1 AWI-ICENet1 results for simulated waveforms

We evaluated the ConvNet performance using MAE, MSE, and RMSE as given in Equations 7, 8 and 9, and by examining the learning curve of the MSE for both the training and test data sets. The learning curve of our final model is shown in Fig. 5. Here, the MSE of training and test data gradually decreases and reaches a constant plateau after approximately 10 epochs. To assess whether the model architecture provides similar metrics for different training and test data, we applied 5-fold cross validation, meaning that the same input data-set is split into five different parts, by using an 80/20 split factor for training/testing. The results of the different folds are listed in Table 1. They show nearly identical values indicating a consistent, repeatable learning

| K-fold | MSE (m) | RMSE (m) | MAE (m) |
|--------|---------|----------|---------|
| 1 | 0.070 | 0.056 | 0.042 |
| 2 | 0.075 | 0.060 | 0.047 |
| 3 | 0.070 | 0.056 | 0.047 |
| 4 | 0.070 | 0.056 | 0.047 |
| 5 | 0.094 | 0.066 | 0.047 |
| Mean | 0.076 | 0.059 | 0.046 |

**Table 1.** K-fold cross-validation results

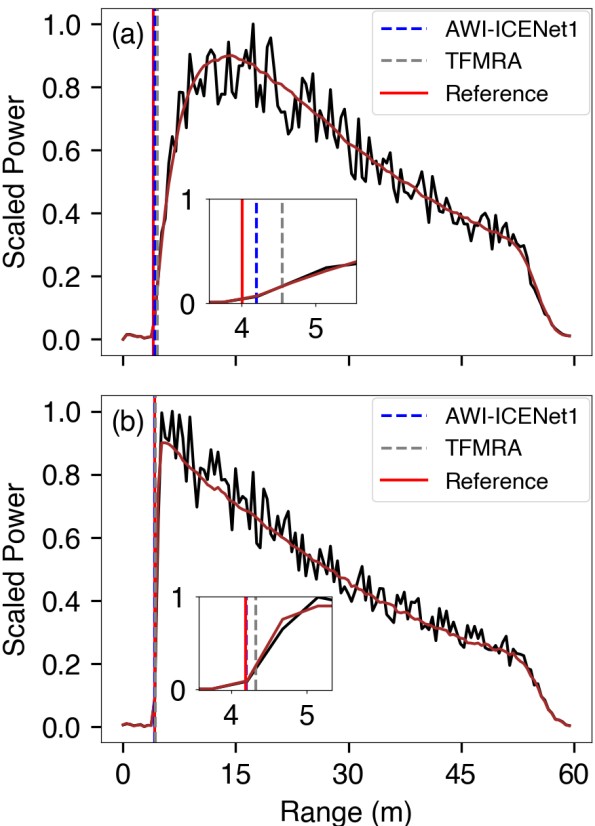

**Figure 6.** Examples of simulated waveforms for (a) low $L_A$ and (b) high $L_A$. Initial noise free waveform is shown as red line and with added noise as black line. The reference range is displayed as vertical bar in red color and superimposed is the AWI-ICENet1 and TFMRA retracked range in blue and grey, respectively. The inlets show a zoom into the leading edge were the retracking takes place.

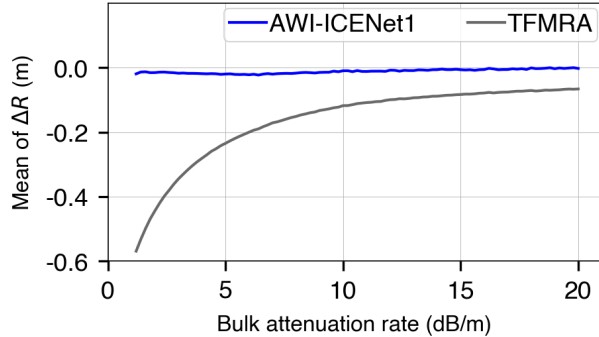

**Figure 7.** Comparison of the Mean of $\Delta R$ with respect to bulk attenuation rate between AWI-ICENet1 and TFMRA.

independent of the training data set. Our final model, which was a hold-out with RMSE of $0.056\,m$ and MAE of $0.042\,m$, was considered for application to real data and further analysis.

To evaluate the retracker performance we estimated the difference $\Delta R = R_{\mathrm{ref}} - R_{\mathrm{RT}}$ between the retracked range $R_{\mathrm{RT}}$ and the reference range $R_{\mathrm{ref}}$. In addition we applied the TFMRA retracker to the same set of simulated waveforms and estimated $\Delta R$ as well. In Fig. 7 the mean values of $\Delta R$ for the AWI-ICENet1 and TFMRA are presented. The statistics are calculated for all location across Antarctica subdivided into bins of the bulk attenuation rate. Here, the influence of high volume scatter, due to low $L_A$, on range retrieval is clearly shown. For low $L_A$ we observe only a small offset ($< 0.03\,\mathrm{m}$) of the AWI-ICENet1

retracked surface elevation compared to the true surface. In contrast, $\Delta R$ of TFMRA forms a kind of exponential function, with differences of up to $-0.5\,\mathrm{m}$ for low $L_A$ as presented in Fig. 7.

As performance test we run the retracking on one of the CPU and GPU compute nodes of the high performance cluster at the Alfred Wegener Institute, Helmholtz Centre for Polar and Marine Research. We applied next to the TFMRA retracker the TCOG retracker and an adapted version of the functional fit of the ICE2 retracker as given in Legresy et al. (2005). To

350 estimate the leading edge width (LEW) based on TFMRA and TCOG we run the retracking for different threshold levels (THL), reaching from 5% to 80%. For each THL a retracked position (RT) is determined. The LEW is the inverse of the linear regression coefficient and is estimated for each waveform as follows: $LEW = 1/m$ with $THL = m * RT + n$

Results of the performance test are shown in Table 2.

|  | AWI-ICENet1 | TFMRA / TFMRA and LEW | TCOG / TCOG and LEW | ICE2 |
|---|---|---|---|---|
| Processing time on CPU (s) | 233 | 31 / 207 | 13 / 134 | 153 |
| Processing time on GPU (s) | 56 | 30 / 204 | 13 / 130 | 149 |

**Table 2.** Results of the performance test of different retracker. Retracking was applied to 1 Million waveforms.

### 3.2 AWI-ICENet1 results for observed waveforms

In this section, the final AWI-ICENet1 retracker is applied to the entire CryoSat-2 time series. For each of the ESA Level1B 20 Hz waveforms a range is retracked and combined with the precise orbit information (latitude, longitude and altitude) to form a point cloud of geo-referenced surface elevations. The cross point analysis is carried out on precise orbits. To derive elevation and elevation change products used for further applications, a slope corrected point cloud data set is generated using the relocation slope correction following Roemer et al. (2007).

#### 3.2.1 Cross point error analysis

To avoid time dependent differences, due to changes in radar volume scattering, our accuracy measure of the individual solutions is based on monthly cross point error analysis (CPE) over the whole Antarctic ice sheet. CPE is defined as the elevation difference between ascending and descending track at cross points (CPs). In total more than 3 Mio CPs from 130 months are used. We filter outliers using the following criteria: $|CPE| > 10$ m. Finally a grid with 5 km × 5 km pixel resolution based on mean and standard deviation (SD) is calculated from all CPEs. Figures 8 and 9 show the time evolution of the median and SD of CPE for four selected regions which are shown in Fig. 1. In the area of Box1 (Fig. 8a) we find a significant negative CPE for TFMRA and both ESA retrackers, with the largest CPE for the ESA ICE2 retracker. AWI-ICENet1 has a very low CPE and performs best in this area. In Box2 the picture is similar, but the CPE is in the positive range with again ESA ICE2 showing the highest CPE (Fig. 8b). Also the SD, displayed in panels (a) and (b) of Figure 9, has exceptional high values for ESA's ICE2 retracker. The other retrackers have similar SD's in the order of 20 cm in Box1 and Box2. Next we consider two regions with specific topographic settings (both areas are shown in Fig. 1): the Vostok region is characterized as a very flat area with low surface undulations, whereas the Recovery region is more complex with higher surface slopes of up to 1° and medium scale topographic undulations. The results for median of CPE are presented in Fig. 8 panels (c) and (d). For both areas, the median of CPE is lower than for the two box areas, with a range of at most 20-30 cm. Again ESA ICE2 shows the highest CPE over the entire time period for both Vostok and Recovery areas. The new AWI-ICENet1, TFMRA and ESA ICE1 have median and SD CPE values at similar order of magnitude with only slight differences. The temporal variability is higher in the Recovery area than in the Vostok region.

The strong difference in surface characteristics become also evident when comparing the SD of CPE: In the Recovery area the range of the SD of CPE is about 0.5-1.5 m, roughly a factor of three compared to the Vostok area and the two box areas. In all cases the trend of median and SD of CPE are rather stable over the years with few exceptions. In the area Box2 we find a decrease in median of CPE for ESA ICE2 from 2018 on-wards, and some incremental changes, a few centimeters in size, are observed by ESA ICE1 and TFMRA, while AWI-ICENet1 remains constant over time. The time series of SD from CPE also show generally stable trends, again with few exceptions: in Box1 and Vostok we find a peak in all retrackers except ESA ICE2, but the trend found for the median in Box 2 is not reflected in the SD. The high temporal variability in the SD of CPE in the Recovery area prohibits analysing trends, but all retracker show a variability of about the same order of magnitude.

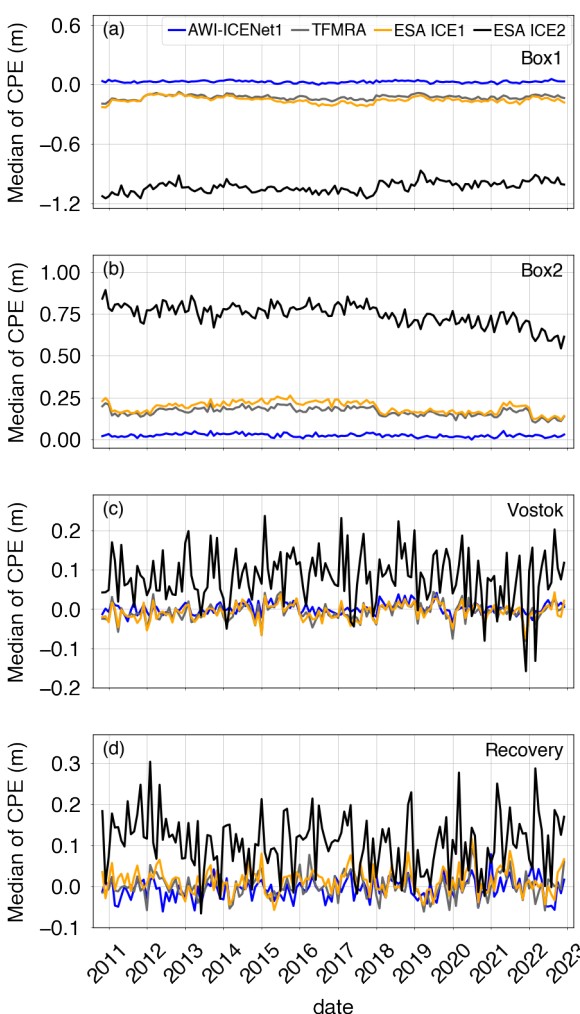

**Figure 8.** Time series of the median of cross-point errors for different areas. Cross-point-errors are determined in monthly resolution within the respective region.

In a next step, we broaden the view and discuss the CPE over the entire ice sheet. Figure 10 displays the time series of SD of CPE for the complete LRM zone of Antarctica using the 12 years observational time period of the CryoSat-2 era. Here, AWI-ICENet1 and ESA ICE1 are similar in SD of CPE, whereas TFMRA is considerable higher in SD over the entire ice sheet. ESA ICE2's has particular high SD, almost twice as large as the first two. The temporal variability is rather low and 390 without any trends for all four retracker solutions. The pan-Antarctic gridded median and SD of CPE are shown in Figures 11 and 12. The results reveal, that the crossover difference is reduced by roughly a factor 4 to 5 for ICE1 and TFMRA compared to ICE2 (note the different value range in panel (c) of Fig. 11). However, remaining CPE's with alternating signs in the order of

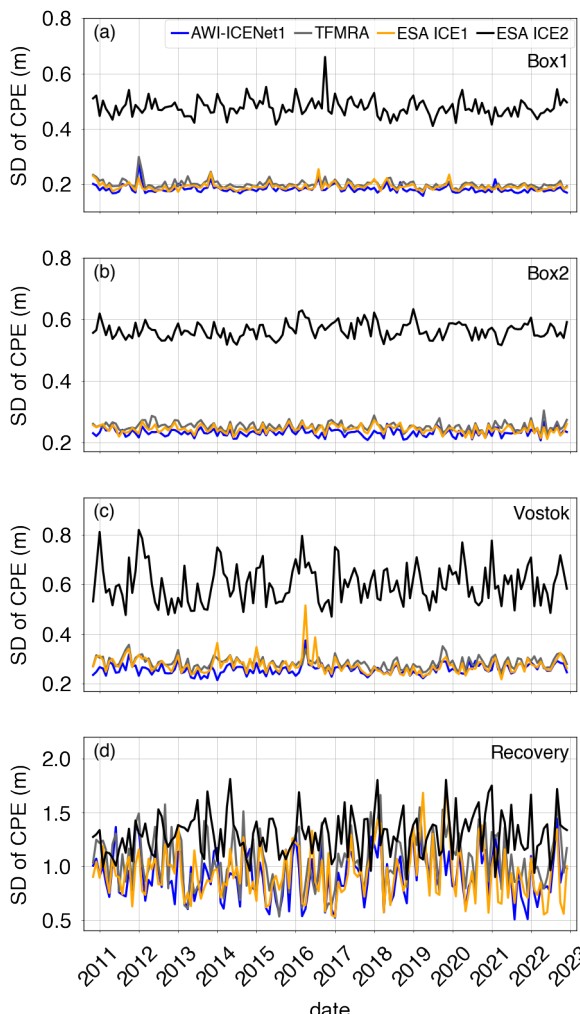

**Figure 9.** Time series of the standard deviation of cross-point errors for different areas. Cross-point-errors are determined in monthly resolution within the respective region.

up to $\pm 0.3$ m are visible all over Antarctica including a prominent pattern in the interior of East Antarctica, close to the polar gap. This pattern, discussed below, is entirely eliminated with AWI-ICENet1, as displayed in panel (a) of Fig. 11. Low median

of CPE with AWI-ICENet1 can be also seen close to the ice divides north of Dome Fuji (indicated with F in Fig. 1), in southern Dronning Maud Land (D) as well as in topographically more complex areas of the Siple Coast (S) and drainage area of Amery ice shelf (A). The SD of CPE in Fig. 12 (a), (b) and (d) is very similar for AWI-ICENet1, ESA ICE1 and TFMRA, respectively and a factor of four to five smaller than for ESA ICE2 (Fig. 12c). In general lowest values are found in the flat interior and

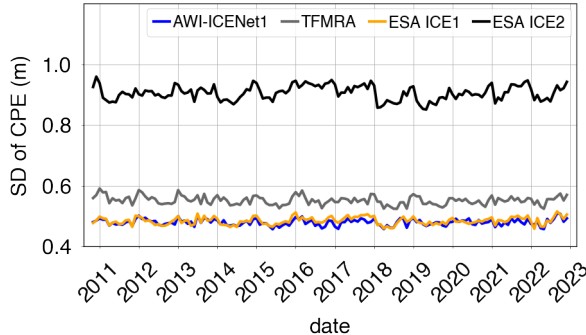

**Figure 10.** Time series of the standard deviation of cross-point-errors over the LRM zone in Antarctica. Cross-point-errors are determined in monthly resolution.

highest values in the sloped areas with complex topography. This is consistent with our findings for the four test areas, where
highest SD of CPE are observed in the Recovery area of roughest topography.

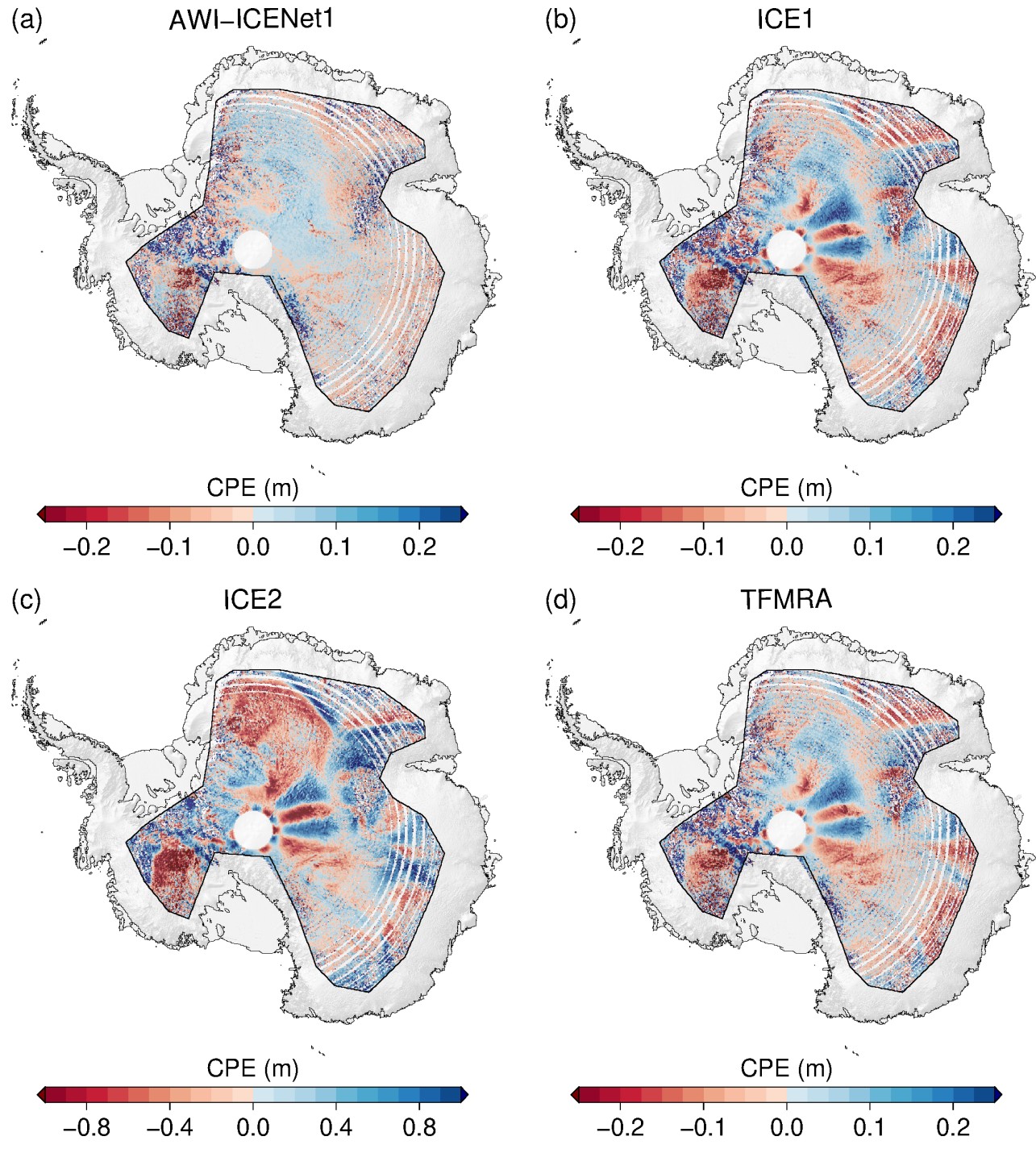

**Figure 11.** Spatial distribution of the median of cross-point-errors over the LRM zone in Antarctica for all four retrackers. Cross-point-errors are determined in monthly resolution within the time period from January 2011 to December 2022. The median is determined for each pixel of size 5 km x 5 km. Please note, panel c has a different range of the color bar then the other three panels.

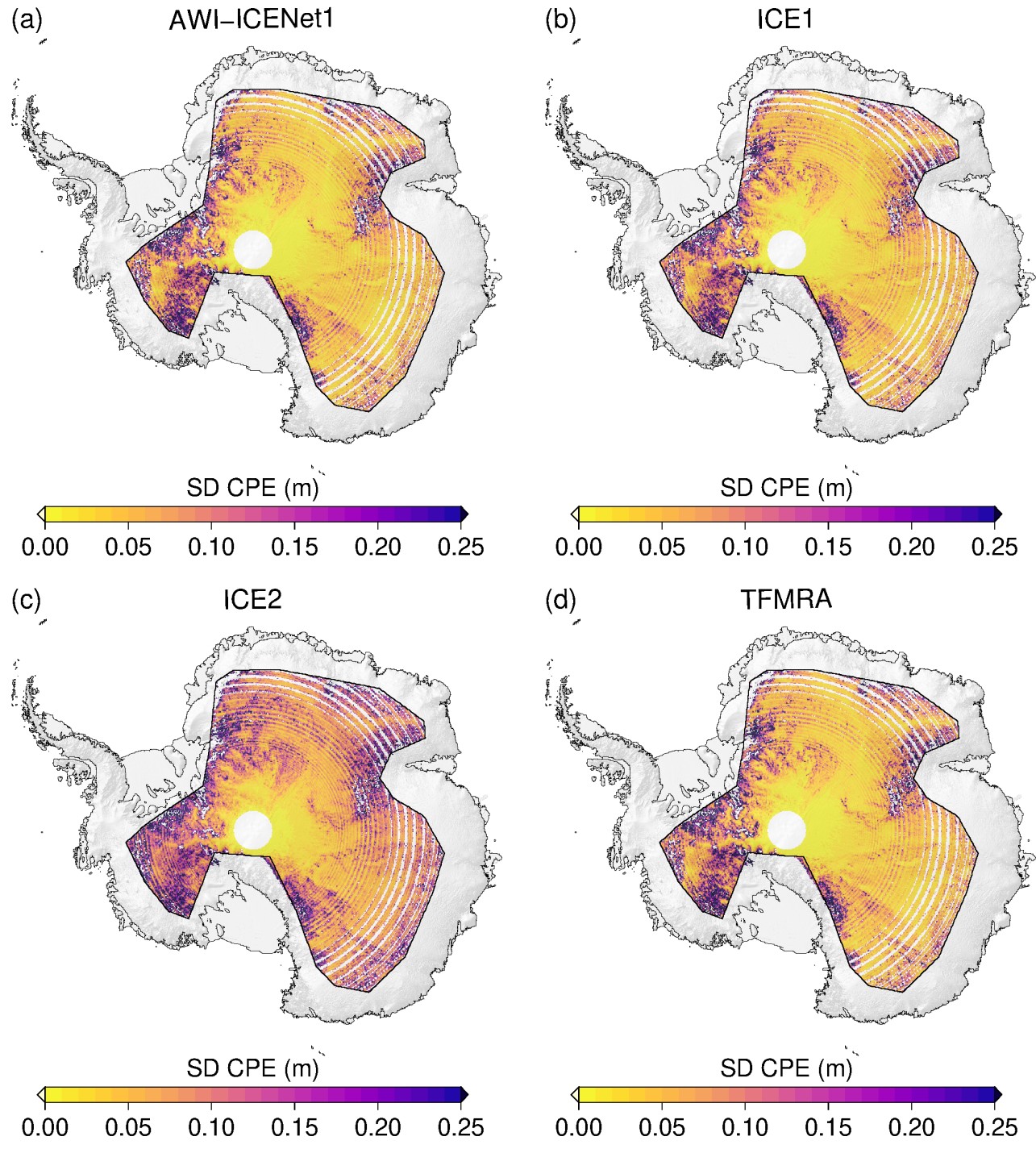

**Figure 12.** Spatial distribution of the standard deviation of cross-point-errors over the LRM zone in Antarctica for all four retrackers. Cross-point-errors are determined in monthly resolution within the time period from January 2011 to December 2022. The standard deviation is determined for each pixel of size 5 km x 5 km.

### 3.2.2 Transient penetration

In this section we analyse the time dependent variability of $\Delta h$, which has been widely discussed in the literature as transient penetration or penetration bias due to changes in firn properties e.g. (Davis and Zwally, 1993; Michel et al., 2014; Slater et al., 2019) for various retracker. To this end, we present in Fig. 13 the $\Delta h$ and its standard deviation ($SD(\Delta h)$) for all retrackers

and over the entire CryoSat-2 time period for the Lake Vostok region, a region with a stable surface height over the last decade (Richter et al., 2014). Except AWI-ICENet1 all retrackers are showing a strong variability with ESA ICE2 having the largest range of up to 0.8 m. The temporal variability of ESA ICE1 and 2 and TFMRA are correlated, but ESA ICE1 and TFMRA are lower in magnitude compared to ESA ICE2. The new AWI-ICENet1 exhibits an $\Delta h$ of only few centimeters and with minor temporal variability, with a slight increase from 2021 on-wards. The SD of the elevation anomaly (Fig. 13b) is largest

for ESA ICE2, while ESA ICE1 and TFMRA have a nearly similar SD, significantly lower than ESA ICE2 but the temporal evolution has the same form as for ESA ICE2. AWI-ICENet1 exhibits not only a low $\Delta h$ but also a low $SD(\Delta h)$. Fig. 14 presents the $\Delta h$ and $SD(\Delta h)$ for an $5000\,km^2$ large high elevated area on the North Greenland plateau ($79°$ N, $45°$ W). Large sudden increase of more than 1.0 m for ESA ICE2 are observed in July 2012 and August 2018. Here TFMRA and ESA ICE1 also experience an increase of 0.3 m whereas AWI-ICENet-1 stays at the same level. Both events can be related to unusual heat

waves transporting warm air up to high elevations resulting in surface melt conditions as reported by the DMI portal. In all other months the anomalies are correlated and follow a similar course but with strong differences in the observed magnitudes with AWI-ICENet1 showing smallest and ESA ICE2 highest values. For the SD of the elevation anomaly (Fig. 14b) we observe the same as for the Vostok area with lowest values for AWI-ICENet1 and highest for ESA ICE2. Below we will discuss how this relates to findings of other studies, but now continue with inspecting the spatial distribution of the $\Delta h$.

Figure 15 presents the spatial distribution of the $SD(\Delta h)$. Here the standard deviation is calculated for each single pixel (5 km posting). ESA ICE1 and TFMRA have a close spatial distribution and magnitude with the largest elevation anomaly along Siple Coast (S in Fig. 1) and the drainage basins in the Queen Elisabeth Land feeding the Ronne Ice Shelf (Q in Fig. 1). For ESA ICE2, the SD of elevation anomaly exceeds 30 cm in an extensive area. The lowest SD of elevation anomaly is found for AWI-ICENet1, where the largest values are also found in West Antarctica, but $SD(\Delta h)$ for the large area of nearly whole

East Antarctica is below 5 cm, whereas for ESA ICE1 and TFMRA $SD(\Delta h)$ is in the range of 8–25 cm.

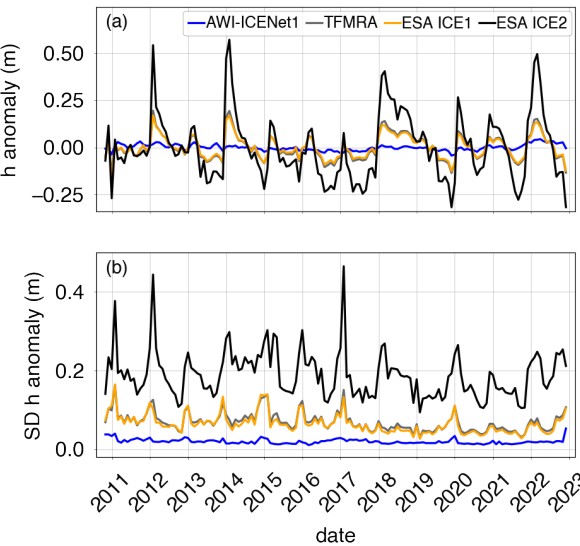

**Figure 13.** Time-dependent mean value (a) and standard deviation (b) of the elevation anomaly in the Vostok region. The elevation anomalies are based on grid with 5 km x 5 km pixel resolution of the spatially interpolated monthly residuals of the elevation trend estimation.

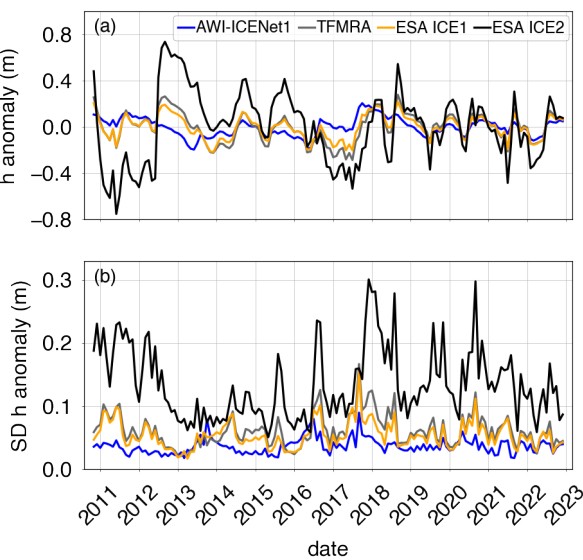

**Figure 14.** Time-dependent mean value (a) and standard deviation (b) of the elevation anomaly in the North-Greenland region. The elevation anomalies are based on grids with 5 km x 5 km pixel resolution of the spatially interpolated monthly residuals of the elevation trend estimation.

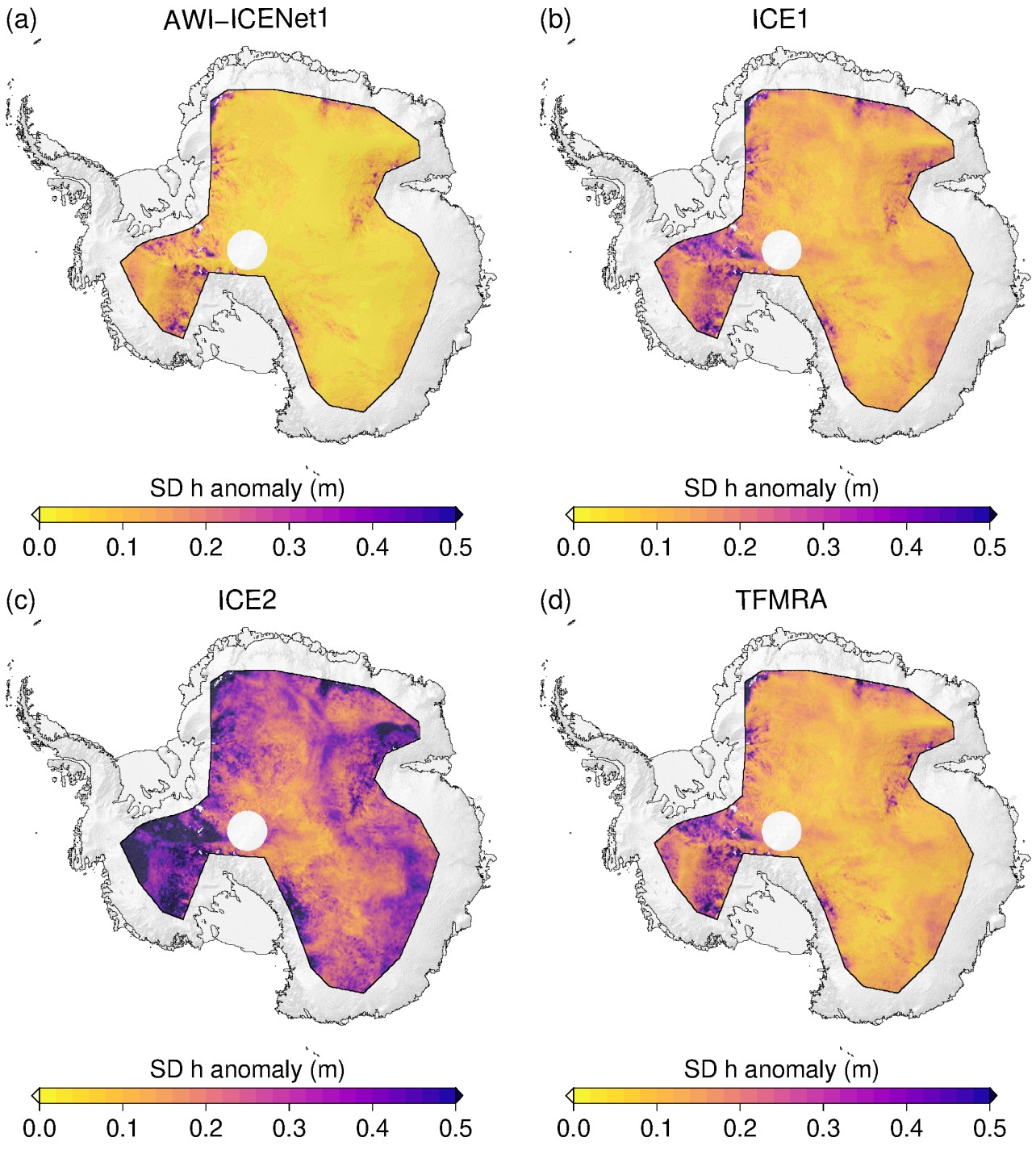

**Figure 15.** Spatial distribution of the standard deviation of the elevation anomaly over the LRM zone in Antarctica for all four retrackers. The elevation anomalies are based on grids with 5 km x 5 km pixel resolution of the spatially interpolated monthly residuals of the elevation trend estimation. The standard deviation is determined for each pixel covering the full time period from January 2011 to December 2022.

### 3.2.3 Empirical correction of transient penetration compared to AWI-ICENet1

Flament and Rémy (2012); Simonsen and Sørensen (2017); Schröder et al. (2019); Nilsson et al. (2022) used an empirical relation of the elevation anomaly with backscatter, LEW and/or TeS to partly reduce the transient penetration effect. Here, we make use of the same correction largely following Nilsson et al. (2022). Exemplary we apply the correction using LEW and backscatter to all retracker for the Vostok area and LEW only for the North Greenland area, which is line with findings of Simonsen and Sørensen (2017). Results of Vostok and North Greenland are shown in Figures 16 and 17, respectively. In both cases $\Delta h$ is strongly reduced for all retracker with exception of AWI-ICENet1. Largest reduction is observed for ESA ICE2 but still showing largest remaining $\Delta h$ which exceed the values for AWI-ICENet1 by a factor of two and higher. TFMRA and ESA ICE1 are similar and closer to AWI-ICENet1 but still show larger magnitudes. The SD of the $\Delta h$ is reduced for all retracker with lowest improvement for AWI-ICENet1. However, our new retracker shows the smallest SD as presented in Figure 16 b). In North Greenland the sudden positive elevation increase, due to a a change of the dominant scattering regime caused by the melt events in summer 2012 and 2018, are strongly suppressed. The different solutions are now correlated over the years. Smallest remaining $\Delta h$ is observed with AWI-ICENet1, followed by TFRMA, ESA ICE1 and ESA ICE2. For North Greenland, TFMRA and ESA ICE1 show larger negative and positive deviations from AWI-ICENet1 in winter 2013/2014 and 2017/2018, respectively, as shown in Figure 17 a). $\Delta h$ in North Greenland is more variable compared to Vostok reaching values of 0.1 m for AWI-ICENet-1. Around Vostok only small perturbations of <3 cm are observed with AWI-ICENet-1, reflecting the low accumulation regime in East Antarctica. Figure 18 presents the spatial distribution of the SD of elevation anomaly corrected with LEW and backscatter. Again, ESA ICE2 show largest anomalies followed by ESA ICE1 and TFMRA. $SD(\Delta h)$ is reduced to 4–10 cm for TFMRA and ESA ICE1 for the majority of the East Antarctic plateau. For AWI-ICENet1 only a minor reduction of $SD(\Delta h)$ could be achieved with values to be less than 4 cm for the majority of East Antarctica. It is worth mentioning that even the corrected TFMRA in Fig. 18 (d) still show larger values than the uncorrected AWI-ICENet1 in Fig. 15 (a). To investigate the extent to which the applied correction mitigates the transient penetration for each retracker, we sort the $SD(\Delta h)$ of the uncorrected and corrected $\Delta h$ within the LRM zone in bins of 0.02 m as shown in Fig. 19. The cumulative frequency in Fig. 19(a) depicts that approx. 80% of $SD(\Delta h)$ for the AWI-ICENet1 is less than 6 cm, while this is true for less than 20% of TFMRA and ESA ICE1. ESA ICE2 reveals values greater than 15 cm for more than 90% and values greater than 30 cm for more than 50% of the area. After the correction, as presented in 19(b), 60% of TFMRA and of ESA ICE1 reach values less than 6 cm. While the effect of the applied correction is significant for all retrackers except AWI-ICENet1, it is not able to minimize the transient penetration bias to the same order of magnitude as the AWI-ICENet1. Since AWI-ICENet1 experiences only a small reduction of the $SD(\Delta h)$, we conclude that most of the transient penetration bias is already corrected by the retracker. Similar results are obtained for the Greenlandic ice sheet but with in general larger magnitudes of $SD(\Delta h)$. Corresponding Figures of the spatial distribution of $SD(\Delta h)$ and the cumulative distribution are presented in the Appendix in Figures A11 A12 and A13.

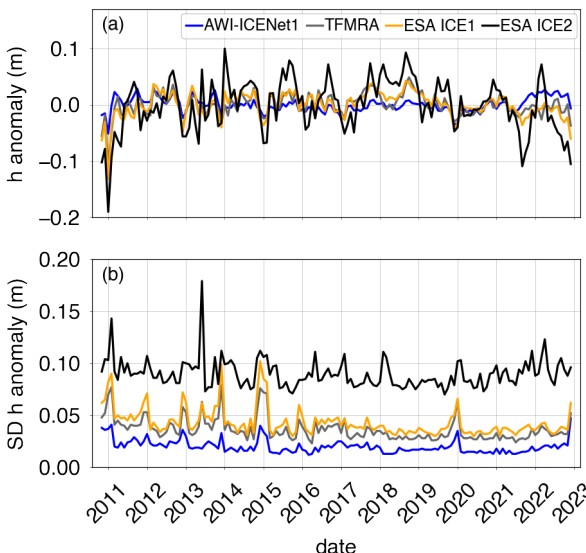

**Figure 16.** Time-dependent mean value (a) and standard deviation (b) of the corrected elevation anomaly in the Vostok region. The elevation anomalies are based on grids with 5 km x 5 km pixel resolution of the spatially interpolated monthly residuals of the elevation trend estimation corrected for transient penetration using LEW and backscatter.

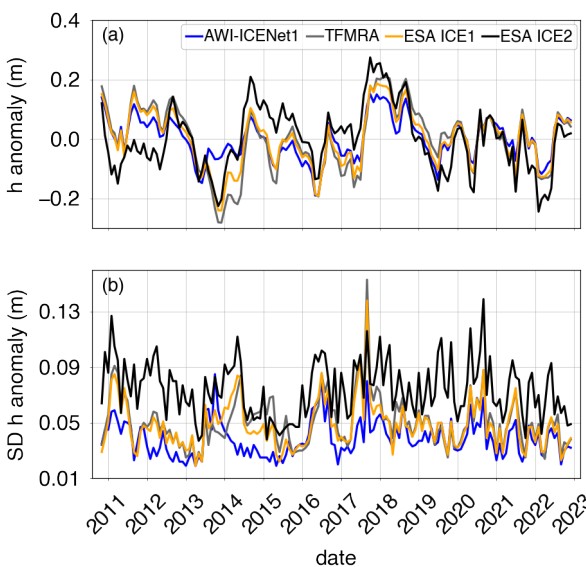

**Figure 17.** Time-dependent mean value (a) and standard deviation (b) of the corrected elevation anomaly of a region in North-Greenland. The elevation anomalies are based on grids with 5 km x 5 km pixel resolution of the spatially interpolated monthly residuals of the elevation trend estimation corrected for transient penetration using LEW only.

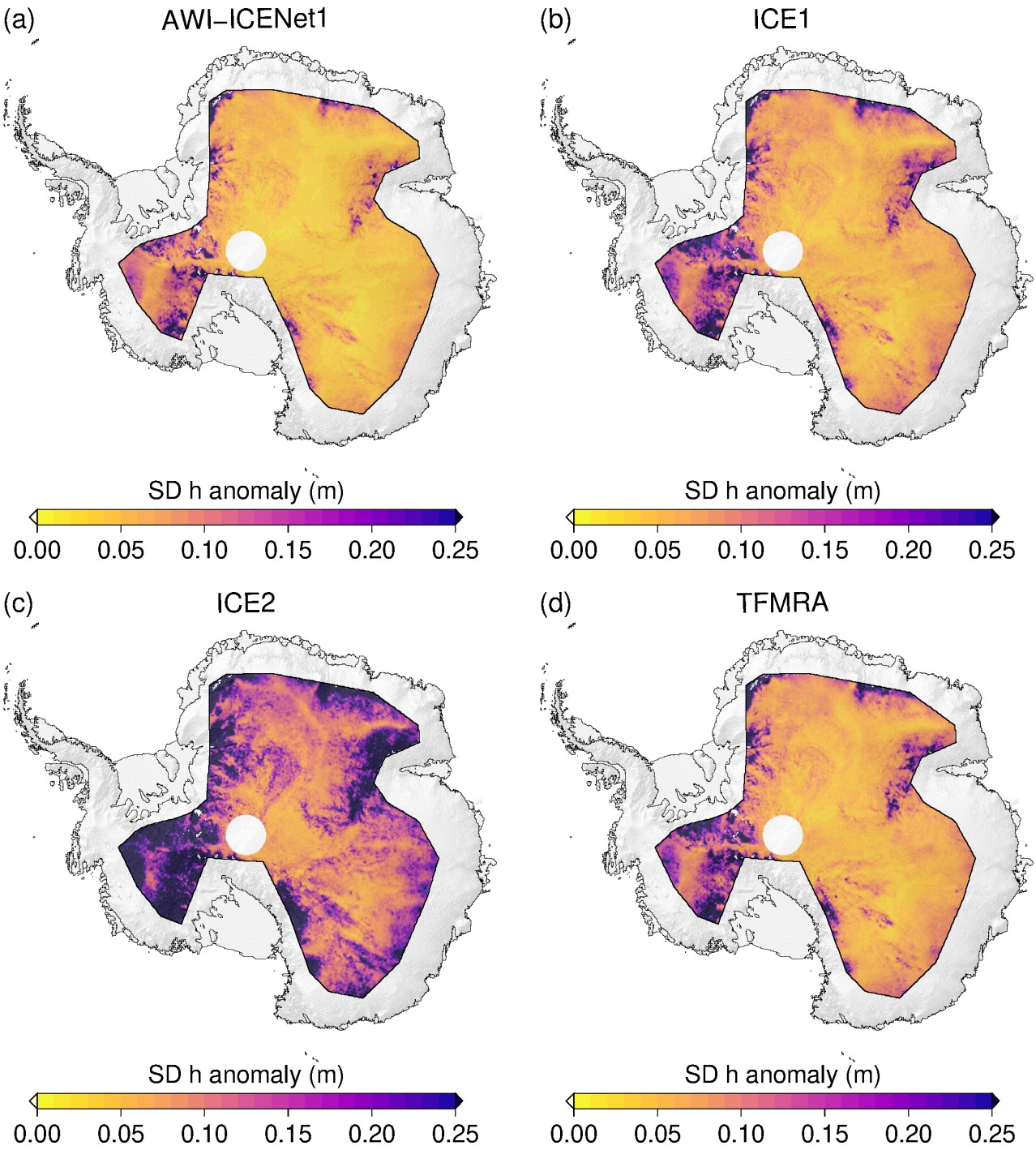

**Figure 18.** Spatial distribution of the standard deviation of the corrected elevation anomaly over the LRM zone in Antarctica for all four retrackers. The elevation anomalies are based on grids with 5 km x 5 km pixel resolution of the spatially interpolated monthly residuals of the elevation trend estimation corrected using a correlation with leading edge width and backscatter. The standard deviation is determined for each pixel covering the full time period from January 2011 to December 2022.

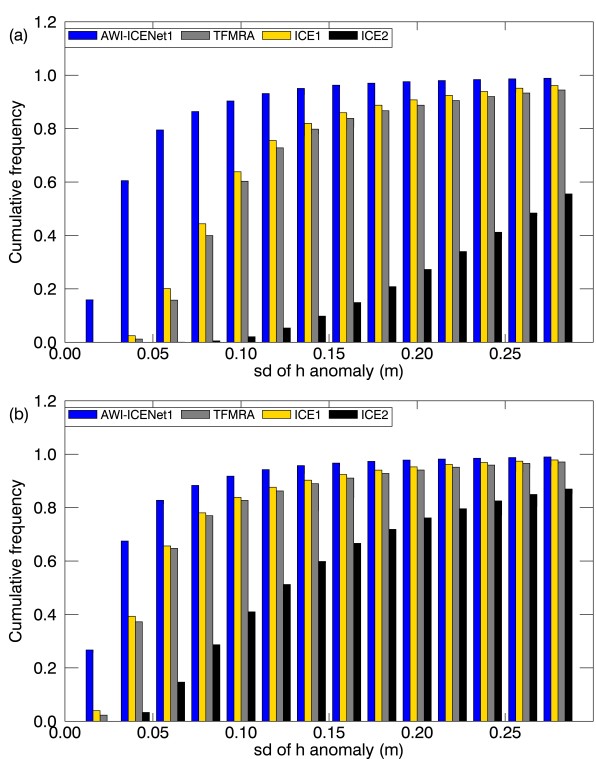

**Figure 19.** Cumulative histogram with bin size of 0.02 m of the standard deviation of the elevation anomaly over the LRM zone in Antarctica for all four retrackers. (a) uncorrected and (b) corrected h anomalies using a correlation with LEW and backscatter. The elevation anomalies are based on grids with 5 km x 5 km pixel resolution of the spatially interpolated monthly residuals of the elevation trend estimation. The standard deviation is determined for each pixel covering the full time period from January 2011 to December 2022.

## 4 Discussion

### 4.1 Assessment of AWI-ICENet1 for simulated waveforms

Several waveform models have been developed to study the effects of physical parameters on waveform shape and their effect on retracked surface height, such as Ridley and Partington (1988); Femenias et al. (1993); Legrésy and Rémy (1997); Adams and Brown (1998); Arthern et al. (2001). Martin et al. (1983) showed that the shape of the waveform echo are affected by large scale surface undulations. The surface slope has a large effect on the width of the leading edge (Femenias et al., 1993). Femenias et al. (1993) also showed that the error on range retrieval due to volume backscatter can be up to $50\,cm$ and mainly affects

the trailing edge of the echo. Small scale roughness can account for $\pm 2\,cm$ of the height bias. Our waveform model captures large scale undulations and surface slope because we use a subset of the REMA elevation model as input for each individual simulated waveform. We do not consider macro-roughness (meter scale sastrugi's) and small scale roughness (centimeter scale ripples) as these are not the main source of error in the height estimates. The subsurface signal can be attenuated by various physical processes and parameters, e.g. absorption losses, scattering from ice grains, multiple reflections within the snow pack

caused by the stratification of the subsurface due to density contrasts (Ridley and Partington, 1988; Femenias et al., 1993; Legrésy and Rémy, 1997). In our model, we do not resolve the various parameters, but consider their combined attenuation effect as an overall or bulk attenuation rate that changes the waveform. In section 3.1, simulated sample waveforms for high and low $L_A$ were shown in Figure 6. In addition to the very different TeS, the broadening of the LEW for low $L_A$ is striking and reflects the observations of Femenias et al. (1993). This subsequently leads to incorrect estimates of the TFMRA-retracked

surface elevation, as the retracked range is estimated to be 25% of the leading edge (Helm et al., 2014). As a result, the retracked range is estimated within the first meter of the snow pack, depending on the width of the leading edge, rather than the true surface which is shown in Fig. 7. In contrast AWI-ICENet1 strongly suppresses this penetration effect and positions the retracked range much closer to the true surface and is therefore less sensitive to errors caused by changes in the volume scattering contribution. Because AWI-ICENet1 has only been trained over the LRM zone, it may not be optimally trained for

the very complex regions near the ice sheet margins. This could lead to larger errors when applied to PLRM data included in the ESA Baseline E data product. If AWI-ICENet1 is to be used for other missions such as EnviSat, SARAL/Altika, or Sentinel-3 (PLRM), new simulation and training would be required to best represent mission-specific sensor characteristics such as centre frequency, bandwidth, or antenna gain pattern.

### 4.2 Assessment of AWI-ICENet1 for satellite altimetry data

#### 4.2.1 Cross point error analysis

In section 3.2.1, CPE results were presented as pan-Antarctic gridded median and SD of CPE, Figures 11 and 12. A prominent feature in the central part (around the pole gap) in East Antarctica was identified as a static crossover pattern. This pattern was explained by Armitage et al. (2014) as the result of an isotropic dependence of the extinction coefficient on the angle between the radar polarization and wind-induced properties of the firn. As a consequence of the Armitage et al. (2014) results, Box1

and Box2 were selected to investigate whether the crossover pattern is static over time and can be reduced with AWI-ICENet1. We have shown, that the results in this area reveal a reduction in CPE of about a factor 4 for ICE1 and TFMRA compared to ICE2 (note the different range of values in panel (c) of Fig. 11). A similar observation has already been made by Helm et al. (2014), where TFMRA showed slightly better results than ESA ICE1. This can be confirmed here as well. Moreover, the pattern is completely eliminated with AWI-ICENet1, as displayed in panel (a) of Fig. 11. We conclude that the influence of anisotropic dependence of the extinction coefficient on wind-driven directional anisotropy of the ice sheet surface and firn on surface height measurements as described by Legresy et al. (1999) is strongly suppressed by the new AWI-ICENet1 retracker. These results are consistent with those of Arthern et al. (2001), who found that the extinction coefficient, or as we refer to it, the bulk attenuation rate, decorrelates between ascending and descending tracks. Because AWI-ICENet1 is able to minimize the effect of bulk attenuation rate on range measurements it is also able to remove the directional effect from ice sheet height measurements.

### 4.2.2 Transient penetration

Time dependent elevation anomalies as measured by radar satellite altimetry is a composite signal of ice dynamical processes, changes in the surface mass balance (SMB), changes in the firn compaction rate and time dependent radar penetration. The latter one can exceed SMB and firn compaction anomalies by one order of magnitude in areas of low accumulation rate such as the majority of the vast East Antarctic ice sheet. To reliable estimate volume and mass change it is very important to correct for the penetration bias as the true elevation change is only driven by changes in SMB, firn compaction and ice dynamics. In Section 3.2.2 the time dependent variability of the elevation anomaly over the Lake Vostok area was presented. All retrackers usually used over ice sheets show undulations of the surface height of a couple of decimeters in an area which is known to be stable over the last decades (Richter et al., 2014). Similar findings for this area were already discussed in the literature for CryoSat-2 as well as for Envisat and ways to partly suppress the penetration bias presented. As this undulation is driven by time dependent changes in the firn properties the widely accepted method is to use the correlation of the $\Delta h$ to backscatter and/or waveform parameters LEW and TeS as they are also effected by firn properties and thus also change in time.

We applied this correction to all retrackers in 3.2.2 and show how effective this correction method works. However, it is still unclear how much penetration bias explains the measured anomaly and how effectively the applied corrections can mitigate the bias. To address this issue we compare our results with ICESat-2 observations and focus on the time period from January 2019 to December 2021 when both mission were in operation. Although ICESat-2 is a green laser altimeter, a negligible penetration bias in dry snow can be assumed (Studinger et al., 2023). This is supported and illustrated in Fig. 20, which shows the $\Delta h$ observed by ICESat-2 in the given time period. In approximately 90% of the area the SD of $\Delta h$ is less than 0.04 m, indicating that for most of the area there is only little variation in penetration, SMB and/or firn compaction. Only in the Ammundsen Sea Embayment (ASE) and at the Siple coast in West Antarctica are h anomalies of more than 0.1 m found. The latter could be a dynamic thickening signal of the stagnant Camp Ice Stream (Nield et al., 2016). The higher variability in the ASE can be explained by the extreme precipitation events in this area reported by Davison et al. (2023) during the winters of 2019 and 2020. In Fig. 21 we compare the cumulative frequency of the $SD(\Delta h)$ with ICESat-2 for the uncorrected and corrected

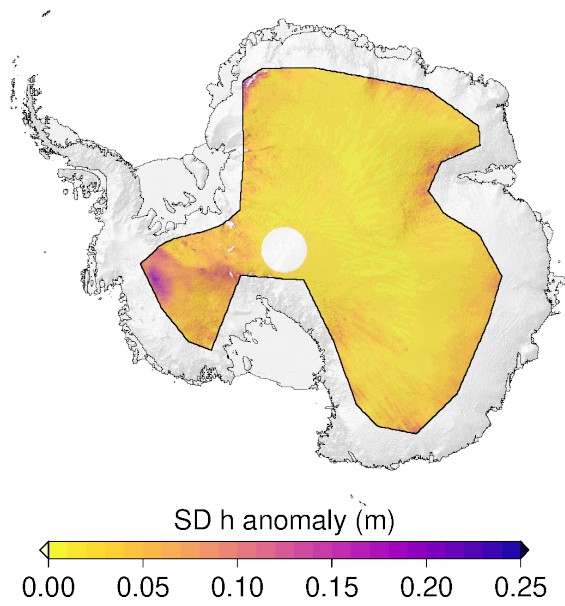

**SD h anomaly (m)**

| | | | | | |
|---|---|---|---|---|---|
| 0.00 | 0.05 | 0.10 | 0.15 | 0.20 | 0.25 |

**Figure 20.** ICESat-2 derived spatial distribution of the standard deviation of the elevation anomaly over the LRM zone in Antarctica. The elevation anomalies are based on grids with 5 km x 5 km pixel resolution of the spatially interpolated monthly residuals of the elevation trend estimation. The standard deviation is determined for each pixel covering the time period from January 2019 to December 2021.

data in (a) and (b), respectively. AWI-ICENet-1 is already in very close agreement with ICESat-2 without any correction and
only slightly improved with the correction. However, for TFMRA and ESA ICE1 the applied correction is very effective but still remaining signals are present. The correction is not able to suppress $SD(\Delta h)$ to less than 4 cm as only 30% of the area for TFMRA and ESA ICE1 but 60% for ICESat-2 and AWI-ICENet1 are observed. Our results illustrate, that backscatter and LEW corrections even if they are applied on a relatively short time span of 3 years, significantly improve the data, but are not able to fully correct erroneous signal contribution due to radar penetration. For longer time series, as in Fig. 19, the effect of
the correction for ice-sheet-wide observations is even smaller.

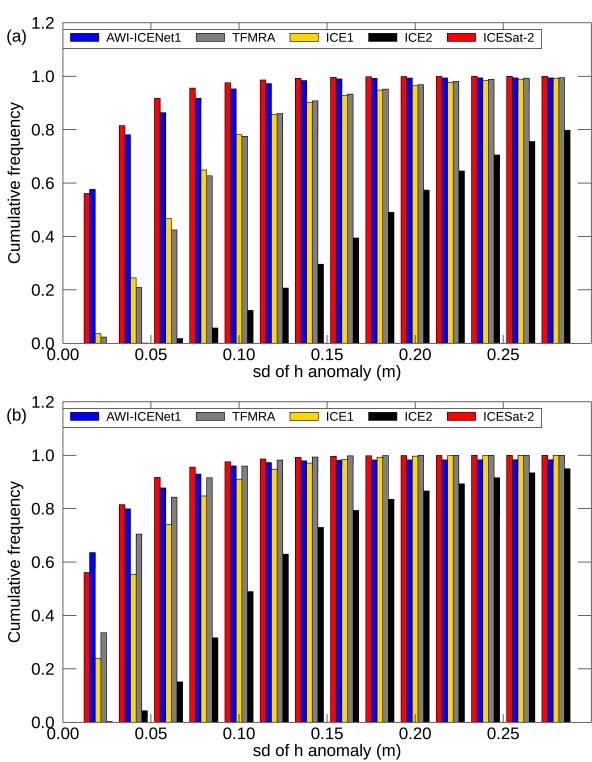

**Figure 21.** Cumulative histogram with bin size of 0.02 m of the standard deviation of the elevation anomaly over the LRM zone in Antarctica for all four retrackers and ICESat-2. (a) uncorrected and (b) corrected h anomalies using a correlation with LEW and backscatter. The elevation anomalies are based on grids with 5 km x 5 km pixel resolution of the spatially interpolated monthly residuals of the elevation trend estimation. The standard deviation is determined for each pixel covering the time period from January 2019 to December 2021.

### 4.2.3 Sudden changes in scattering properties and its effect on elevation change

Nilsson et al. (2015, 2016); McMillan et al. (2016); Simonsen and Sørensen (2017) presented and discussed the impact of the unusual melt event on Cryosat-2 observations that occurred over the Greenland Ice Sheet in July 2012. Nilsson et al. (2015) reported an improved performance by using a 20% threshold retracker for the LRM data with less sensitivity to changes in near-surface scattering properties, which is in close agreement with findings of Helm et al. (2014). On the other hand McMillan et al. (2016) introduced a step function to mitigate the observed elevation step. This procedure however, is not applicable as a general approach as multiple melt events, which can be expected in future, cannot be taken into account. Simonsen and Sørensen (2017) studied in detail, by using the ESA LRM_L2 data, the implication of different waveform parameters (LEW and backscatter) to correct for the scattering properties in Cryosat-2 observations for the time span November 2010 until November 2014. Their findings suggest for the LRM zone as best approach the LEW correction. They also found that the bias of temporal changes is not entirely removed which agrees with our results for ESA ICE1 and ESA ICE2. In order to evaluate if the AWI-ICENet1 is capable to handle sudden scattering changes we applied the dhdt processing for all retrackers to the time span January 2011 until December 2014 covering nearly the same time span as investigated in Simonsen and Sørensen (2017). In Fig. 22 the uncorrected dhdt and in Fig. A17 the corrected dhdt using LEW and backscatter are shown. The same unusual elevation increase with mean rates of 0.1–0.2 $m/yr$ is observed with ESA ICE1 and TFMRA for large parts of the high elevated area. ESA ICE2 shows rates of $> 0.25 \, m/yr$ for almost the whole LRM zone. Only the South Eastern part experiences a surface lowering. In contrast, the AWI-ICENet1 derived dhdt ranges between -0.05 to 0.05 $m/yr$ and does not seem affected by the change of scattering characteristics due to the surface melt event. When compared to the corrected dhdt estimates we find the best agreement with ESA ICE1 and TFMRA. In agreement with Simonsen and Sørensen (2017) we find that in the corrected TFMRA, ESA ICE1 and ESA ICE2 result still some remaining signal is left. The application of a low threshold retracker as suggested by Nilsson et al. (2015) strongly reduces the sensitivity to scattering changes but still needs to be combined with waveform parameter corrections to provide reliable results. Only the AWI-ICENet1 incorporates most of the correction directly within the retracking. The differences between uncorrected and corrected dV/dt estimates are within the uncertainty of $8 \, km^3/yr$ as listed in Table 3. AWI-ICENet1 is in close agreement with the ESA ICE1-LEW estimates. These are found to match best to surface elevation changes measured by LiDAR during the Operation Icebridge campaigns, as presented in Simonsen and Sørensen (2017), and agree well with our corrected TFMRA estimates.

| Correction | AWI-ICENet1 | ICE1 | ICE2 | TFMRA |
|---|---|---|---|---|
| none | $-74 \pm 7$ | $-42 \pm 13$ | $156 \pm 27$ | $-43 \pm 14$ |
| LEW | $-70 \pm 8$ | $-64 \pm 12$ | $60 \pm 20$ | $-78 \pm 13$ |
| Backscatter | $-73 \pm 7$ | $-39 \pm 12$ | $169 \pm 25$ | $-35 \pm 13$ |
| LEW and backscatter | $-67 \pm 7$ | $-57 \pm 11$ | $72 \pm 19$ | $-73 \pm 12$ |

**Table 3.** Volume change $dV/dt$ estimates for the time span 2011/01–2014/12 for Greenland derived by different retrackers and corrections. All values are given in $(km^3/yr)$.

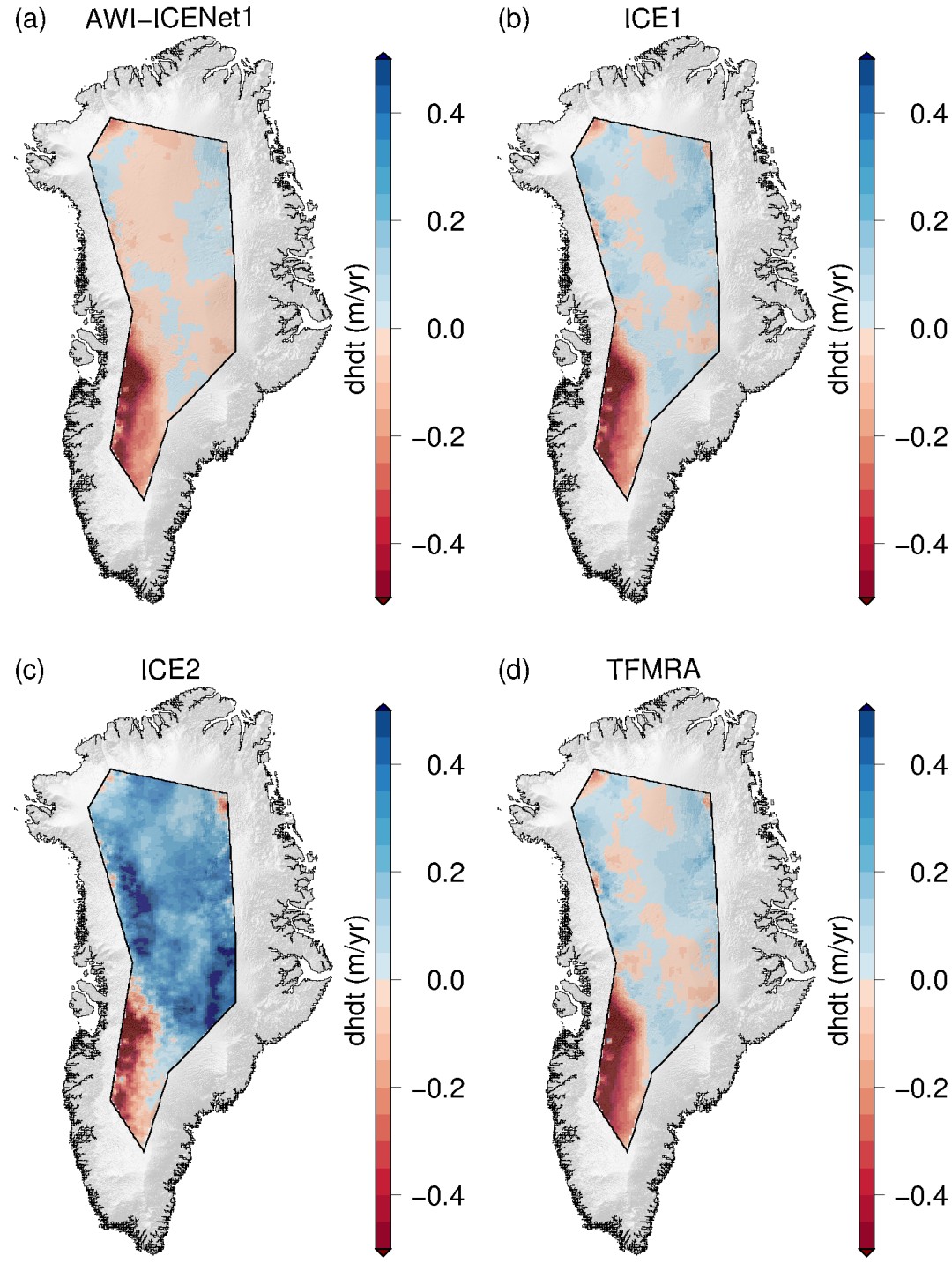

**Figure 22.** Greenland wide dhdt estimates of four retrackers for the period from January 2011 to December 2013 , which includes the July melt event in 2012.

### 4.2.4 Comparison of elevation change estimates to ICESat-2

As an additional way to assess the performance of the retrackers, we conduct a comparison of elevation change (dhdt) to ICESat-2 for the time span from January 2019 to December 2021. The difference in elevation change (dhdt-CryoSat-2 – dhdt-ICESat-2) (dhdt difference) is displayed in Fig. 23 for each retracker individually. Again the pattern of ESA ICE1 and TFMRA are similar spatially as well as magnitude-wise. ESA ICE2 is furthest off the ICESat-2 elevation change. The new AWI-ICENet1 deviates by less than $\pm0.1$ m/yr from ICESat-2 over approx. 90% of the entire area. In Table 4 estimated volume changes ($dV/dt$) observed in the LRM zone for the time span from January 2019 to December 2021 are listed for ICESat-2 and all retrackers including the applied corrections for both Ice Sheets. In Antarctica, the difference between switching the correction on/off is tremendously high for ICE2 ranging from $84\,\mathrm{km}^3$/yr to $276\,\mathrm{km}^3$/yr, followed by TFMRA with a $72\,\mathrm{km}^3$/yr and ESA ICE1 with a $56\,\mathrm{km}^3$/yr spread. AWI-ICENet1 shows the smallest range with only $9\,\mathrm{km}^3$/yr indicating that most of the corrections are already covered by the retracker itself. The results also show the absolute need to apply the corrections for all retrackers except AWI-ICENet1 as otherwise $dV/dt$ and in consequence mass balance estimates are unreliable, especially for East Antarctica. The uncorrected estimates are prone to errors which are strongly related to the elevation anomalies originating from changes in the electromagnetic properties of the upper layers of the snow/firn pack. In Greenland the same conclusion can be drawn. Here, the spread of $dV/dt$ is smaller. The spread is $41\,\mathrm{km}^3$/yr for ESA ICE2, $8\,\mathrm{km}^3$/yr for ESA ICE1, $6\,\mathrm{km}^3$/yr for TFMRA, and only $4\,\mathrm{km}^3$/yr for AWI-ICENet1. In all cases the uncertainty of the estimates could be reduced using the combined correction of LEW and backscatter. The smallest spread between the different retracker solutions is also found for LEW and backscatter corrections applied for Antarctica and Greenland, respectively. Interestingly, as shown in Figures 15, 18, A11 and A12 none of the corrections applied to ESA ICE2, ESA ICE1 and TFMRA are capable to reduce the $\Delta h$ as strong as the AWI-ICENet1. This is reflected especially for Greenland in Table 4 where none of the $dV/dt$ estimates are reaching values around $5\,\mathrm{km}^3$/yr as observed by ICESat-2. For Greenland and Antarctica AWI-ICENet1 also reveal smallest offsets to ICESat2, in both cases the difference of $dV/dt$ is less than $20\,\mathrm{km}^3$/yr, where ICESat-2 tend to show larger values (compared to panel (a) of Figures 23, 24, A15 and A16). This small $< 1\,\mathrm{cm}$/yr difference in dhdt to ICESat-2 is most likely a residual signal due to radar penetration and not related to any kind of ICESat-2 mission bias as observed for the predecessor mission ICESat-1 (NSIDC, 2021). Sea-level trends computed from ICESat-2 observations agree with independent measurements from radar altimetry and tide gauges as shown by Buzzanga et al. (2021) and do not indicate any trend bias, at least over oceans. In Antarctica the best match with AWI-ICENet1 is found for TFMRA with LEW and backscatter corrections applied. In the appendix in Figures A5, A6, A7, A8 the correlations between the elevation anomalies and LEW and backscatter are shown, respectively. For AWI-ICENet1 the correlations are lower than for the other three retracker but this is expected, since the h anomalies are already strongly suppressed by AWI-ICENet1 which should lead to a much lower correlation with backscatter or LEW. Especially the correction with backscatter needs to be carefully checked as it can introduce errors, which most likely originate from an error in the power scaling factors applied in the L1B LRM data as reported by ESA (Exprivia, 2021). This caused the backscatter computed at Level 1 to differ between Baseline D and Baseline E resulting in a jump in backscatter in

August 2019 when Baseline E was operational (Exprivia, 2021). The issue has been solved in the reprocessed Baseline E data. In our processing we used this reprocessed Baseline E data and thus don't expect any issues with the backscatter.

| Area | Correction | ICESat2 | AWI-ICENet1 | ESA ICE1 | ESA ICE2 | TFMRA |
|------|-----------|---------|-------------|----------|----------|-------|
| ANT | none | $96 \pm 6$ | $93 \pm 14$ | $159 \pm 23$ | $276 \pm 57$ | $162 \pm 26$ |
| | LEW | | $87 \pm 13$ | $130 \pm 21$ | $136 \pm 45$ | $117 \pm 23$ |
| | Backscatter | | $85 \pm 13$ | $112 \pm 18$ | $143 \pm 42$ | $100 \pm 20$ |
| | LEW and backscatter | | $84 \pm 13$ | $103 \pm 17$ | $84 \pm 37$ | $90 \pm 19$ |
| GRE | none | $5 \pm 6$ | $-24 \pm 7$ | $-48 \pm 10$ | $-88 \pm 17$ | $-45 \pm 11$ |
| | LEW | | $-27 \pm 7$ | $-40 \pm 11$ | $-47 \pm 18$ | $-47 \pm 13$ |
| | Backscatter | | $-23 \pm 7$ | $-41 \pm 10$ | $-74 \pm 19$ | $-41 \pm 12$ |
| | LEW and backscatter | | $-24 \pm 7$ | $-40 \pm 10$ | $-52 \pm 17$ | $-45 \pm 11$ |

**Table 4.** Volume change $dV/dt$ estimates for the time span 2019/01–2021/12 for Greenland (GRE) and Antarctica (ANT) derived by different retrackers compared to ICESat2. All values are given in $(km^3/yr)$.

### 4.2.5 Computational performance

Unlike many other machine learning applications used in remote sensing, AWI-ICENet1 was not designed to replace manual labour or save time, but to improve observation quality. In fact, our computational performance test results, shown in Table 2, highlight that AWI-ICENet1 requires more processing time than conventional empirical retracking methods. In particular, AWI-ICENet1 requires about 8 and 18 times as much computing time compared to the low computing costs of cheap empirical retracking methods such as TFMRA and TCOG, respectively. However, if the estimation of the LEW is also taken into account in TCOG and TFMRA, the difference in processing time compared to AWI-ICENet1 is considerably reduced. Therefore, we do not consider the higher computational cost of AWI-ICENet1 to be a significant drawback for its use, regardless of the computing infrastructure. Compared to more complicated waveform fitting methods based on analytical descriptions of the waveform, such as dedicated classical ocean retrackers like MLE3/4 (Amarouche et al., 2004; Thibaut et al., 2010), adaptive MLE3/4 (Thibaut et al., 2021), SAMOSA+ (Dinardo et al., 2018) or SAMOSA++ (Dinardo et al., 2021), we would expect that CNN-based approaches can help to significantly reduce processing costs. Re-processing campaigns could benefit from neural network-based approaches. However, for each of the analytical re-trackers mentioned above, a specific CNN model that best represents the analytical solution would have to be trained in advance with a considerable number of waveforms covering the entire spectrum of possible waveforms.

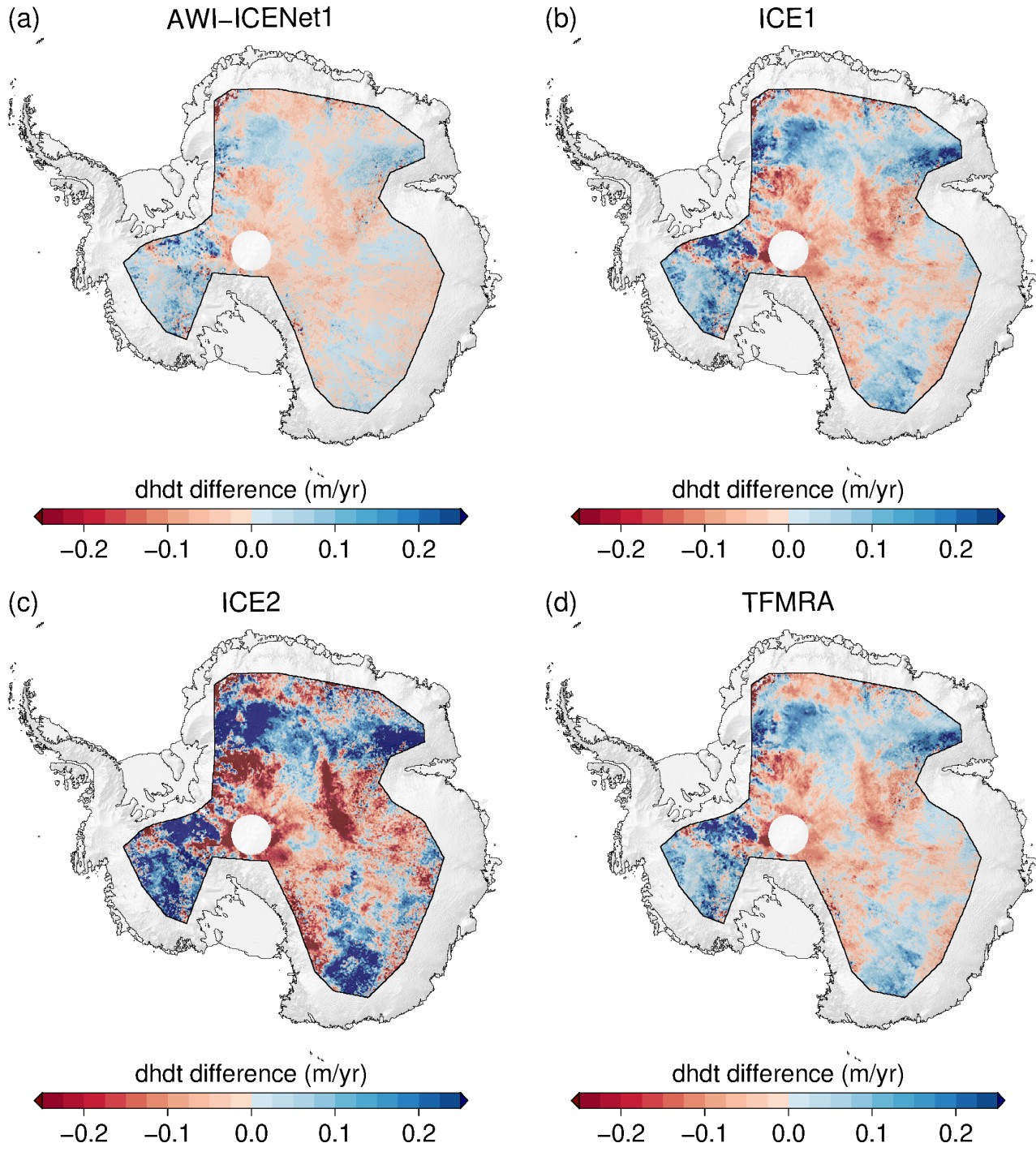

**Figure 23.** Antarctic wide dhdt difference to ICESat2 of four retrackers for the period from January 2019 to December 2021

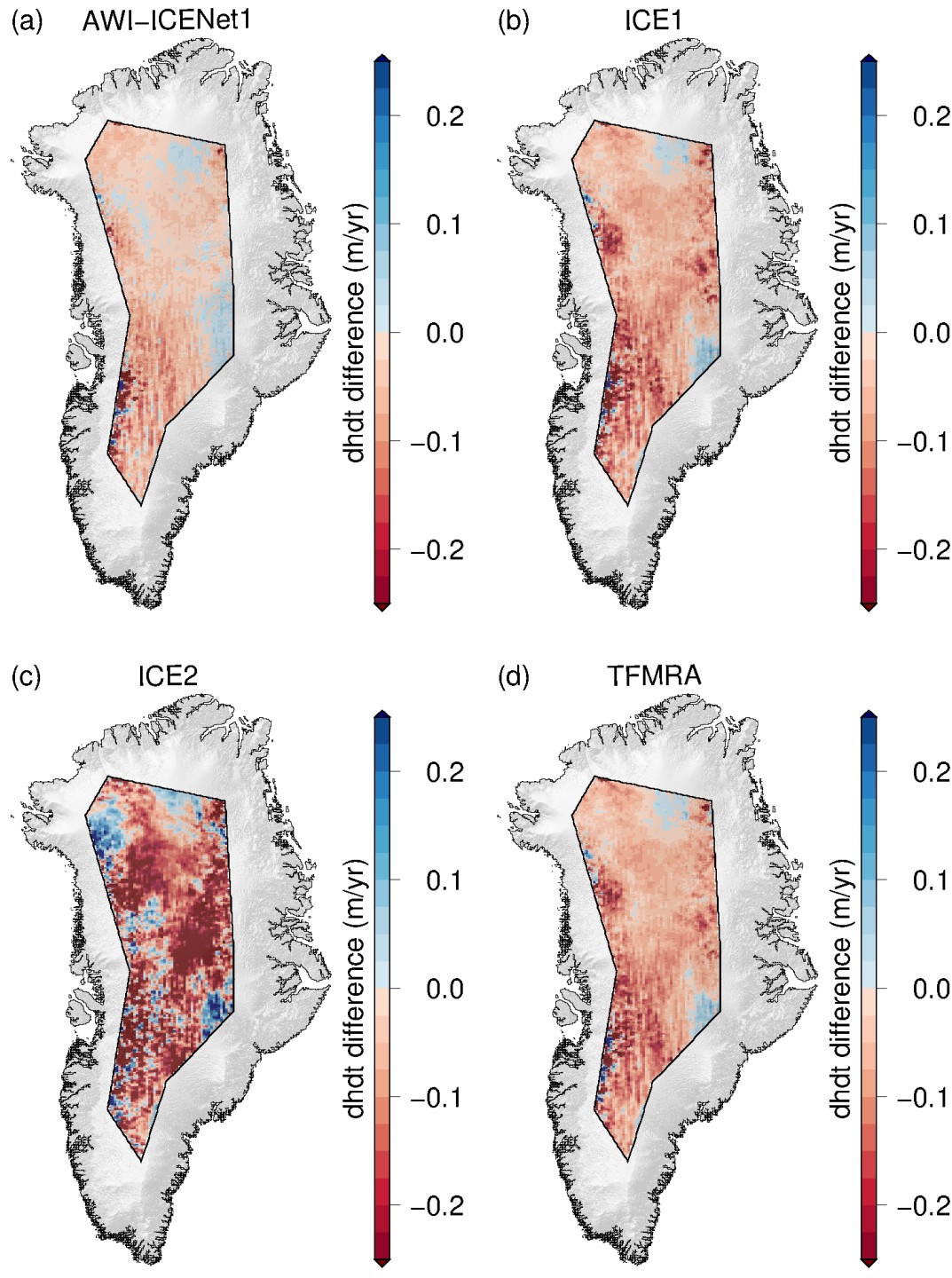

**Figure 24.** Greenland wide dhdt difference to ICESat2 of four retrackers for the period from January 2019 to December 2021.

## 5  Conclusions

We conclude that the use of the AWI-ICENet1 a new retracker for ice altimetry based on a convolutional network provide better performance in deriving ice sheet wide elevation products than currently used retracker, such as the TFMRA or the ESA Level2 ICE1 and ICE2 products. In all terrain types, the performance of AWI-ICENet1 in terms of CPE is similar to that of ESA Level2 ICE1, the highest precision retracker. The main improvement obtained with our new approach is the large reduction in sensitivity to transient signal penetration due to temporal changes in near-surface scattering properties of the firn compared to other approaches. The transient penetration bias can be in the order of several decimeters and is reduced to a few centimeters by AWI-ICENet1. AWI-ICENet1 can handle abrupt changes in the prevailing scattering mechanism due to surface melt, as occurred over Greenland in July 2012, as well as temperature-induced changes of the bulk attenuation rate of the upper firn layers, which are mainly observed as a seasonal signal in East Antarctica. The contribution of the transient penetration to the measured elevation anomaly is greatly reduced compared to equivalent products, even when compared to empirically corrected data using a correlation between elevation changes and changes in backscatter or waveform shape parameters such as LEW. This results in a much more accurate sampling of the true elevation change due to changes in SMB or firn compaction, and estimates agree well with ICESat-2 derived estimates that are not susceptible to transient penetration. Using the new retracking method for CryoSat-2 LRM data, we determine the volume change in the LRM zone of the Antarctic and Greenland Ice Sheets for the period January 2019 to December 2021 to be $84 \pm 13 \ km^3 a^{-1}$ and $-24 \pm 7 \, km^3 a^{-1}$, respectively. This agrees well with ICESat-2 estimates obtained over both ice sheets during the same period ($96 \pm 6 \, km^3 a^{-1}$ and $5 \pm 6 \, km^3 a^{-1}$). The improvements in our new retracking approach reduce the need of correlation corrections in post-processing, which can introduce additional bias as a result of anomalous backscatter variations due to different processing baselines. In addition, empirical corrections are sensitive to the length of the time series used to apply the correction. The new retracking method presented, provides a higher intrinsic accuracy in the measured surface elevation and thus lower uncertainty in derived products such as elevation change, and volume change estimates. This will lead to an improved understanding of the response of ice sheets to climate change, especially in areas of low elevation change such as the East Antarctic Plateau, where high measurement accuracy is required to separate true elevation change from noise originating from transient radar penetration.

In the future, AWI-ICENet1 should also be applied to pulse-limited altimetry products (LRM and PLRM) of past (Envisat, ERS 1/2), present (Sentinel-3 A/B, SARAL/Altika), and future (CRISTAL) altimetry missions to extend the time series and to realize a consistent elevation change product with the least possible penetration biased estimates. Furthermore an extension of the waveform simulator to SAR waveforms would be of high interest to be able to apply a similar approach to CRISTAL and Sentinel-3 SAR data.

*Data availability.*  The complete simulated reference data set, elevation change grids as well as monthly elevation anomalies were uploaded to the World Data Center PANGAEA and can be found in the given reference (Helm, 2024) or following DOI:10.1594PANGAEA.964596.

**Appendix A**

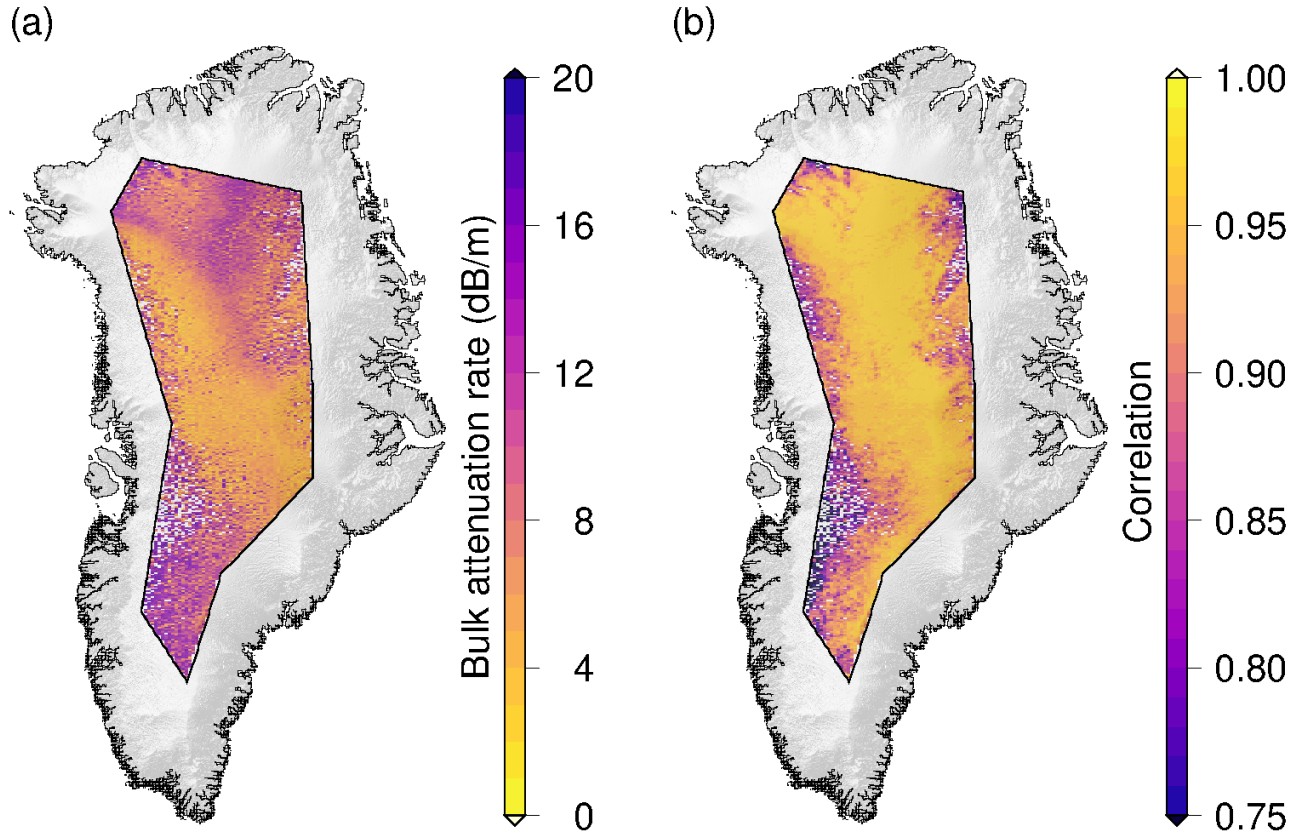

**Figure A1.** Results of the test where a flat surface waveform model is fitted to real CryoSat-2 waveforms by adjusting the attenuation $L_A$ as the only parameter. (a) gridded median of the attenuation rate estimated by a MLE fit and (b) median of the correlation between observed and fitted waveforms.

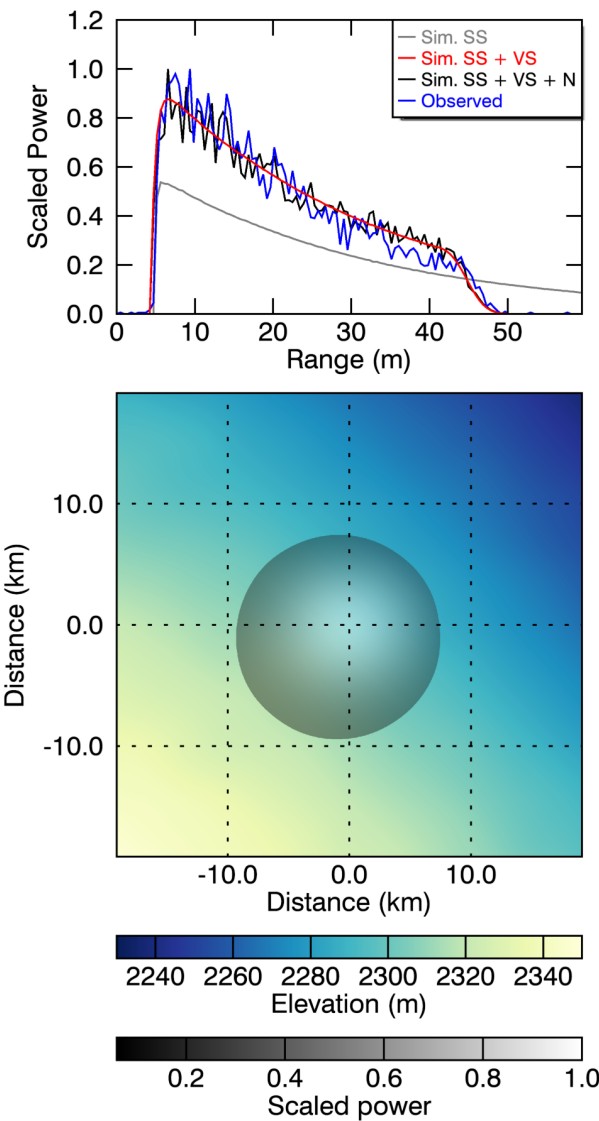

**Figure A2.** Example of the different simulation steps shown for a typical waveform around the NEEM ice core site in Greenland. Blue line depicts the observed CryoSat-2 LRM waveform and the gray line the simulated surface waveform based on Eq. 1. The Red line shows the simulated waveform including volume scattering according to Eq. 3, while the black line displays the final simulated waveform including surface and volume scattering as well as noise as given in Eq.5. The lower panel represents the 2D elevation model and scaled $P_{r_s}$, which is mainly controlled by the Gaussian antenna pattern.

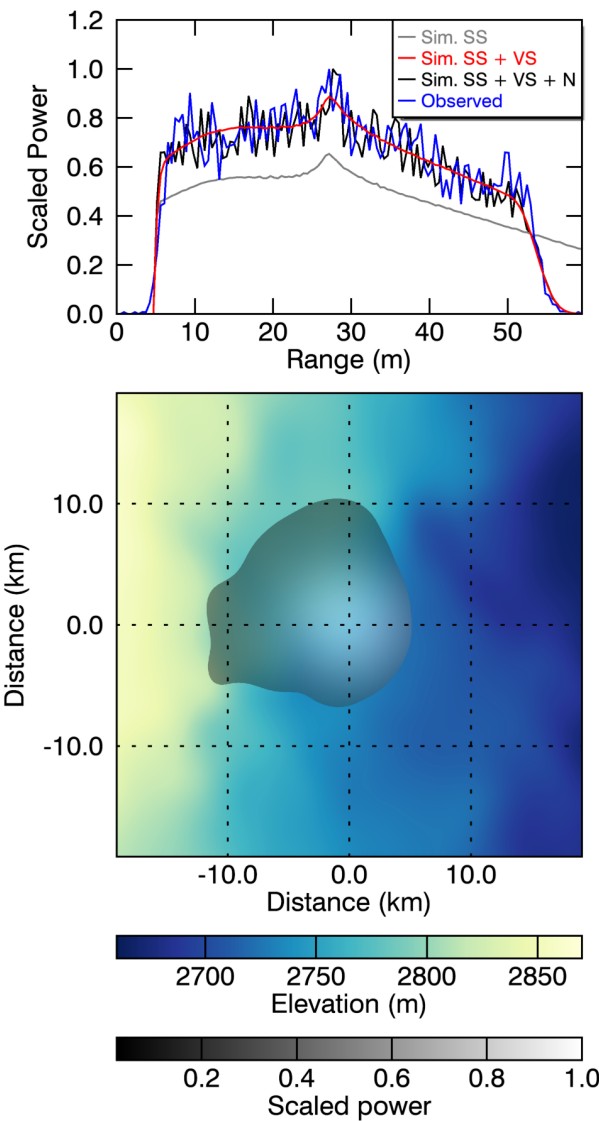

**Figure A3.** Example of the different simulation steps shown for a typical waveform in the Amery drainage basin (region A in 1). Blue line depicts the observed CryoSat-2 LRM waveform and the gray line the simulated surface waveform based on Eq. 1. The Red line shows the simulated waveform including volume scattering according to Eq. 3, while the black line displays the final simulated waveform including surface and volume scattering as well as noise as given in Eq.5. The lower panel represents the 2D elevation model and scaled $P_{r_s}$, which is mainly controlled by the Gaussian antenna pattern.

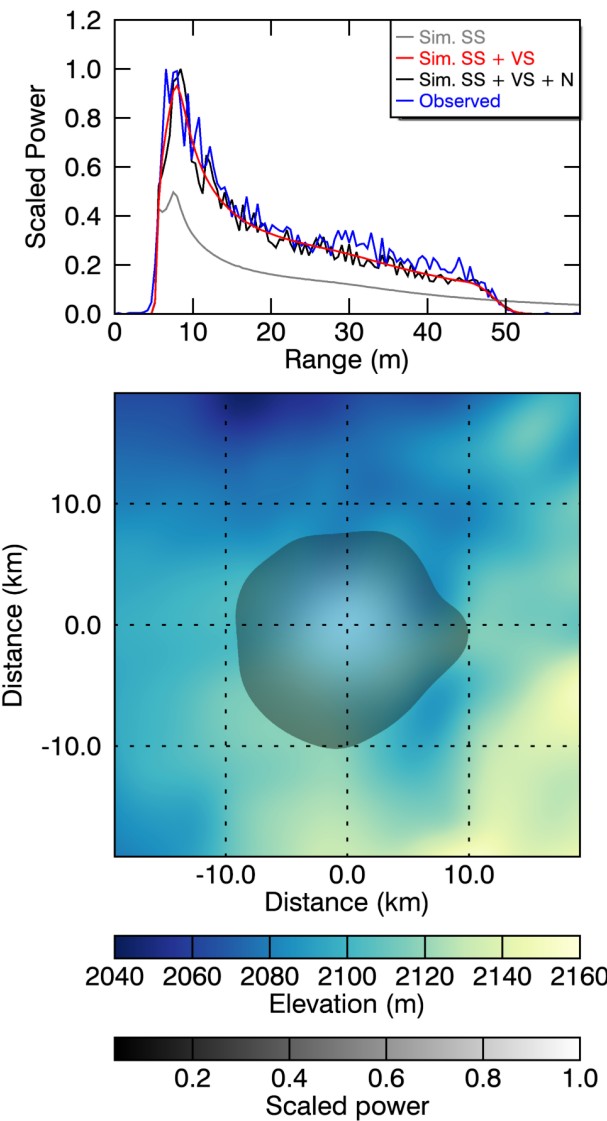

**Figure A4.** Example of the different simulation steps shown for a typical waveform in the Recovery area (see 1).. Blue line depicts the observed CryoSat-2 LRM waveform and the gray line the simulated surface waveform based on Eq. 1. The Red line depicts the simulated waveform including volume scattering according to Eq. 3, while the black line shows the final simulated waveform including surface and volume scattering as well as noise as given in Eq.5. The lower panel represents the 2D elevation model and scaled $P_{r_s}$, which is mainly controlled by the Gaussian antenna pattern.

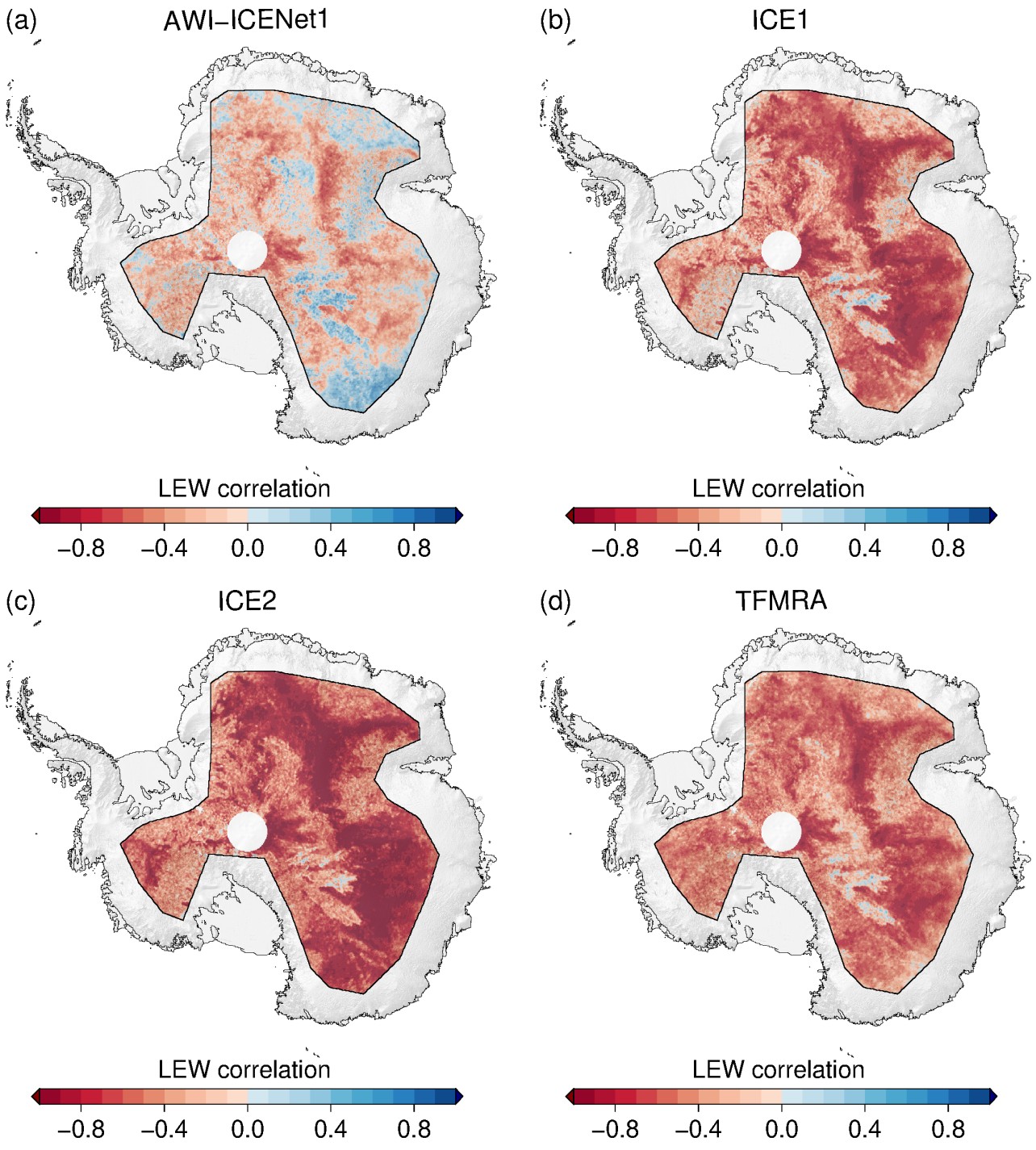

**Figure A5.** Correlation between elevation anomalies and LEW for different retracker over Antarctica. The correlation is based on the full time series from January 2011 to December 2022.

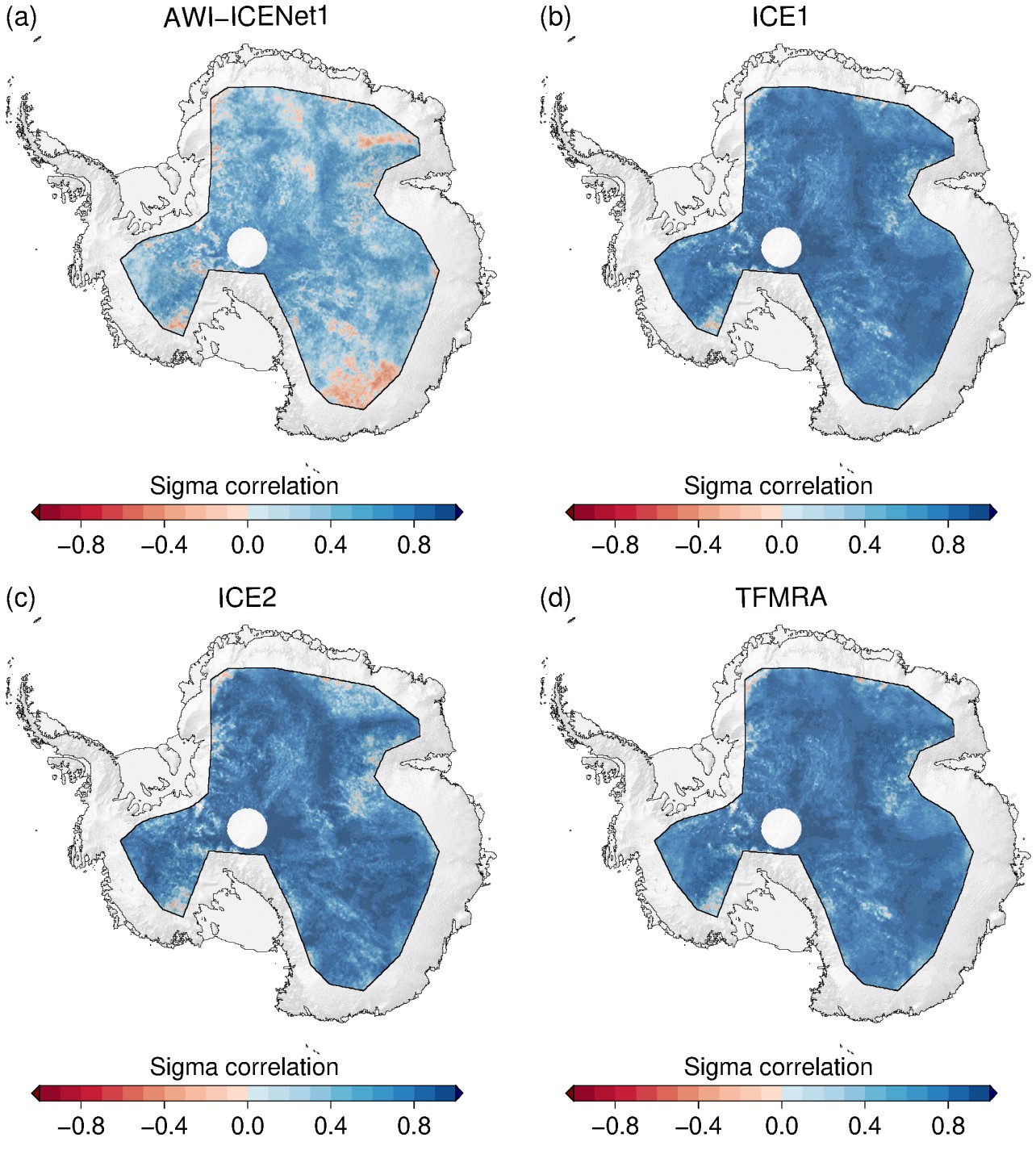

**Figure A6.** Correlation between elevation anomalies and backscatter for different retracker over Antarctica. The correlation is based on the full time series from January 2011 to December 2022.

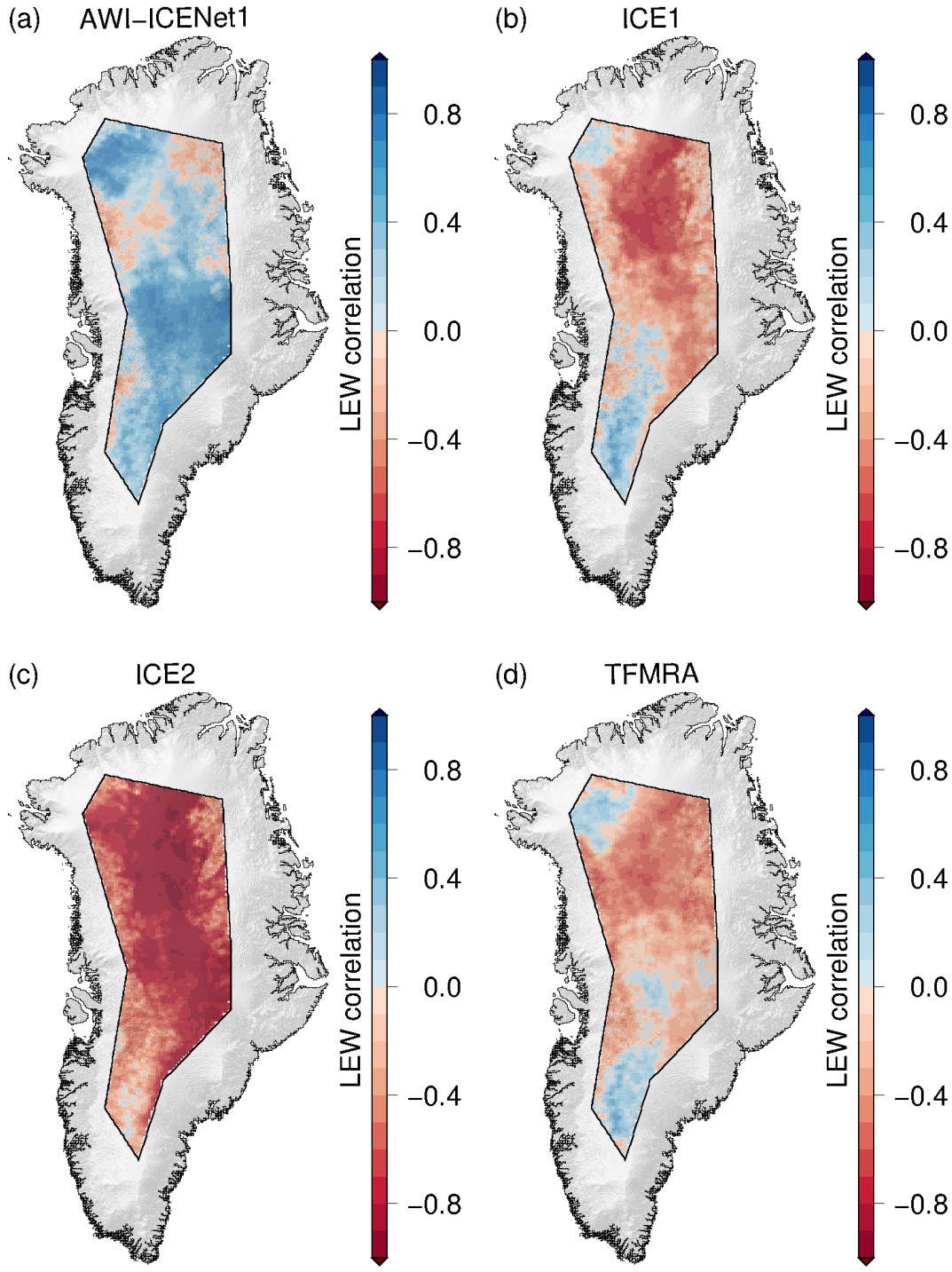

**Figure A7.** Correlation between elevation anomalies and LEW for different retracker over Greenland. The correlation is based on the full time series from January 2011 to December 2022.

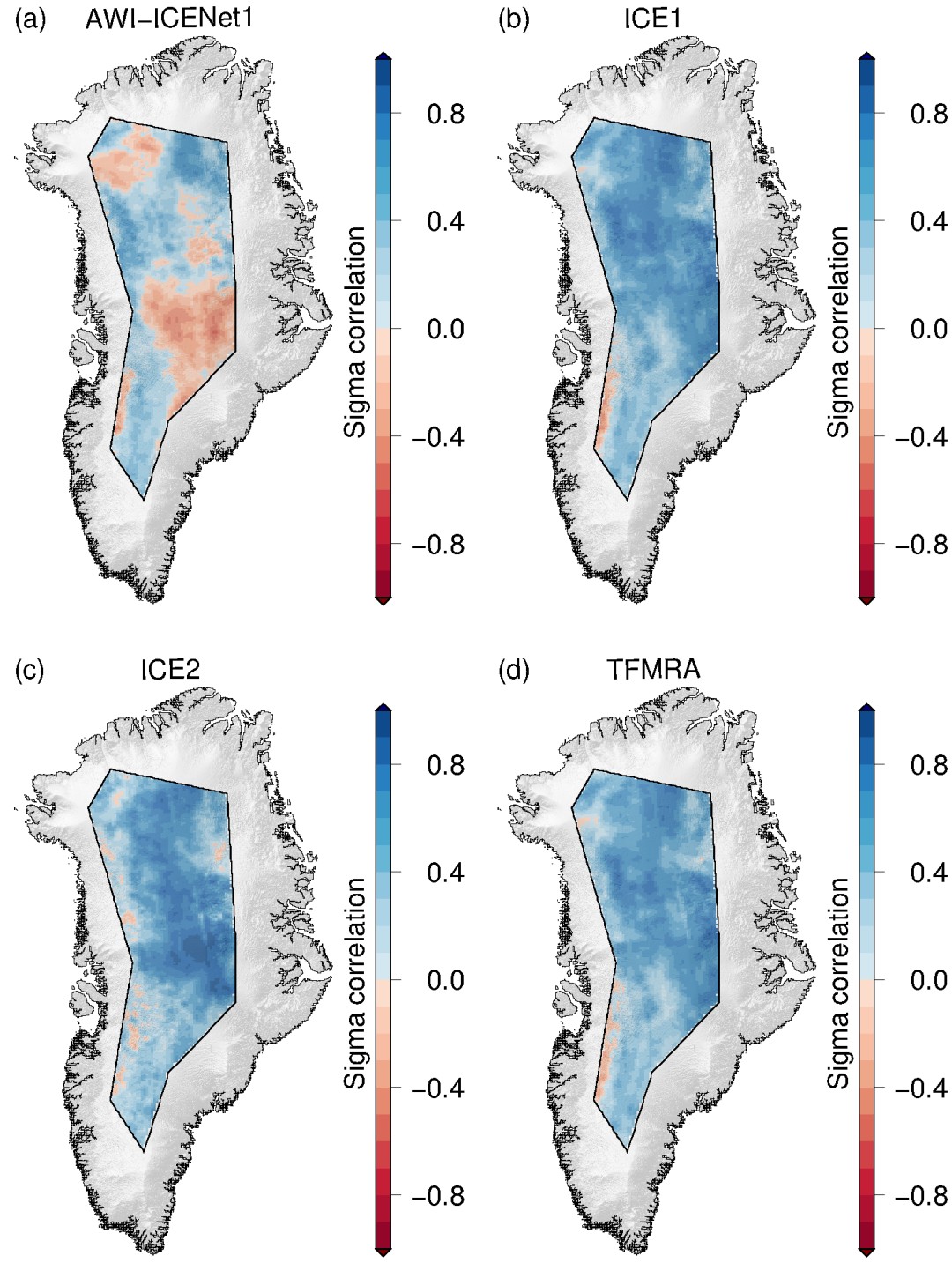

**Figure A8.** Correlation between elevation anomalies and backscatter for different retracker over Greenland. The correlation is based on the full time series from January 2011 to December 2022.

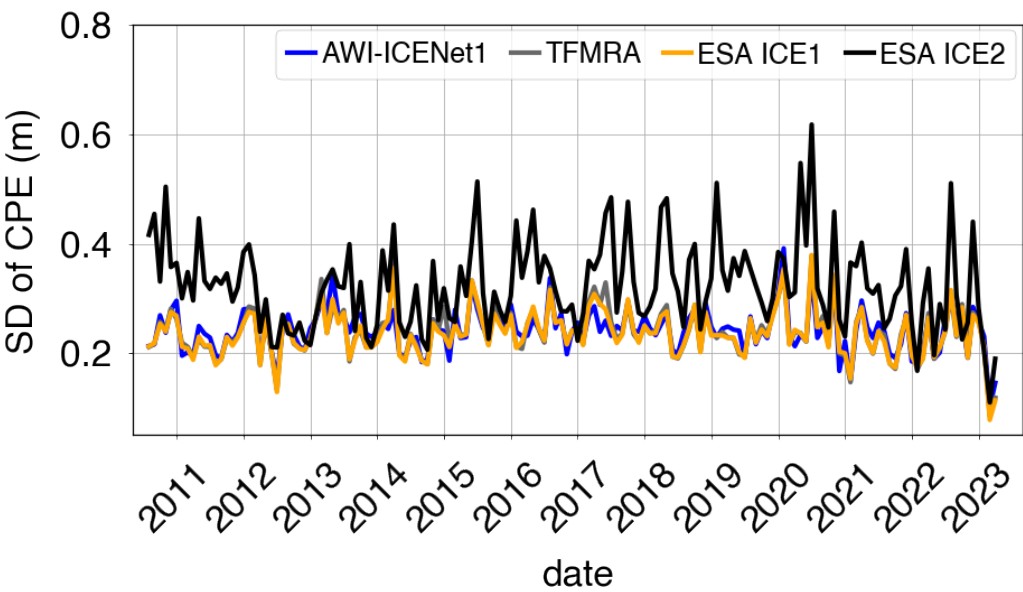

**Figure A9.** Time series of the standard deviation of cross-point errors for a region in North-Greenland. Cross-point-errors are determined in monthly resolution.

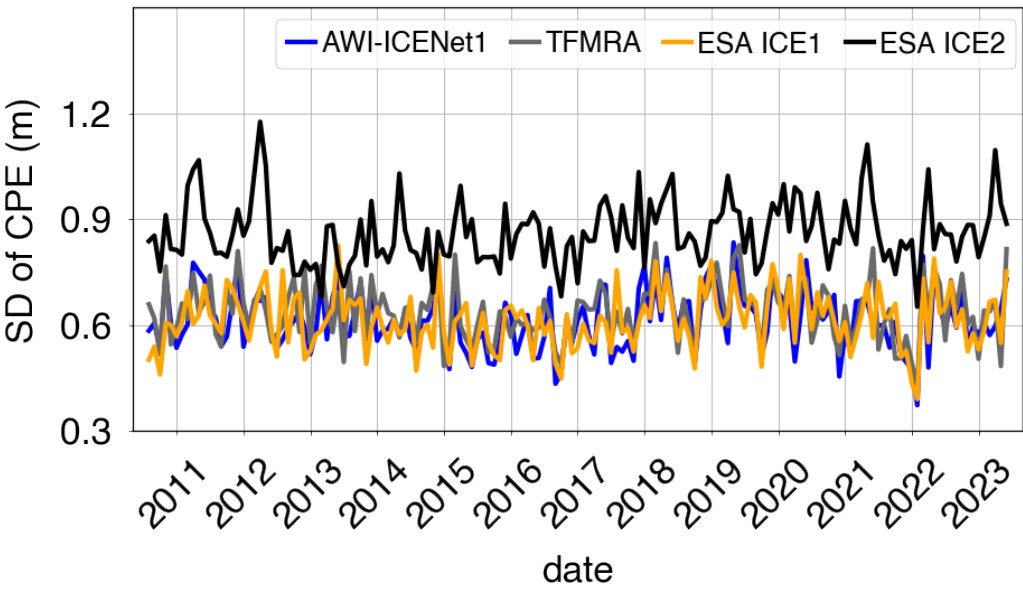

**Figure A10.** Time series of the standard deviation of cross-point errors for the LRM zone in Greenland. Cross-point-errors are determined in monthly resolution.

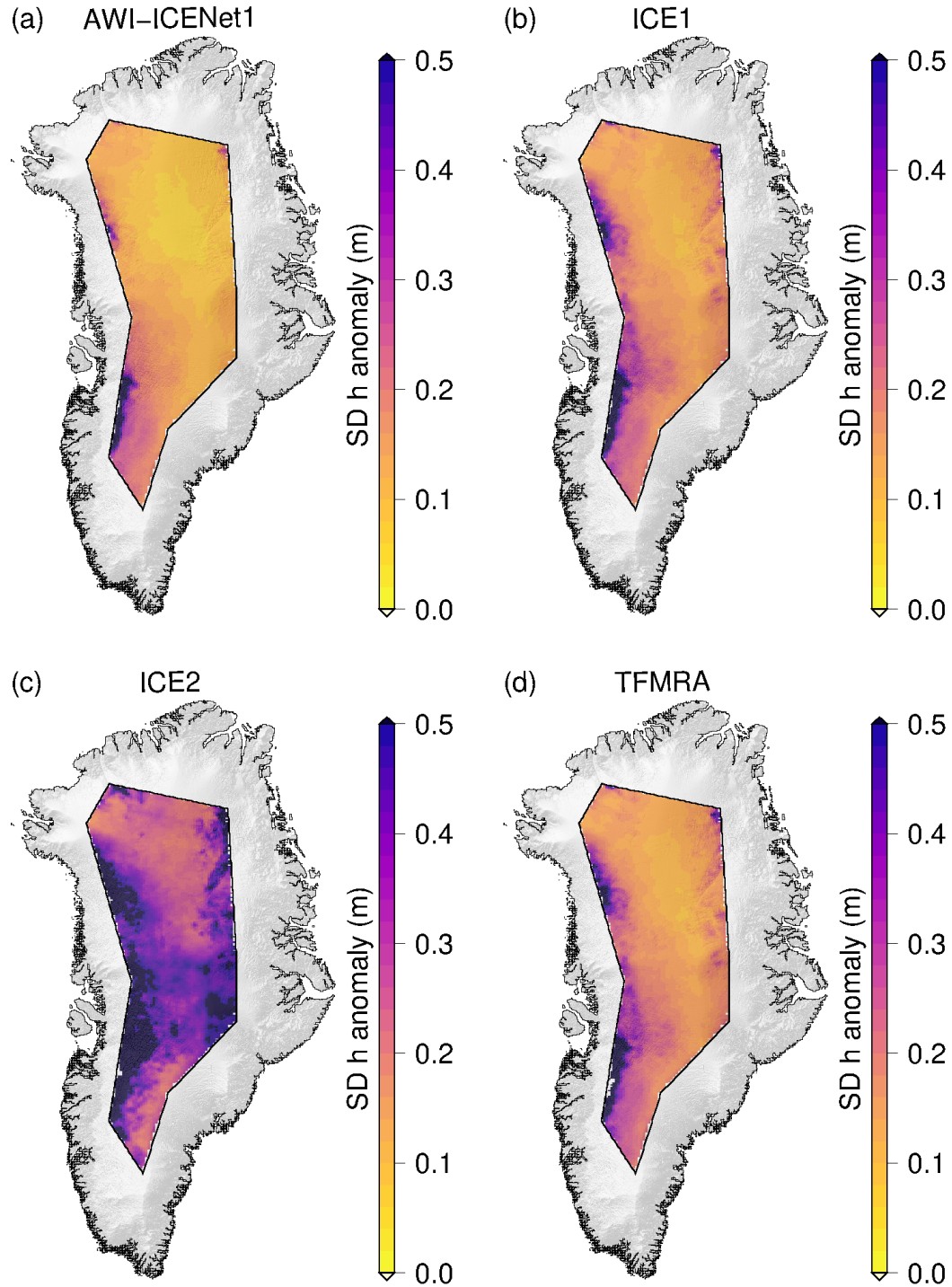

**Figure A11.** Spatial distribution of the standard deviation of the elevation anomaly over the LRM zone in Greenland for all four retrackers. The elevation anomalies are based on grids with 5 km x 5 km pixel resolution of the spatially interpolated monthly residuals of the elevation trend estimation. The standard deviation is determined for each pixel covering the full time period from January 2011 to December 2022.

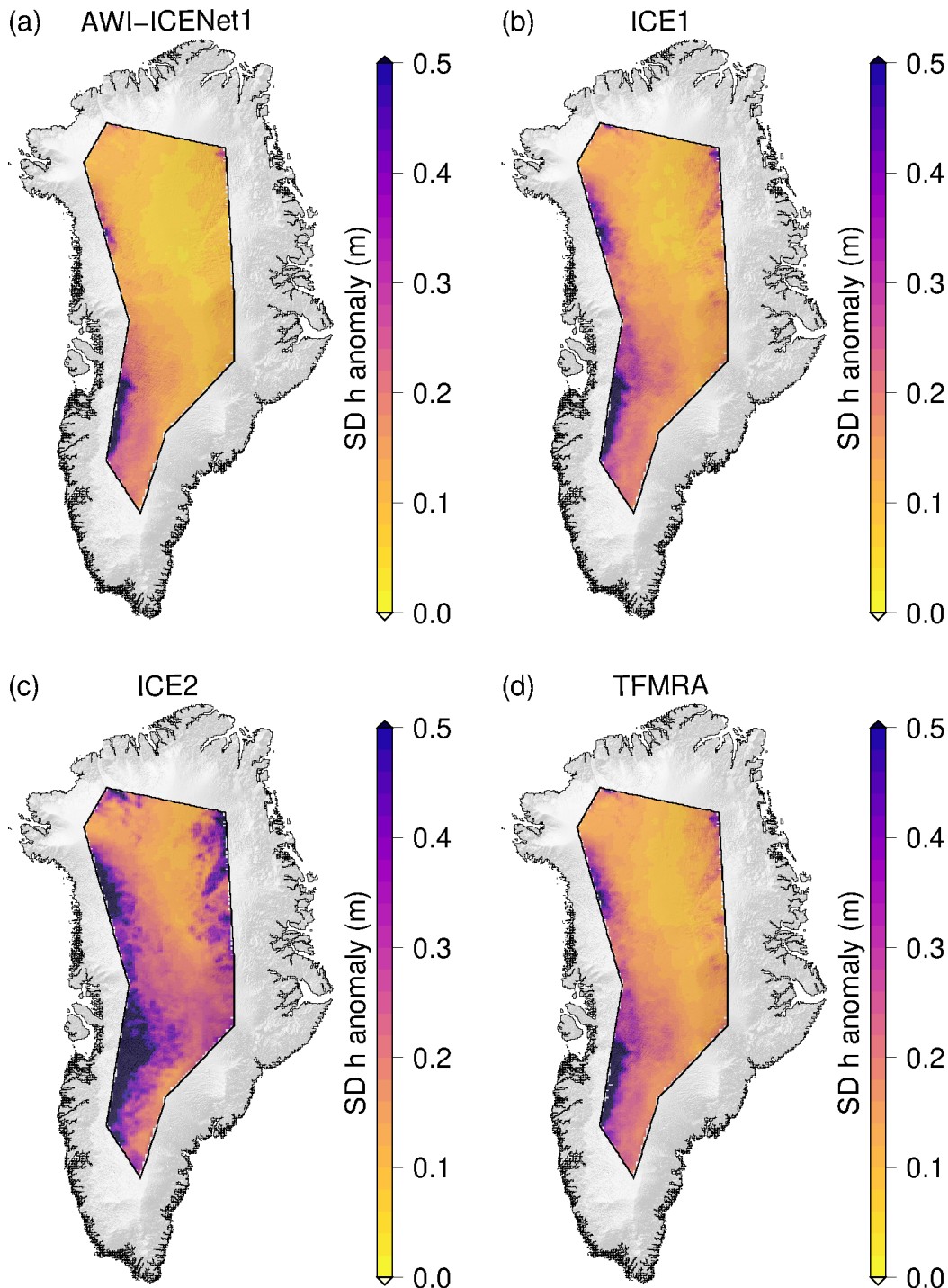

**Figure A12.** Spatial distribution of the standard deviation of the corrected elevation anomaly over the LRM zone in Greenland for all four retrackers. The elevation anomalies are based on grids with 5 km x 5 km pixel resolution of the spatially interpolated monthly residuals of the elevation trend estimation corrected using a correlation with leading edge width and backscatter. The standard deviation is determined for each pixel covering the full time period from January 2011 to December 2022.

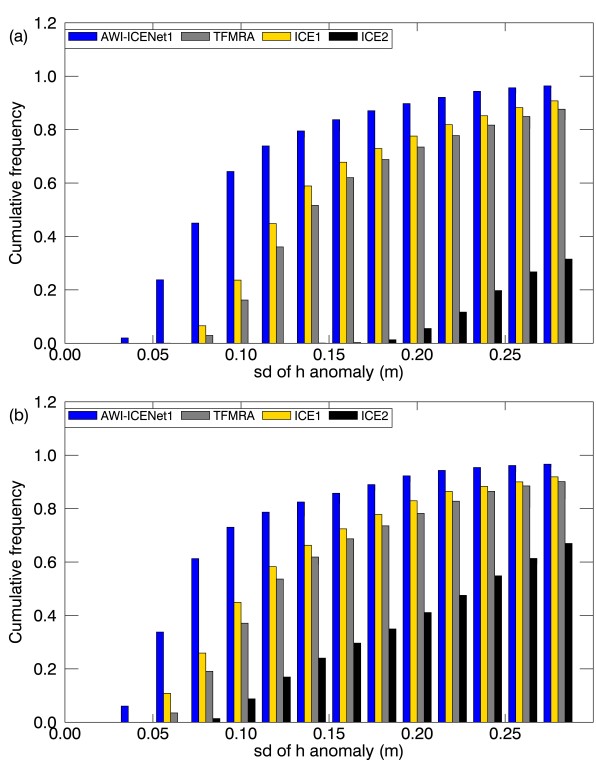

**Figure A13.** Cumulative histogram with bin size of 0.02 m of the standard deviation of the elevation anomaly over the LRM zone in Greenland for all four retrackers. (a) uncorrected and (b) corrected h anomalies using a correlation with LEW and backscatter. The elevation anomalies are based on grids with 5 km x 5 km pixel resolution of the spatially interpolated monthly residuals of the elevation trend estimation. The standard deviation is determined for each pixel covering the full time period from January 2011 to December 2022.

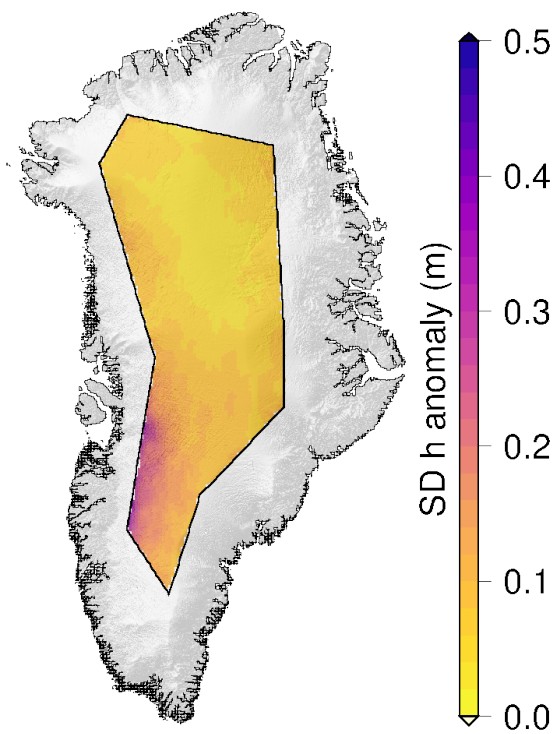

**Figure A14.** ICESat-2 derived spatial distribution of the standard deviation of the elevation anomaly over the LRM zone in Greenland. The elevation anomalies are based on grids with 5 km x 5 km pixel resolution of the spatially interpolated monthly residuals of the elevation trend estimation. The standard deviation is determined for each pixel covering the time period from January 2019 to December 2021.

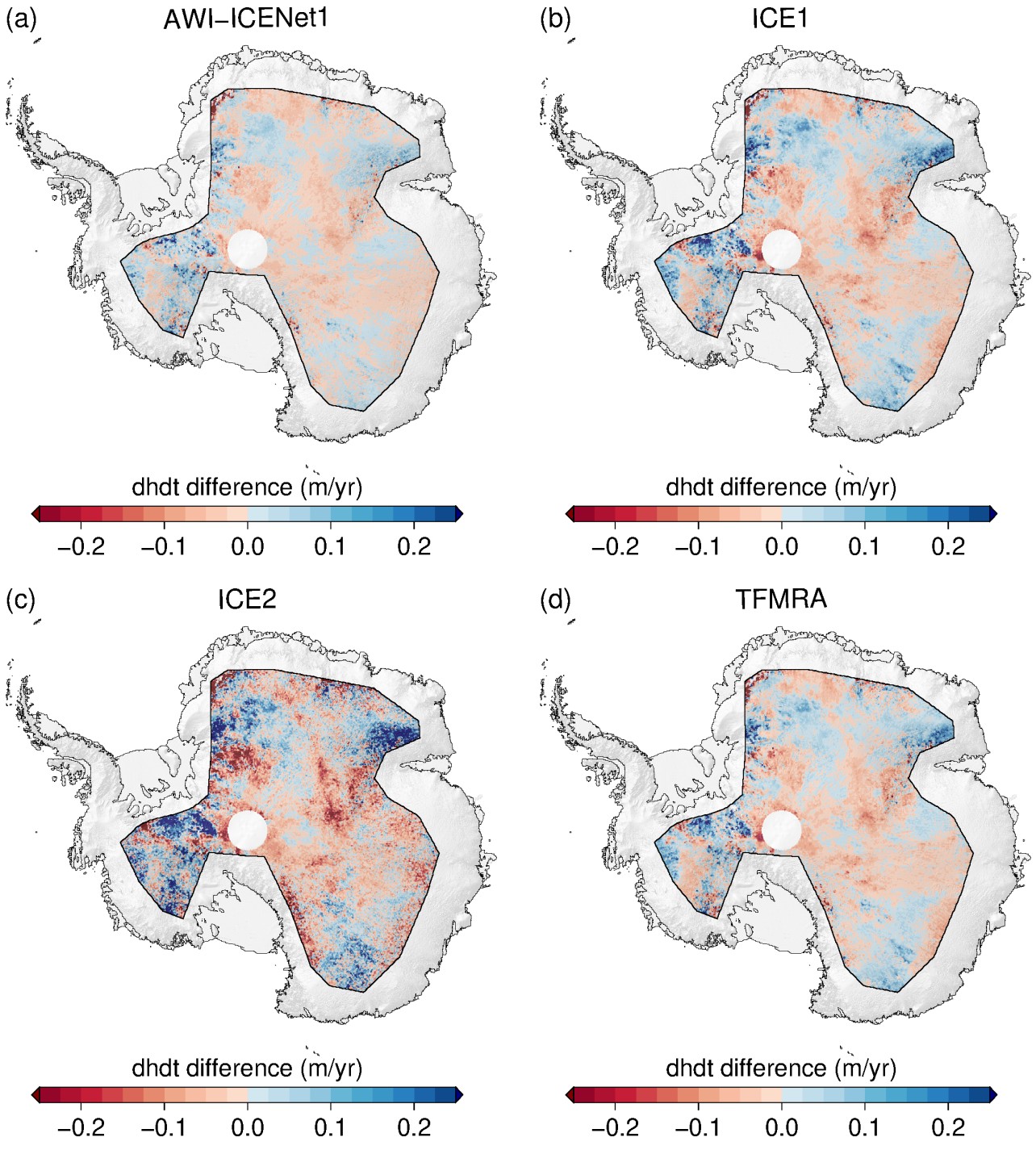

**Figure A15.** Antarctic wide dhdt difference to ICESat2 of four retrackers for the period from January 2019 to December 2021 with LEW and backscatter correction.

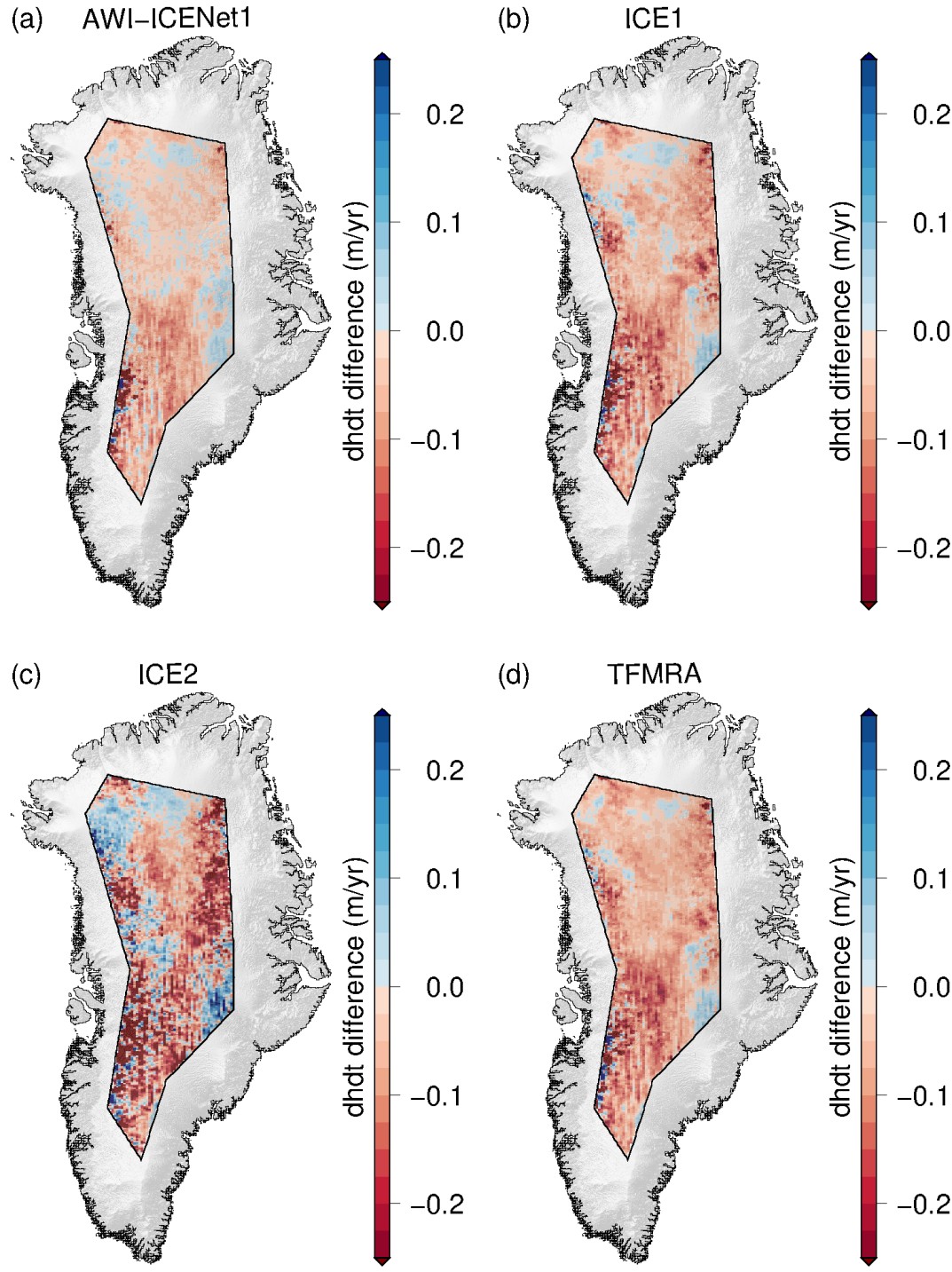

**Figure A16.** Greenland wide dhdt difference to ICESat2 of four retrackers for the period from January 2019 to December 2021 with LEW and backscatter correction

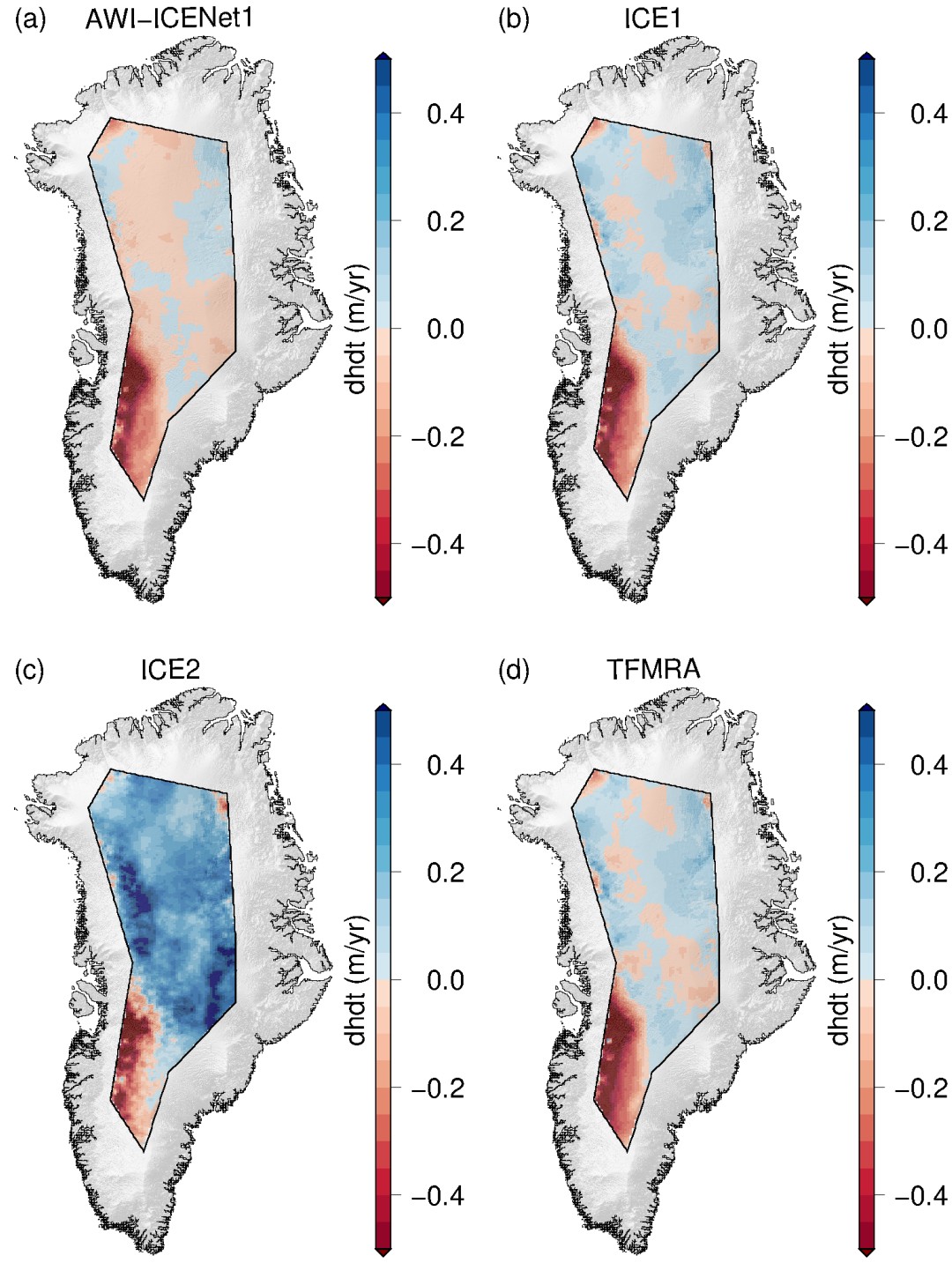

**Figure A17.** Greenland wide dhdt estimates including corrections for LEW and backscatter of four retrackers for the period from January 2011 to December 2014 , which includes the July melt event in 2012.

*Author contributions.* V.H. led the study, simulated the waveforms, applied AWI-ICENet1 to satellite data, analysed the results and contributed figures. A.D. and E.L. developed the ConvNet. A.D. analysed results and contributed figures. V.H. and A.H. wrote large parts of the text. R.H. wrote machine learning specific sections of the text. M.H. and V.H. acquired funding for this research. All authors discussed the results and contributed to editing the manuscript.

*Competing interests.* The authors declare no competing interests.

*Acknowledgements.* This work was supported by the Deutsche Forschungsgemeinschaft (DFG) in the framework of the priority programme 1158 "Antarctic Research with comparative investigations in Arctic ice areas" by a grant 442929109, by DeepSAR: Physical parameter estimation from SAR Data via joint model inversion and deep learning (grant no., ZT-I-PF-5-9, Helmholtz AI Cooperation Unit) and by the Helmholtz Association of German Research Centers as part of the Helmholtz Information and Data Science Incubator, and project "Artificial Intelligence for Cold Regions" (AI-CORE, grant no. ZT-I-0016). The network was trained using a NVIDIA A100 SXM4 80 GB GPU videocard and the tensorflow 2.15.0 / keras 2.15.0 package (Abadi et al., 2015). Waveform retracking was applied on AMD Rome Epyc 7702 cores using tensorflow 2.15.0 / keras 2.15.0 package. Acknowledging PGC Services (including data access). The REMA DEM was provided by the Byrd Polar and Climate Research Center and the Polar Geospatial Center under NSF-OPP awards 1043681, 1542736, 1543501, 1559691, 1810976, and 2129685. The ArctiCDEM was provided by the Polar Geospatial Center under NSF-OPP awards 1043681, 1559691, 1542736, 1810976, and 2129685. Access to the ArctiDEM was provided by the Polar Geospatial Center under NSF-OPP awards 1043681, 1559691, and 2129685.

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
