# Peer review of "AWI-ICENet1: A convolutional neural network retracker for ice altimetry"

_The Cryosphere, 2023_

## Author Comment (AC1)

Authors point-to-point responses Referee 'Comment on tc-2023-80', Anonymous Referee #1, 29 Sep 2023
Please find the author's responses in blue below the reviewer's comments.

Many thanks for the review, which help to improve the quality of the manuscript.

The authors have demonstrated commendable expertise and innovation in their work by introducing CNN in the field of radar altimetry retracking. Their research represents a significant contribution to the field, offering valuable insights and promising results. By employing CNN, the authors have opened new possibilities for improving the accuracy and efficiency of altimetry data processing, and possibly (not addressed) speeding up otherwise time-consuming retracking. The paper introduces this innovative approach and provides convincing evidence of its advantages. While comprehensive research is crucial for presenting a well-rounded perspective, the excessive length of the papers can be challenging for readers to digest. One suggestion would be to separate the retracking from a second paper on the applications or move some of the backgrounds to an appendix. This being said, I only have minor comments on the paper.

**Thanks for this suggestion. We are aware of the length of the paper but we don't want to split the manuscript in a retracker and application part. We need to show the improvement which can be derived with our new approach and therefore need to apply it to data and compare to other products. We therefore think that we cannot decouple the application part from our new method.**

One aspect of AWI-ICENet1 which could be addressed is the possible increased efficiency in the processing time. As the authors have great experience in retracking radar altimetry data, it would also be beneficial to highlight the efficiency of the AWI-ICENet1 compared to other methods. We are often faced with very long reprocessing times from the agencies.

**Thanks for the advice. We will add a table comparing processing times of different retracking methods.**

In general, the caption for the many figures is very shallow, please read through them and elaborate on them, so a reader who is not reading all 61 pages can follow the main conclusions of the figures.

**Yes, we do understand that it is beneficial to add more info into the caption and will follow the advice in the revised version.**

L2: "long-term observations of surface elevation change are required to", agree with the meaning but I would suggest that you also acknowledge other methods and hence remove the "Surface"

**We fully agree that it is important to highlight other methods, too. Many thanks for pointing this out! We will change it accordingly.**

L5: The snow penetration is an issue in most places, just leave this sentence open and remove "especially over the…"

**We agree, unfortunately, it is an issue in most places! We are removing 'especially over ice sheets' in the revised version.**

L15: This shows a broader application and suggests writing "This technique provides new opportunities to utilize convolutional neural networks in the processing of satellite altimetry data, which can be applied to historical, recent, and future missions."

**Thank you, we will change the sentence accordingly.**

L21: missing reference at "2010 onwards"

**We will add a reference here**

L30 may start with "Ku-band satellite altimeters…" as the mentioned satellites are all Ku.

**Thanks, we will follow your advice.**

L40 Why add the PLRM, this is a pure post-processing product.

**Thank you, we will remove this sentence as it is not necessary in the context here. However, we think the PLRM data is still very useful as it allows comparing SAR and LRM and enable to link older missions to the new SAR generation.**

L43: ICESat-2 operating at green-wavelength is penetrating the snow. And add references for ICESat-2.

**We will add a reference for ICESat-2. To our knowledge there is little known about penetration of green lasers into snow/ice . A TCD paper of Studinger et.al. (currently under review) propose a "differential penetration which likely exist in lower-level ICESat-2 data products" at least over thin or finger rafted sea ice (https://tc.copernicus.org/preprints/tc-2023-126/tc-2023-126.pdf). However, their modeling approach suggest only minor penetration in dry snow. We will add this reference here and weaken our statement that laser is not penetrating the snow firn pack.**

L46: please elaborate on the sentence "Because…",

**We will reformulate the sentence.**

L73: the common abbreviation is CNN, please consider using this

**We have chosen ConvNet on purpose and would prefer to use it in our manuscript. The reason is the following: Both CNN and ConvNets are common abbreviations / names of the same machine learning framework, e.g. Tan and Le, 2020, "EfficientNet: Rethinking Model Scaling for Convolutional Neural Networks" uses ConvNet as abbreviation for Convolutional Neural Network. Non-academic examples of the broader community showing that both terms are valid include**

**IBM: https://www.ibm.com/topics/convolutional-neural-networks**

**MathWorks: https://www.mathworks.com/discovery/convolutional-neural-network-matlab.html**

**SuperAnnotate: https://www.superannotate.com/blog/guide-to-convolutional-neural-networks**

**We prefer ConvNet since we agree with large parts of the Machine Learning community that a too strong emphasis on "neural" (as referring to the visual cortex of mammals) is misleading and bears risks. As a consequence, several leading examples (see a few below) even prefer "Convolutional Networks" instead of "Convolutional Neural Networks" which is not only more precise but also historically more correct as this was the term originally used (see e.g. LeCun and Bengio, 1995, "Convolutional Networks for Images, Speech, and Time-Series"):**

**LeCun et al.,, "Character-level Convolutional Networks for Text Classification", NeurIPS 2015**

**Zeiler and Fergus, "Visualizing and Understanding Convolutional Networks", ECCV 2014**

**Long et al., "Fully convolutional networks for semantic segmentation", CVPR 2015"**

L124: why exclude Greenland?

**We selected Antarctica to build the training data set, as it covers various kinds of flat and complex topography. Finally, we applied the saved model to observed CryoSat-2 waveforms also over Greenland. Results are very promising and show that the training over Antarctica is sufficient to improve the retracking over both ice sheets. As we show and discuss in the paper the new approach can also handle the melt event in 2012. Although possible, we therefore do not see the need to include Greenland in the training of the CNN.**

L128: to help the reproducibility of the study please give more insights into the chosen backscatter cross-section values.

**sigma_0 = 10 dB . At this stage we didn't consider any heterogeneity nor angular dependency of sigma. We will refine our the text to be clear in the revised version.**

L144: "rate in the following", the paper has a couple of these please proofread the paper once more.

**Thank you, we will proofread and correct spelling mistakes.**

L153 What is the resolution of the applied DEM in the modelling?  (it might not fit here but should be discussed in this relation)

**We used the REMA DEM with resolution of 1 km and slightly smoothed the DEM with a kernel of 3km to mirror the effect of an integrated signal within the pulse limited footprint. We tested different smoothing kernels to best match observed CryoSat waveforms.**

L177: move the specific tensorflow package to the acknowledgement, and add a reference to this library.

**Thanks for this advice. We will add this to the revised version.**

L189: With possible differences/drift in Bs in the processing baseline only one should be used. For consistency use E.

**We will rerun our processing and update products and tables accordingly as Baseline E is now fully released. By the time of writing the paper ESA still solved issue in their reprocessed products which affected data within the considered time period from 2019 to 2022.**

L194: add a reference for ATL06.005

**We will add a reference to the newly released ATL06.006 product. In addition we will rerun the ICESat2 processing and update figures and table accordingly.**

L196: Add the resolution used.

**Thanks. We will add this information to the revised version.**

L210: so it is the 2 km product which is used, why is this chosen?

**We found that 2km is the best choice to have enough data points to apply a solid regression but with less influence of topography.**

L221: "the correction adapted from Nilsson et al. (2022)" elaborates on what is used.

**We will be more precise and add the following sentence:**

**We calculate different variants of corrections for transient penetration effects by using empirical linear relations between Delta h and LEW, Delta h and backscatter, or Delta h and LEW and backscatter. Instead of doing this in the context of multi-parameter fitting as in e.g. Flament .. Simonsen .. Schröder, we do it on the level of spatially interpolated monthly anomalies of Delta h , LEW and backscatter, following the approach of Nilsson et al. (2022). We assume that changes of electromagnetic properties ..."**

F5: add a plot of the model vs. test point cloud. Maybe retracked range vs model range.

**We can add the following scatter plot, but we don't really see why this will help to further evaluate the retracker performance as already shown in Fig 7.**

[Figure]

L238: As Greenland is different and possibly more complicated it would be nice the see an ROI in Greenland.

**We applied the CP analysis for the compete LRM zone in Greenland as well as for the ROI in North Greenland, which was used in figure 14. However, as expected the Greenland results (see below) of the CP error show very similar results as Figures 8 and 9. We selected representative areas in Antarctica for the CP analysis over flat and complex topography. If needed we would suggest placing this requested additional figure in the supplements.**

[Figure]

*Figure 1: CP error analysis a region in North Greenland*

[Figure]

*Figure 2: CP analysis fort the LRM zone in Greenland*

F7: thank you for this very convincing figure. How does this look with respect to slope?

**Thanks for this suggestion. We have to think about how this can be accomplished to be meaningful. When we plot with respect to slope the influence of attenuation is dominating. We agree that a sloped surface will also slightly widen the leading edge. I guess this is what you want to see in such a figure. A suggestion would be to select a couple of attenuation bands and use those in the suggested figure.**

L265: Roemer is a better solution however the LEPTA relocation seems to improve even further, how does this affect the results?

**This is a good question. However, it is not within the scope of this paper as we focus on the improved retracking methodology. As we apply the same slope correction to all of the different retracked geolocated elevation points we avoid any influence of slope correction at least in the intra mission analysis. For the comparison to ICESat-2 we agree that the LEPTA slope correction might improve the product even more but this is out of the scope if this paper. We will add a sentence to mention this additional option to further improve the elevation change product.**

L268: monthly crossovers are a very long time span for cross-overs on ice sheets please add a lower time constraining on the timing between orbital crossing evaluated.

**Thanks for this suggestion, but we don't agree. The area covered with the LRM mode is the plateau of the Antarctic Ice sheet. Maximum elevation changes to be expected are less than +/- 1m/yr. This results in <0.08m elevation difference in a month. As the observed standard deviation is approx. 0.5m we think that uncertainties due to elevation change are not affecting the overall performance. Furthermore, if we only consider cross overs in < 10d we would considerably reduce the number of cross over points reducing the spatial coverage even more.**

**Exemplary, we run the CP error analysis with <31d, <10d and <5d conditions and found the following:**

|  | <31d | <10d | <5d |
|---|---|---|---|
| **Median CP error (m)** | 0.003 | 0.01 | 0.01 |
| **SD CP error (m)** | 0.45 | 0.46 | 0.46 |
| **valid cross points** | 126491 | 62902 | 33173 |
| **filtered cross points:** | 90 (0.07%) | 46 (0.07%) | 27 (0.08%) |

**Based on this evaluation we will stick with the old approach as it gives a better spatial coverage and thus also considers areas at lower latitudes.**

L270: how many fall within this outlier filtering (maybe in %)

**Usually less than 0.1%. Please see table above. Outlier removal is also dependent on the chosen retracker.**

F8-F10: The ESA ICE2 seems as an outlier compared to the others, suggest removing this from the plots to see the specific difference in the others.

**Thanks for this suggestion but as we consider all 4 retracker in the paper we would like to stick with the figures. We also show ESA ICE2 in all the remaining figures and tables and don't want to remove this comparison, as we think it is important to show how large the effect of retracker can be for higher level products.**

L423: "…focus on the time from January 2019 to December 2021…"

**Thanks for spotting this. Will be changed .**

F20: Is the trackiness due to errors in the CS2 or ICESat-2 data?

**The plot shows the standard deviation of the h anomaly for ICESat-2 only.**

L472: Guess this is ATL15 this should be mentioned.

**No this is not the case. We apply our dhdt processing to the ATL06.006 point cloud data product to avoid differences in the processing.**

---

## Author Response (AR1)

**Authors point-to-point responses Referee** 'Comment on tc-2023-80', Anonymous Referee #1, 29 Sep 2023 Please find the author's responses in blue below the reviewer's comments.

**Many thanks for the review, which helped to improve the quality of the manuscript.**

The authors have demonstrated commendable expertise and innovation in their work by introducing CNN in the field of radar altimetry retracking. Their research represents a significant contribution to the field, offering valuable insights and promising results. By employing CNN, the authors have opened new possibilities for improving the accuracy and efficiency of altimetry data processing, and possibly (not addressed) speeding up otherwise time-consuming retracking. The paper introduces this innovative approach and provides convincing evidence of its advantages. While comprehensive research is crucial for presenting a well-rounded perspective, the excessive length of the papers can be challenging for readers to digest. One suggestion would be to separate the retracking from a second paper on the applications or move some of the backgrounds to an appendix. This being said, I only have minor comments on the paper.

**Thanks for this suggestion. We are aware of the length of the paper but we don't want to split the manuscript in a retracker and application part. We need to show the improvement which can be derived with our new approach and therefore need to apply it to data and compare to other products. We therefore think that we cannot decouple the application part from our new method.**

One aspect of AWI-ICENet1 which could be addressed is the possible increased efficiency in the processing time. As the authors have great experience in retracking radar altimetry data, it would also be beneficial to highlight the efficiency of the AWI-ICENet1 compared to other methods. We are often faced with very long reprocessing times from the agencies.

**Thanks for the advice. We added a table in section 3.1 comparing processing times of different retracking methods and inserted the following text.**

**"As perfomance test we run the retracking on one of the CPU and GPU compute nodes of the high performance cluster at the Alfred Wegener Institute, Helmholtz Centre for Polar and Marine Research. We applied next to the TFMRA retracker the TCOG retracker and an adapted version of the functional fit of the ICE2 retracker as given in Legresy et al. (2005). To estimate the leading edge width (LEW) based on TFMRA and TCOG we run the retracking for different threshold levels (THL), reaching from 5% to 80%. For each THL a retracked position (RT) is determined. The LEW is the inverse of the linear regression coefficient and is estimated for each waveform as follows: LEW = 1/m with THL = m ∗ RT + n.**

**Results of the performance test are shown in Table 2."**

In general, the caption for the many figures is very shallow, please read through them and elaborate on them, so a reader who is not reading all 61 pages can follow the main conclusions of the figures.

**Yes, we do understand that it is beneficial to add more info into the figure caption and we followed the advice in the revised version and added more content to most of the figure captions.**

L2: "long-term observations of surface elevation change are required to", agree with the meaning but I would suggest that you also acknowledge other methods and hence remove the "Surface"

**We fully agree that it is important to highlight other methods, too. Many thanks for pointing this out! We changed the sentence in the revised version.**

*"This can be achieved by three different methods: Directly by measuring regional changes in the Earth's gravity field using the GRACE(FO) satellite missions, or indirectly by measuring changes in ice thickness using satellite altimetry, or by estimating changes of the mass budget using a combination of regional climate model data output and ice discharge across the grounding line based on multi-sensor satellite radar observations of ice velocity (Hanna, et.al., 2013). Satellite radar altimetry has been used to measure elevation change since 1992 using a combination of various missions"*

L5: The snow penetration is an issue in most places, just leave this sentence open and remove "especially over the…"

**We agree, unfortunately, it is an issue in most places! We removed 'especially over ice sheets' in the revised version.**

L15: This shows a broader application and suggests writing "This technique provides new opportunities to utilize convolutional neural networks in the processing of satellite altimetry data, which can be applied to historical, recent, and future missions."

**Thank you, we have changed the sentence accordingly.**

L21: missing reference at "2010 onwards"

**We added a couple of references here: The sentence reads now:**

*"The non-linearity of mass loss from Antarctica is driven by West Antarctica, where glacier acceleration and retreat has caused an increasing contribution from 1992 onwards (Rignot et al., 2002, 2014; Mouginot et al., 2014; Scheuchl et al., 2016; Milillo et al., 2022;Christie et al., 2023)"*

L30 may start with "Ku-band satellite altimeters…" as the mentioned satellites are all Ku.

**Thanks, we followed your advice.**

L40 Why add the PLRM, this is a pure post-processing product.

**Thank you, removed this sentence as it is not necessary in the context here. However, we think the PLRM data is still very useful as it allows comparing SAR and LRM and enable to link older missions to the new SAR generation.**

L43: ICESat-2 operating at green-wavelength is penetrating the snow. And add references for ICESat-2.

**We added references for ICESat and ICESat-2.**

**To our knowledge there is little known about penetration of green lasers into snow/ice . A TCD paper of Studinger et.al. (currently under review) propose a "differential penetration which likely exist in lower-level ICESat-2 data products" at least over thin or finger rafted sea ice (https://tc.copernicus.org/preprints/tc-2023-126/tc-2023-126.pdf). However, their modeling approach suggest only minor penetration in dry snow.**

**We changed the sentence to:**

*"Next to the radar altimeters, two laser altimeters surveyed polar areas: NASA's ICESat-1 operated from 2003-09 (Zwally and Thomas., 2014) and since 2018 ICESat-2 is operational (Markus et al., 2017; Smith et al.,2020). The great advantage of laser altimetry is the high precision of a single distance measurement and its low penetration into dry snow (Smith et al., 2020; Studinger et al., 2023)."*

L46: please elaborate on the sentence "Because…",

**We reformulated the sentence as follows:**

*"As we use data from all six available laser beams and a three-year measurement period from January 2019 to December 2021, the data coverage is exceptionally good. The advantages of high precision, dense sampling, small footprint size and low penetration depth outweigh the disadvantages of occasional data loss due to cloud cover. Therefore, we use ICESat-2-based estimates of rates of elevation change as a reference to compare our radar altimetry-based results."*

L73: the common abbreviation is CNN, please consider using this

**We have chosen ConvNet on purpose and would prefer to use it in our manuscript. The reason is the following: Both CNN and ConvNets are common abbreviations / names of the same machine learning framework, e.g. Tan and Le, 2020, "EfficientNet: Rethinking Model Scaling for Convolutional Neural Networks" uses ConvNet as abbreviation for Convolutional Neural Network. Non-academic examples of the broader community showing that both terms are valid include**

**IBM: https://www.ibm.com/topics/convolutional-neural-networks**

**MathWorks: https://www.mathworks.com/discovery/convolutional-neural-network-matlab.html**

**SuperAnnotate: https://www.superannotate.com/blog/guide-to-convolutional-neural-networks**

**We prefer ConvNet since we agree with large parts of the Machine Learning community that a too strong emphasis on "neural" (as referring to the visual cortex of mammals) is misleading and bears risks. As a consequence, several leading examples (see a few below) even prefer "Convolutional Networks" instead of "Convolutional Neural Networks" which is not only more precise but also historically more correct as this was the term originally used (see e.g. LeCun and Bengio, 1995, "Convolutional Networks for Images, Speech, and Time-Series"):**

**LeCun et al.,, "Character-level Convolutional Networks for Text Classification", NeurIPS 2015**

**Zeiler and Fergus, "Visualizing and Understanding Convolutional Networks", ECCV 2014**

**Long et al., "Fully convolutional networks for semantic segmentation", CVPR 2015"**

L124: why exclude Greenland?

**We selected Antarctica to build the training data set, as it covers various kinds of flat and complex topography. Finally, we applied the saved model to observed CryoSat-2 waveforms also over Greenland. Results are very promising and show that the training over Antarctica is sufficient to improve the retracking over both ice sheets. As we show and discuss in the paper the new approach can also handle the melt event in 2012. Although possible, we therefore do not see the need to include Greenland in the training of the CNN.**

L128: to help the reproducibility of the study please give more insights into the chosen backscatter cross-section values.

**sigma_0 = 10 dB . At this stage we didn't consider any heterogeneity nor angular dependency of sigma. We changed the text in the revised version to:**

*"For simplicity σ0 (θ) = 10 dB is chosen to be homogeneous and without any angular dependency within the radar footprint and the ..."*

L144: "rate in the following", the paper has a couple of these please proofread the paper once more.

**Thank you, we proofread and corrected spelling mistakes.**

L153 What is the resolution of the applied DEM in the modelling? (it might not fit here but should be discussed in this relation)

**We used the REMA DEM with resolution of 1 km and slightly smoothed the DEM with a kernel of 3km to mirror the effect of an integrated signal within the pulse limited footprint. We tested different smoothing kernels to best match observed CryoSat waveforms.**

**We added the following sentence:**

*"As input DEM we used a slightly smoothed (kernel size of 3 km) version of the REMA DEM (Howat et al., 2019) with pixel resolution of 1 km to mirror the effect of an integrated signal within the pulse limited footprint."*

L177: move the specific tensorflow package to the acknowledgement, and add a reference to this library.

**Thanks for this advice. We added the following text and a reference in the acknowledgments of the revised version:**

*"The network was trained using a NVIDIA A100 SXM4 80 GB GPU videocard and the tensorflow 2.15.0, keras 2.15.0 package (Abadi et.al., 2015). Waveform retracking was applied on AMD Rome Epyc 7702 cores using tensorflow 2.15.0, keras 2.15.0 package."*

L189: With possible differences/drift in Bs in the processing baseline only one should be used. For consistency use E.

**We did rerun our processing using Baseline E and updated all products, figures and tables accordingly as Baseline E is now fully released.**

L194: add a reference for ATL06.005

**We added a reference to the newly released ATL06.006 product. In addition, we did rerun our ICESat2 processing and updated all figures and table accordingly.**

L196: Add the resolution used.

**Thanks. We added this information in the revised version. The new sentence reads:**

*"Instead of using the quality flag given in the ATL06 product we filter the data based on the version 2 of the REMA Antarctic elevation model in 1 km pixel resolution (Howat et al., 2022) and version 4.1 of the ArcticDEM mosaic in 500 m pixel resolution (Porter et al., 2023)"*

L210: so it is the 2 km product which is used, why is this chosen?

**We updated our processing using a variable search radius. We found, that this is the best choice to have enough data points to apply a solid regression but with less uncertainties due to unresolved topography within the search radius, low processing cost for densely covered areas and a reduced number of unobserved pixels. We added a more detailed description in the revised version.**

*"The interpolated elevation anomaly product and rates of elevation change (dhdt) are generated using a slightly different approach as described in e.g. McMillan et al. (2016); Schröder et al. (2019); Nilsson et al. (2022). For each pixel with a size of 1 km x 1km we collect all geo-referenced data points within a variable distance ranging from 500 m to 2500 m (step width 500 m) and correct for topography using a bilinear interpolation of the REMA-DEM and/or ArcticDEM, respectively, rather than fitting any kind of linear or quadratic surface as McMillan et al. (2016); Schröder et al. (2019); Nilsson et al. (2022). The variable search radius is enlarged step wise until a threshold of number of points is reached. This threshold is defined to match at least 75 % of the selected time period (n_months) and the following criteria: For CryoSat-2: nmonths ∗ 12 and and due to the higher data coverage of six beams and less along track point spacing for ICESat2: nmonths ∗48 . This kind of processing allows to minimize uncertainties due to unresolved topography within the search radius but keeping enough data points for the linear regression as the search radius is tried to keep as low as possible. Processing costs for pixels with very dense data coverage in the interior of Antarctica are kept low as only a small radius and thus less data points are selected. At the same time less unobserved pixels in areas of coarse data coverage remain as the search radius can be enlarged up to 2500\,m. We then estimate rates of elevation change using a linear regression for each pixel with sufficient data coverage (criteria: max(time)−min(time) > 50 % of selected time period and npoints > nmonths), again without using additional information such as LEW, TeS, backscatter or seasonal components."*

L221: "the correction adapted from Nilsson et al. (2022)" elaborates on what is used.

**We will be more precise and added the following sentence:**

*"We calculate different variants of corrections for transient penetration effects by using empirical linear relations between Delta h and LEW, Delta h and backscatter, or Delta h and LEW and backscatter. Instead of doing this in the context of multi-parameter fitting as in e.g. Flament .. Simonsen .. Schröder, we do it on the level of spatially interpolated monthly anomalies of Delta h , LEW and backscatter, following the approach of Nilsson et al. (2022). We assume that changes of electromagnetic properties ..."*

F5: add a plot of the model vs. test point cloud. Maybe retracked range vs model range.

**We can add the following scatter plot, but we don't really see why this will help to further evaluate the retracker performance as already shown in Fig 7.**

**Therefore, we leave this out in the revised version as long as a discission is made that we need to include this figure.**

[Figure]

L238: As Greenland is different and possibly more complicated it would be nice the see an ROI in Greenland.

**We applied the CP analysis for the compete LRM zone in Greenland as well as for the ROI in North Greenland, which was used in figure 14. However, as expected the Greenland results (see below) of the CP error show very similar results as Figures 8 and 9 as we selected representative areas in Antarctica for the CP analysis over flat and complex topography. We followed your suggestion and placed the following figures in the appendix.**

[Figure]

*Figure 1: CP error analysis a region in North Greenland*

[Figure]

*Figure 2: CP analysis fort the LRM zone in Greenland*

F7: thank you for this very convincing figure. How does this look with respect to slope?

**Thanks for this suggestion. We have to think about how this can be accomplished to be meaningful. When we plot with respect to slope the influence of attenuation is dominating and**

a any slope dependency cannot be resolved. However, we think that a sloped surface will slightly widen the leading edge. Our suggestion is to select a couple of attenuation rates and plot the difference to the reference with respect to slope for each of the attenuation rates as shown below in Figure 3. As supposed the effect of slope is only marginal (a slight positive trend is observed for TFMRA at low attenuation rates). We don't think that this Figures needs to be included in the manuscript but if needed we could add it to the Appendix.

[Figure]

*Figure 3 Effect of slope for selected attenuation rates shown for TFMRA and AWI-ICENet1 retracker.*

L265: Roemer is a better solution however the LEPTA relocation seems to improve even further; how does this affect the results?

**This is a good question. However, it is not within the scope of this paper to compare different slope correction as we focus on the improved retracking methodology. As we apply the same slope correction to all of the different retracked geolocated elevation points we avoid any influence of slope correction at least in the intra mission analysis. For the comparison to ICESat-2 we agree that the LEPTA slope correction might improve the product even more but this is out of the scope if this paper. We will add a sentence to mention this additional option to further improve the elevation change product.**

**In the revised version we added a reference to LEPTA and the following sentence:**

*"Li (2022) developed the LEPTA method, an improved version of the relocation slope correction which includes points in the underlying DEM that contribute to the rise of the leading edge. Their results show an improved cross point error between CryoSat-2 and ICESat-2 compared to the method of Roemer (2007). However, as we only consider intra-mission cross point errors and apply the same slope correction to all retracker solutions the slope correction method does not play any role in our CPE analysis. "*

L268: monthly crossovers are a very long time span for cross-overs on ice sheets please add a lower time constraining on the timing between orbital crossing evaluated.

**Thanks for this suggestion, but we don't agree. The area covered with the LRM mode is the plateau of the Antarctic Ice sheet. Maximum elevation changes to be expected are less than +/- 1m/yr. This results at maximum in <0.08m elevation difference in a month. As the observed standard deviation is approx. 0.5m we think that uncertainties due to elevation change are not affecting the overall performance. Furthermore, if we only consider cross overs in < 10d we would considerably reduce the number of cross over points and reduce the spatial coverage even more.**

**Exemplary, we run the CP error analysis with <31d, <10d and <5d conditions and found the following:**

| | <31d | <10d | <5d |
|---|---|---|---|
| **Median CP error (m)** | 0.003 | 0.01 | 0.01 |
| **SD CP error (m)** | 0.45 | 0.46 | 0.46 |
| **valid cross points** | 126491 | 62902 | 33173 |
| **filtered cross points:** | 90 (0.07%) | 46 (0.07%) | 27 (0.08%) |

**Based on this evaluation we will stick with the old approach as it gives a better spatial coverage and thus also considers areas at lower latitudes.**

L270: how many fall within this outlier filtering (maybe in %)

**Usually less than 0.1%. Please see table above. Outlier removal is also dependent on the chosen retracker.**

F8-F10: The ESA ICE2 seems as an outlier compared to the others, suggest removing this from the plots to see the specific difference in the others.

**Thanks for this suggestion but as we consider all four retracker in the paper we would like to stick with the figures. We also show ESA ICE2 in all the remaining figures and tables and don't want to remove this comparison, as we think it is important to show how large the effect of retracker can be for higher level products.**

L423: "…focus on the time from January 2019 to December 2021…"

**Thanks for spotting this. Will have changed it in the revised version.**

F20: Is the trackiness due to errors in the CS2 or ICESat-2 data?

**The plot shows the standard deviation of the h anomaly for ICESat-2 only. With the reprocessing the figure was updated and show less trackiness.**

L472: Guess this is ATL15 this should be mentioned.

**No this is not the case. As mentioned in the text we apply the same dhdt processing to the ATL06.006 point cloud data product as we do for the Level 2 relocated and retracked CryoSat-2 point cloud to avoid differences in the processing and also to be able to generate the SD anomaly analysis, which is based on monthly anomalies. ATL15 is based on quarterly and not monthly products and only provides trends of elevation change over different time periods. For completeness and as comparison we added the volume change estimates based on the three years time period trend product given in ATL15.006 in table 4 and 5.**

**Authors point-to-point responses Referee** 'Comment on tc-2023-80', Anonymous Referee #2, 9 Nov 2023 Please find the author's responses in blue below the reviewer's comments.

**Many thanks for the review, which helped to improve the quality of the manuscript.**

**General comments:**

This paper presents a convolutional neural network (CNN) approach to measure and quantify surface elevation change in Greenland and the Antarctic ice sheets via satellite radar altimetry data. Through extensive analysis, the authors show that their proposed method displays improved performance and reduced uncertainty over traditional retrackers.

The primary strengths of this paper are in the thoroughness of analysis of the performance of AWI-ICENet1 and in comparisons to conventional retracking algorithms. Cross point error analysis is a good way of comparing the performance of each method for identifying the ice surface, as it does not rely on a ground truth (as is typical in supervised machine learning).

Another strength of the paper is the construction of a synthetic dataset that, after training a CNN on it, performs at least as well as (if not better than) conventional methods. It is an impressive contribution in itself to be able to construct a synthetic dataset that is sufficiently close in distribution to the training and testing data such that a deep learning model can be adequately trained on the synthetic data alone.

**Specific comments:**

Despite the strengths and contributions, my main concern for this paper is that it does not situate itself within the context and literature of deep learning approaches applied on data from satellite or airborne sounding of ice sheets. To my knowledge, the majority of this work has involved using deep learning to track ice and bedrock layers beneath the ice surface, but these approaches still seem quite relevant, at least to briefly discuss. These are some such prior works:

1. S. Dong, X. Tang, J. Guo, L. Fu, X. Chen, and B. Sun, "EisNet: Extracting bedrock and internal layers from radiostratigraphy of ice sheets with machine learning," IEEE Trans. Geosci. Remote Sens., vol. 60, pp. 1–12, 2021.
2. M. Liu-Schiaffini, G. Ng, C. Grima, and D. Young. "Ice thickness from deep learning and conditional random fields: application to ice-penetrating radar data with radiometric validation," IEEE Trans. Geosci. Remote Sens., vol. 60, pp. 1-14, 2022.
3. M. H. Garcia, E. Donini, and F. Bovolo, "Automatic segmentation of ice shelves with deep learning," in Proc. IEEE Int. Geosci. Remote Sens. Symp., Jul. 2021, pp. 4833–4836.
4. H. Kamangir, M. Rahnemoonfar, D. Dobbs, J. Paden, and G. Fox, "Deep hybrid wavelet network for ice boundary detection in radra imagery," in Proc. IEEE Int. Geosci. Remote Sens. Symp. (IGARSS), Jul. 2018, pp. 3449–3452.
5. R. Ghosh and F. Bovolo, "TransSounder: A hybrid TransUNet-TransFuse architectural framework for semantic segmentation of radar sounder data," IEEE Trans. Geosci. Remote Sens., vol. 60, pp. 1–13, 2022.
6. E. Donini, F. Bovolo, and L. Bruzzone, "A deep learning architecture for semantic segmentation of radar sounder data," IEEE Trans. Geosci. Remote Sens., vol. 60, pp. 1–14, 2021.
7. Y. Cai, S. Hu, S. Lang, Y. Guo, and J. Liu, "End-to-end classification network for ice sheet subsurface targets in radar imagery," Appl. Sci., vol. 10, no. 7, p. 2501, Apr. 2020.

The authors discuss some prior machine learning methods applied to the cryospheric sciences, but this discussion only includes one deep learning approach (Fayad et al. (2021)). I would recommend that the authors include a brief discussion of what distinguishes Fayad et al. (2021)'s setting/model from the current paper. I would also recommend the authors incorporate an additional discussion of the above (and related) references on page 3, or where relevant.

**Thank you for this comprehensive list of additional references. We inserted them in the introduction as ML methods used in radar stratigraphy as most of them are dealing with identifying bed rock and/or internal layers or are used for classification of different ice regimes in radar images. We also added a brief discussion on the differences of our approach and the one of Fayad et.al. and tried to make clear how our approach is different to Fayad.**

**We inserted at page 3 the following paragraph by introducing mentioned references:**

*"In recent years Machine learning has been applied to various kind of image data in polar areas. E.g. Loebel et al. (2022, 2023) monitored calving front motion at sub-seasonal resolution for 23 Greenlandic outlet glaciers using a U-Net deep learning ap-plication (Ronneberger et al., 2015) on multi-spectral Landsat-8 imagery data. Baumhoer et al. (2019) extracted automaticallyAntarctic Glacier and Ice Shelf Fronts from Sentinel-1 Imagery by using U-net deep Learning to create a dense time series of the Antarctic coastline to assess calving front change. Mohajerani et al. (2021) used fully-convolutional neural network to auto-matic delineate glacier grounding lines in differential interferometric synthetic-aperture radar data. They applied their approach to more than 20000 interferograms along the Getz Ice Shelf, in West Antarctica and demonstrate that grounding zones are one order of magnitude wider than expected. Beside satellite imagery airborne or ground based radar images of Ice-Penetrating Radar systems has been extensively studied and new insights could be achieved through application of machine learning ap-proaches in recent years. Liu-Schiaffini et al. (2022) propose a deep learning model based on convolutional neural networks and continuous conditional random fields (CCRFs) to automate ice bed identification. They deployed their approach to high-capability radar sounder (HiCARS) radargrams and were able to capture the global ice bed geometry as well as identifying fine-grained basal details even in areas with complex and rough ice bed conditions. Kamangir et al. (2018) presented a deep hybrid wavelet network for detecting ice surface and bottom boundaries, compared it with other edge detection approaches by using the NASA Operation IceBridge Mission data set . Dong et al. (2022) designed a Neural Network Fusion, called EisNet to extract next to the Bedrock also Internal Layers from Radiostratigraphy data. Eisnet composes of three coupled deep neural networks which are based on U-net architecture (Ronneberger et al., 2015). Other application of Machine learning approaches deal with the segmentation of different structures in radargrams. E.g. García et al. (2021) developed an automatic analysis technique based on W-Net (Xia and Kulis, 2017), a fully convolutional autoencoder to distinguish floating ice over ice shelves from grounded ice in coastal areas in radargrams recorded with the Multichannel Coherent Radar Depth Sounder MCoRDS2. Another segmentation scheme to segment Radargrams into englacial layers, bedrock, basal units, and noise-limited regions such as the echo-free zone (EFZ) is based on a U-Net with attention gates and the Atrous Spatial Pyramid Pooling (ASPP) module is proposed by Donini et al. (2022). Their focus is the identification and mapping of basal layer and basal units and the network was successfully applied to two datasets acquired in North Greenland and West Antarctica using the MCoRDS3 data set. A very similar approach was developed by (Cai et al., 2020) using bilateral filtering to reduce noise and a deep residual learning (He et al., 2016) as well as the ASPP module to classification free space, internal layers, bedrock, and noise (including EFZ region) and applied it to MCoRDS and MCoRDS2 radar images acquired between 2009 and 2011 in Antarctica. Finally, Ghosh and Bovolo (2022) constructed the TransSounder a hybrid TransUNet-TransFuse architectural framework to systematically characterize the different subsurface targets and compared it to other state of the art framweorks by using a MCoRDS radar depth sounder dataset. All the above mentioned ML approaches are using images or two dimensional data sets as input and thus differ from the classical one dimensional echoes or waveforms detected by satellite altimetry. However, Machine learning has been applied in various other studies for waveform analysis ...."*

Most of these prior approaches applying CNNs to identify ice and bedrock layers beneath the ice surface use 2D CNNs to capture spatial correlations in the along-track direction. However, to my understanding AWI-ICENet1 only performs 1D convolutions in the radar return at a specific waveform in time. Why was this design choice made? It seems likely that capturing spatial correlations could aid the prediction of a deep learning model, especially in regions where data is noisy and measurements are highly variable. Please add a discussion/comparison of AWI-ICENet1 to prior 2D CNNs methods in the paper.

Thank you for raising this question.

Our choice for a 1D CNN has several reasons. First, it is simple and fast and can directly be applied to level 1B waveform data without any preprocessing. Second, our focus is the individual waveform and not like in radar stratigraphy continuous layering or bedrock. Third, a 2D representation of subsequent altimeter waveforms as a radargramm look much different to airborne soundings as the receiving range window is adjusting to topographic changes. This means that the position of the waveform within the window can suddenly jump. A 2D CNN would try to use spatial or alongtrack correlations and would possibly misinterpret such jumps. We agree that spatial correlations could help to handle noisy measurements and we could imagine that especially for coastal or ocean altimetry a 2D approach might be suitable as well, as sudden waveform jumps are not expected or could be reduced by a preprocessing which shifts the waveform to a constant range gate. However, over the ice sheet this not appropriate as we face large elevation differences of a couple of hundred meters along track. Lastly "our simple" 1D approach shows very good results and we don't see a need to make it more complicated. In addition our 1D single parameter waveform based approach can be extended to a multi parameter approach to gather more information than just the retracking point from the waveform itself, as shown by Fayad et.al.

We addressed your point in connection with a more detailed comparison to the study of Fayad et. al. by inserting the following text in the revised version:

*„However, the regression task to accurately estimate surface elevation has been barely addressed. Fayad et al. (2021) used DL for the detection of surface heights from space-borne laser altimeter data of the GEDI mission (Dubayah et al., 2020). Fayad et al. (2021) used two ConvNets, a one dimensional for the individual waveform and reshaping it into two-dimensional representation to constrain biophysical parameters, such as canopy height and wood volume. Their results confirm, that ConvNets can be used to extract useful information from LiDAR waveforms and compare well with classical but complex and expensive random forest methodologies. Furthermore, Fayad et al. (2021) find that the 1D representation of the waveform produced slightly less accurate results than its 2D counterpart, both, for single and multi-parameter output (estimation of canopy height and wood volume at the same time). They argue, that the reason for this being a larger gradient around an information peak, such as a vegetation or ground return, is generally larger in the 2D representation of the waveform. As the data set contains peaks and the aim is to detect those peaks, the filters of the 2D-ConvNet model are better adapted to recognize signal content which are concentrated in small areas with high signal contrast, (Fayad et al., 2021).*
*However, over ice sheets we only deal with one prominent return waveform, which is an integrated signal originating from a large footprint with a diameter of roughly 15 km including contributions from of the upper snow/firn layer up to a depth of less than 10m,. Therefore, signal gradients are not as large and single or multi peak waveform are usually only occurring in very complex terrain. Furthermore, the noise level for a single radar waveform is much higher than for a LiDAR waveform, which results in noise peaks on top of the gentle signal. In addition, our application developed for satellite radar altimetry, is also very contrasting to typical application of 2D DL approaches such as layer or feature detection, or classification within images (radargrams) recorded by radar depth sounders. Those systems can penetrate up to 4km of ice and thus are capable to provide detailed information of internal structures, bed rock as well as basal features within the recorded radargrams. Here 2D ConvNets are used to capture spatially correlated signals in the along-track direction. Since the receive range*

*window of a satellite radar altimeter is adjusted to follow the terrain by the on-bord tracker, consecutive waveforms are not necessarily aligned and may jump within the radar range window, especially when the satellite samples changing undulating surfaces such as ice sheets. This can lead to erroneous results when using a 2D Convnet that captures spatially correlated signals. However, over the open ocean or in coastal altimetry applications, a 2D approach could be promising. Since neither peak detection, image classification nor spatial correlated layer detection is the objective of our approach, we decided that a 1D representation of the ConvNet is sufficient to accomplish our task of accurately determining or retrack the beginning of the leading edge of a single waveforms.*

*As peak detection, image classification nor layer detection is the objective of our approach we decided that a 1D representation of the ConvNet should be sufficient to fulfill our task to retrack the beginning of the waveforms leading edge. We use single waveforms of CryoSat-2 and represent them as sequential data to a 1D ConvNet that applies a series of processing layers (in particular  convolutions with learned kernels along the time dimension of  the waveform) to automatically extract features and agglomerate information. The output of the network is the retracked range that corresponds to the snow/firn surface."*

The authors motivate the use of a synthetic dataset by discussing how ground truth data cannot be obtained by using airborne or ground-borne sounders due to the different footprint sizes. While the answer may be clear to someone in the cryospheric community, some members of the machine learning community may ask why the ground truth cannot simply be set to be the output from a retracking algorithm that the CNN can simply learn to approximate (albeit potentially improving runtime). I would recommend that the authors briefly address this question in the introduction as well.

**Thanks for this advice. We briefly addressed this and tried to make clear that a radar satellite measurement is an integrated signal over a large area including an unknown and variable signal contribution from the subsurface. The topography in this area is dominating the waveform shape and the volume contribution changes the shape and especially the leading-edge width as well. Thus, the ConvNet cannot simply provide a surface elevation when it is trained by point measurements as obtained by ground truth data like GNSS, airborne laser or satellite laser measurements.**

**This text is added to the revised version:**

*"In order to engage supervised machine learning for processing of the satellite radar altimeter waveforms, a large data set with known range is needed. In contrast to (Fayad et al., 2021) who trained their models on a subset of GEDI waveforms, where ground truth measurements existed, a ground-truth based learning approach cannot be achieved here. The area covered by airborne or ground-borne soundings of the ice surface using laser scanners or GNSS traverses are orders of magnitudes smaller than those of satellite measurements. Space borne laser altimetry as ICESat-2 to be used as test data set in a DL approach to improve radar derived elevation measurements is in our opinion also not applicable. The reasons for this are the very different footprints of the two systems. While the ICESat-2 laser points to areas of less than 0.02 km^2, satellite radar altimeters illuminate large areas of up to 10 $km_2$ , so that the two are not spatially assigned and cannot be directly compared with each other. Even more, the large scale topographic undulation and surface slope influence the waveform shape but also involve a slope correction in the post processing to reposition the radar elevation measurement to its point of closest approach. As this correction cannot be extracted from the waveform shape itself a direct comparison between laser or GNSS derived surface elevation and radar derived elevation as a gound truth for a DL approach is not possible"*

Can the authors also provide a brief description/comparisons of runtimes between the algorithms?

**Yes, we added a table in section 3.1 showing the runtimes of the different algorithms**

**Technical corrections:**

There are several typos in the paper, and some of the language is unclear; please proofread the paper closely again. For instance, there are

**We have proofread the paper.**

two typos in line 144

**Thanks, we have changed this in the revised version.**

line 30 "esa" should be "ESA."

**Thanks, we have changed this in the revised version.**

On line 270, there seems to be an extra $x$.

**Thanks, we have changed this in the revised version.**

In lines 504-505, it is unclear what is meant by "the nature of things."

**We have change the sentence to:**

**The reason that the correlations are lower for AWI-ICENet1 is that the seasonal h-anomalies are already strongly suppressed, resulting in a much lower correlation with backscatter or LEW and thus reducing the correction.**

The wording in line 93 should also be tweaked for grammar and combined with the previous sentence: "Reason is the very different footprint size of the two systems."

**We have change the sentence to:**

**The reason for this are the very different footprints of the two systems. While the ICESat-2 laser points to areas of less than 0.02 km2 , satellite radar altimeters illuminate large areas of up to 10 km2 , so that the two are not spatially assigned and cannot be directly compared with each other.**

---

## Author Response (AR2)

**Authors point-to-point responses to Editor** 'Comment on tc-2023-80', 07 May 2024

Please find the author's responses in blue below the editor's comments.

**Many thanks for the positive response and additional hints to improve the quality of the manuscript.**

My main point for the minor revision is that I think it is important to address the performance of the method as well in the discussion. Both reviewers rightly expected the AWI-ICENet1 to be much faster than traditional methods (as is often a motivation for machine learning methods) so the fact that the method is often (way) slower than traditional retrackers needs to be highlighted and explained/discussed in more detail in the discussion.

For the minor revision, I would like to ask you now to re-upload the:
1. revised version of the manuscript
2. track change version with changes highlighted.

**Many thanks for this advise:**
**We added a small chapter of the  Computational performance in the discussion section 4.2.5 to address this issue:**

*"Unlike many other machine learning applications used in remote sensing, AWI-ICENet1 was not designed to replace manual labour or save time, but to improve observation quality. In fact, our computational performance test results, shown in Table 2, highlight that AWI-ICENet1 requires more processing time than conventional cheap empirical retracking methods. In particular, AWI-ICENet1 requires about 8 and 18 times as much computing time compared to the low computing costs of empirical retracking methods such as TFMRA and TCOG, respectively. However, if the estimation of the LEW is also taken into account in TCOG and TFMRA, the difference in processing time compared to AWI-ICENet1 is considerably reduced. Therefore, we do not consider the higher computational cost of AWI-ICENet1 to be a significant drawback for its use, regardless of the computing infrastructure. Compared to more complicated waveform fitting methods based on analytical descriptions of the waveform, such as dedicated classical ocean retrackers like MLE3/4 (Amarouche et al., 2004; Thibaut et al., 2010), adaptive MLE3/4 (Thibaut et al., 2021), SAMOSA+ (Dinardo et al., 2018) or SAMOSA++ (Dinardo et al., 2021), we would expect that CNN-based approaches can help to significantly reduce processing costs. Re-processing campaigns could benefit from neural network-based approaches. However, for each of the analytical re-trackers mentioned above, a specific CNN model that best represents the analytical solution would have to be trained in advance with a considerable number of waveforms covering the entire spectrum of possible waveforms."*

In addition, we uploaded the dataset to PANGAEA and received a DOI: 10.1594/PANGAEA.964596.

Many thanks for handling the manuscript.

Best regards

Veit Helm